# Dissolvr: An Interpretable and Fast Framework for Aqueous and Organic Solubility Prediction

**Vansh Ramani** [1] **Har Ashish Arora** [1] [*] **Dhairya Kuchhal** [1] [*] **Sayan Ranu** [1] **Tarak Karmakar** [1]

## Abstract

High-fidelity solubility prediction is fundamental to pharmaceutical development and environmental partitioning, where accurate modeling must couple molecular structure with thermodynamic behavior across diverse chemical environments. However, recent advancements have been dominated by deep learning architectures that often sacrifice physical interpretability for predictive power. We challenge this trend by showing that state-of-the-art performance does not require such non-transparent architectures. To address this, we introduce Dissolvr, a transparent framework for molecular solubility prediction. In addition, we perform a comprehensive literature review and a benchmarking study against various methods. We show that Dissolvr approaches the aleatoric limit of experimental uncertainty and achieves OOD generalization through structural invariance, derived by mapping molecules to physically-grounded descriptors. Then, we present an LLM-assisted post-hoc explanation pipeline that bridges the gap between symbolic model artifacts and chemically grounded narratives. Finally, a comparative benchmark of a survey involving 22 expert chemists reveals that expert evaluators provide deep insights.

## 1. Introduction and Related Works

Predicting solubility is a fundamental problem in chemistry, necessitating the development of predictive models. The field lies on the critical path of pharmaceutical and chemical development (Hopfinger et al., 2009; Llinas et al., 2020; Boobier et al., 2020). Poor aqueous solubility is a major developmental hurdle: it plagues nearly 40% of currently marketed drugs and contributes to the failure of up to 90% of active candidates during preclinical and clinical development (Kalepu & Nekkanti, 2015). Thus accurate solubility prediction is essential for optimizing drug formulations (Savjani et al., 2012) and reliably assessing bioavailability (Kumari et al., 2023). Moreover, solubility is a key determinant of the environmental fate of chemicals, controlling the mobility, persistence, and accumulation of contaminants in ecosystems (National Research Council, 2014; Khan et al., 2020). Despite this, experimental measurements remain prohibitively time-intensive and error-prone (Murdande et al., 2011), necessitating computational models across diverse media.

Solubility prediction has evolved from relatively simple empirical equations such as the General Solubility Equation (GSE) (Yalkowsky & Valvani, 1980) and the ESOL model (Delaney, 2004), to deep learning architectures trained on large-scale curated datasets such as AqSolDB (Sorkun et al., 2019), BigSolDB (Krasnov et al., 2023), and BigSolDB2.0 (Krasnov et al., 2025). Current state-of-the-art models, including FastSolv (Attia et al., 2025), and Solvaformer (Broadbent et al., 2025) push accuracy toward the 'aleatoric limit', the irreducible noise floor of experimental variability (Palmer & Mitchell, 2014). However, recent surveys (Llompart et al., 2024; Attia et al., 2025) identify critical gaps: deep models sacrifice transparency for marginal gains, incur prohibitive computational costs, and often exhibit performance inflation from inconsistent curation.

A more thorough review of literature is given in Appendix A.

**Contributions:** We present Dissolvr, demonstrating state-of-the-art solubility prediction requiring neither expensive computation nor deep learning. Dissolvr (i) achieves faster training and inference than deep learning baselines, (ii) enforces thermodynamic consistency under the standard endothermic-dissolution regime via hard monotonic constraints, and (iii) produces natural-language explanations grounded in model evidence. Our contributions are:

- **Comprehensive study and open-source benchmarking.** We conduct an extensive field review and establish reproducible evaluation protocols across multiple publicly available datasets. We demonstrate that many published models underperform their reported metrics when stan-

---

[*]Equal contribution. [1]Indian Institute of Technology, Delhi, India. Correspondence to: Har Ashish Arora <harashish.cs124@cse.iitd.ac.in>.

*Proceedings of the 43rd International Conference on Machine Learning*, Seoul, South Korea. PMLR 306, 2026. Copyright 2026 by the author(s).

dardized curation is applied. We identify multiple works reporting performance beyond the aleatoric limit ($\sim$0.6-0.8 $\log S$), which is physically implausible and indicative of curation artifacts or information leakage.

- **State-of-the-art transparent architecture** We introduce DISSOLVR, a Gradient Boosted Decision Tree framework that replaces non-transparent architectures with explicitly engineered molecular descriptors derived from structural information. DISSOLVR achieves superior predictive accuracy while remaining computationally efficient and transparent.
- **Physics-aligned dual-regime modeling.** We propose a unified strategy: (i) a descriptor-only regressor for single-solvent systems, (ii) an Interaction Layer for multi-solvent environments to model solute–solvent coupling, and (iii) enforced monotonic constraints on temperature features.
- **LLM-assisted post-hoc explanations and validation.** We augment an LLM-assisted explainer that maps symbolic model artifacts, such as decision paths and interaction layer coefficients, into chemically grounded narratives. A six-stage guardrail pipeline (Sec. 4) hides the predicted $\log S$ until the symbolic evidence is analysed first We conduct a small evaluation study with 22 expert chemists to analyse how such explanations are interpreted and acted upon in practice; the resulting observations are discussed in Section 5.6.

## 2. Preliminaries and Problem Formulation

**Definition 2.1** (Thermodynamic Solubility)**.** Solubility $S$ is defined as the maximum concentration of a solute $\mathcal{A}$ in a solvent $\mathcal{B}$ at thermodynamic equilibrium (fixed $T, P$), where the chemical potentials of the crystalline and solvated phases are equal. The standard Gibbs free energy change of solution, $\Delta G^o_{\text{sol}}$, under dilute solution conditions, is related to the equilibrium solubility $S$ as $\Delta G^o_{\text{sol}} = -RT \ln S + C$, where $R$ is the gas constant, $T$ is the temperature, and $C$ is a constant determined by the choice of the standard state (Boothroyd et al., 2018; Palmer & Mitchell, 2014).

While this provides the formal definition of solubility, approaches to accurately calculate $\Delta G^o_{\text{sol}}$ as a physical property typically require calculation of computationally-expensive quantum-mechanical descriptors (Boothroyd et al., 2018). We thus use the linear relationship between $\Delta G^o_{\text{sol}}$ and $\log S$ to estimate solubility.

Following common practice in solubility modeling and benchmark construction (*e.g.*, in the ESOL dataset derived from (Delaney, 2004)), we represent solubility as $\Sigma = \log_{10}(S)$, where $S$ is measured in mol L$^{-1}$.

We define a molecular solubility prediction task as follows.

**Definition 2.2** (Molecular Solubility Task)**.** Let $\mathcal{X}_\mathcal{A}$ and $\mathcal{X}_\mathcal{B}$ denote the feature representations of a solute $\mathcal{A}$ and a

solvent $\mathcal{B}$, respectively. The task of molecular solubility prediction is to learn a mapping $f : (\mathcal{X}_\mathcal{A}, \mathcal{X}_\mathcal{B}, T) \to \Sigma$, where $T$ represents the absolute temperature of the system.

To ensure that this learned mapping $f$ remains computationally feasible, we define descriptors for the solute and solvent, $\mathcal{X}_\mathcal{A}$ and $\mathcal{X}_\mathcal{B}$, respectively, which are easy to compute in comparison to descriptors that require the modelling of complex physical systems.

In this work, we distinguish between two distinct predictive regimes - (i) solubility prediction in a fixed solvent and (ii) solubility prediction in multiple solvents.

**Problem 2.1** (Solubility Prediction in a Single Solvent)**.** Given a fixed solvent $\mathcal{B}$ (*e.g.*, water), the task of prediction reduces to learning a mapping $f_\mathcal{B} : (\mathcal{X}_\mathcal{A}, T) \to \Sigma$.

**Problem 2.2** (Solubility Prediction in Multiple Solvents)**.** When both the solute $\mathcal{A}$ and solvent $\mathcal{B}$ vary, the prediction task requires modelling the interactions $\mathcal{X}_\mathcal{I}$ between the compounds. This involves defining an interaction mapping: $\mathcal{I} : (\mathcal{X}_\mathcal{A}, \mathcal{X}_\mathcal{B}) \to \mathcal{X}_\mathcal{I}$. The mapping $f$ then incorporates these interactions alongside the individual component features and temperature: $f : (\mathcal{X}_\mathcal{A}, \mathcal{X}_\mathcal{B}, \mathcal{X}_\mathcal{I}, T) \to \Sigma$ [1].

**Definition 2.3** (Aleatoric Limit)**.** Defined as the noise floor $\epsilon_{noise}$, representing the inter-laboratory variability and experimental stochasticity reported across independently measured solubility studies. This has been estimated to lie between $\sim 0.6$ and $0.8 \log_{10}(S)$ units for heterogeneous, large-scale data (Attia et al., 2025; Palmer & Mitchell, 2014) like AqSolDB (Sorkun et al., 2019) and BigSolDB (Krasnov et al., 2023; 2025). Because this estimate is derived from inter-laboratory disagreement on matched compounds, it is an upper bound on the true irreducible noise; intra-laboratory replicate variance (when reported) is typically lower (App. D). For datasets arising from singular sources (*e.g.*, ESOL (Delaney, 2004)), the aleatoric limit is undefined.

Formally, the expected Mean Squared Error (MSE) for any model $M(\mathbf{x})$ is lower-bounded by the conditional variance :

$$\mathbb{E}_{\mathbf{x},y} \left[ (y - M(\mathbf{x}))^2 \right] \geq \mathbb{E}_\mathbf{x} \left[ \text{Var}(y|\mathbf{x}) \right] = \epsilon^2_{noise} \quad (1)$$

This bound implies an irreducible RMSE floor of $\epsilon_{noise}$, which arises from inherent measurement inconsistencies and experimental variability and sets an effective performance limit on any predictive model regardless of capacity [2].

Alongside accurate predictions within the aleatoric limit, we

---

[1] Strictly speaking, the term $\mathcal{X}_\mathcal{I}$ is not required - we augment this as a term to $f$ to separate the significance of interaction terms with the features of the individual molecules.

[2] While real-world experimental solubility noise is inherently heteroscedastic, treating $\epsilon_{\text{noise}}$ as a single constant provides an approximation of the expected irreducible noise floor across the dataset. See Appendix D.1.

also wish to develop a model that is transparent. We thus define a model as **symbolically accessible:**

**Definition 2.4** (Symbolic Accessibility). A predictive model $F : \mathcal{X} \rightarrow \mathcal{Y}$ is symbolically accessible if it can be expressed as a weighted sum of $K$ logical predicates:

$$f(x) = \sum_{k=1}^{K} w_k \cdot \mathbb{I}(P_k(x)) \qquad (2)$$

where $w_k \in \mathbb{R}$ and each $P_k(x)$ is a Boolean conjunction of literal constraints (*e.g.*, $x_i \leq \theta$). This structure ensures that the decision boundary is a deterministic aggregation of explicit, human-readable rules, decomposable into explicit rule predicates, in contrast to deep learning or GNN architectures. Such models are thus transparent by decomposition.

## 3. Proposed Methodology: DISSOLVR

We propose that an ideal model should have three attributes: (i) it achieves accuracy close to the aleatoric limit, (ii) its features and training complexity are computationally inexpensive, and (iii) it is symbolically accessible (Definition 2.4).

Guided by these three objectives, we introduce DISSOLVR. We adopt an explicit, descriptor-driven representation where each feature encodes a simple, physically interpretable property and maps these descriptors to solubility using Cat-Boost (Prokhorenkova et al., 2018), leveraging *oblivious trees* for symmetric splitting and *ordered boosting* for bias reduction. We also assume a standard endothermic dissolution model for thermodynamic consistency.

**Physicochemical Featurizer:** DISSOLVR's descriptors are grounded in 3 complementary physicochemical pillars:

1. **Moietal Identity (Subgraph Isomorphism Census)** We define the molecular graph $\mathcal{G} = (\mathcal{V}, \mathcal{E})$ and a library of functional group subgraphs $\mathcal{F} = \{\mathcal{S}_1, \ldots, \mathcal{S}_k\}$. The moietal identity feature vector, $\mathbf{x}_{\text{moi}} \in \mathbb{N}^k$ is constructed as a subgraph isomorphism counting function $\psi$:

$$\mathbf{x}_{\text{moi}}[i] = \psi(\mathcal{S}_i, \mathcal{G}) = |\{g \subseteq \mathcal{G} : g \cong \mathcal{S}_i\}|, \quad (3)$$

where $\cong$ denotes graph isomorphism. That is, $\mathbf{x}_{\text{moi}}$ is simply a vector of substructure counts: each entry records how many times a given functional group (e.g. a hydroxyl, carbonyl, or benzene ring) occurs in the molecule. This quantifies distinct functional moieties which reflect the primary active sites for specific solute-solvent interactions.

2. **Topological Structure (Homomorphism Profiles)** To capture higher-order steric and connectivity patterns, we employ Motif Structural Encoding (MoSE) (Bao et al., 2025). Let $\mathcal{H} = \{H_1, \ldots, H_d\}$ be a set of pattern graphs representing cycles ($C_3$–$C_8$), paths ($P_3$–$P_5$), and stars ($S_3$–$S_4$). For a molecular graph $G$, we define the topological feature vector $\mathbf{x}_{\text{topo}} \in \mathbb{R}^d$ by the counts of

adjacency-preserving mappings $\phi : \mathcal{V}(H_j) \rightarrow \mathcal{V}(G)$. We augment this homomorphism profile with global topological indices (BalabanJ, BertzCT, Kappa statistics) (Balaban, 1982; Bertz, 1981) to improve injectivity (and reduce degeneracy) with respect to branching patterns.

3. **Thermodynamic Reconstruction (Energetics).** To capture thermodynamic signatures related to molecular cohesion and disorder, we employ the Joback group-contribution method (Joback & Reid, 1987), which provides an estimate of the melting point $T_{m,\text{pred}}$:

$$T_{m,\text{pred}}(G) = 122.5 + \sum_{g \in \mathcal{G}} N_g \cdot \Delta T_g \qquad (4)$$

Complementing these energetic terms, we reconstruct the five **Abraham solvation parameters** (Abraham, 1993) $\alpha = [A, B, E, S, V]^{\top}$ *via* physicochemical proxies. $(A)$ combines H-bond donor count over acidic groups, basicity $(B)$ sums H-bond acceptors over basic functional groups, polarizability $(S)$ is estimated from heteroatom counts and aromatic ring content, excess molar refraction $(E)$ is derived from molecular refractivity (MolMR/10), and McGowan characteristic volume $(V)$ is calculated from atomic volume increments. We also require descriptors that encode both electronic structure and surface topology. We concatenate electronic descriptors $\mathbf{e}$ (EState indices, partial charges), surface area distributions $\mathbf{s}$ (PEOE_VSA, TPSA), and a drug-likeness index $\mathbf{d}$ (QED) to form the final state vector $\mathbf{x}_{\text{phys}} = [\mathbf{e} \oplus \mathbf{s} \oplus \mathbf{d} \oplus \text{MolLogP}]$. These features encode electrostatic and dispersion (van der Waals surface) characteristics that are critical for solvation thermodynamics.

A complete specification of the feature hierarchy and individual descriptors is provided in Table 12 in Appendix G. The final feature vector has dimensionality $\mathbf{x} \in \mathbb{R}^{176}$.

### 3.1. Regime I: Single-Solvent Prediction

For fixed-solvent tasks (*e.g.*, aqueous solubility), the solvent environment is constant, and solubility prediction reduces to a solute-centric regression problem. Each solute molecule is represented by the fixed descriptor vector generated by the featurizer, and solubility is predicted directly from this representation. We model this relationship using a gradient-boosted decision tree ensemble trained to minimise RMSE.

### 3.2. Regime II: Multi-Solvent Interaction Learning

When both the solute and solvent vary, solubility prediction requires modelling solvent-dependent interactions in addition to intrinsic molecular properties (Boobier et al., 2020; yeyunchen, 2025). From a thermodynamic perspective, this regime is governed by the enthalpy of mixing, $\Delta H_{\text{mix}}$, which emerges from pairwise compatibilities between the solute and solvent functional groups as well as their associated configurational degrees of freedom. (Mi-

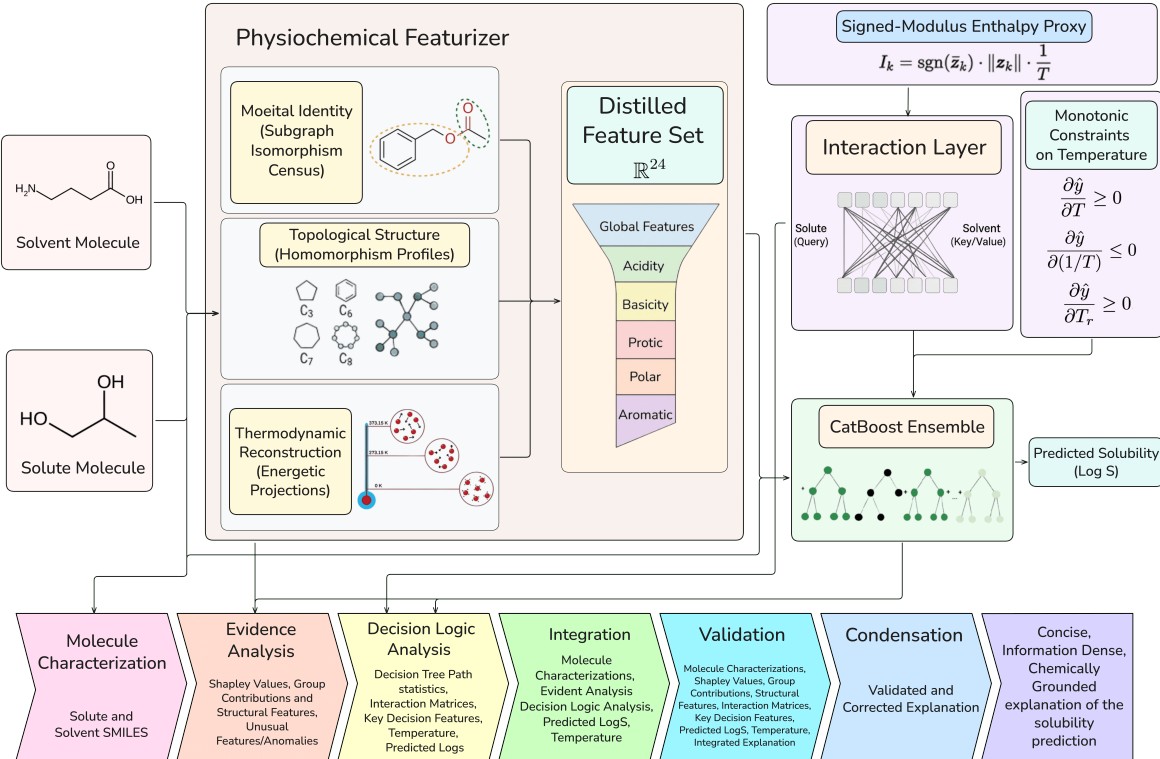

*Figure 1.* The figure depicts the pipeline for Regime 2. We featurize both molecules, calculate interaction features from the distilled feature set, and pass them along with the physically-constrained thermodynamic terms to the regressor. The approach for Regime 1 is much simpler, with the solute features passed directly to the ensemble.

anowski & Łabojko, 2023)

To address this, we use an *Interaction Layer*, a mechanism to model solute-solvent interactions. The model treats the solute ($\mathcal{A}$) and solvent ($\mathcal{B}$) as distinct chemical entities and learns interaction-weighted representations that reflect the physical principle of 'like dissolves like' (Zhuang et al., 2021).

**Refined Feature Set.** The direct application of rotations to the full descriptor space is computationally expensive ($\mathcal{O}(D^2)$). We thus define the *Refined Feature Set*, a physicochemical representation that compacts the descriptor set to a few interaction-relevant features.

**Definition 3.1** (Refined Feature Set). Let $\phi_{\text{raw}} : \mathcal{M} \to \mathbb{R}^D$ denote the molecular featurization. We define a deterministic feature-selection transformation $\Psi : \mathbb{R}^D \to \mathbb{R}^{24}$ (a subset of $\phi_{\text{raw}}$, not a learned projection) such that the resulting vector $\mathcal{C}$ is formed by concatenating physically interpretable descriptors: $\mathcal{C} = \Psi(\phi_{raw}(\mathcal{M})) = [\mathbf{p}_{\text{global}} \parallel \mathbf{f}_{\text{acid}} \parallel \mathbf{f}_{\text{base}} \parallel \cdots \parallel \mathbf{f}_{\text{arom}}]$, where $\mathbf{p}_{\text{global}}$ encodes bulk physicochemical and thermodynamic properties, and $\mathbf{f}_{i=\text{acid/base/.../arom}}$ represents aggregated functional groups such as acidic, basic, and aromatic moieties, providing a coarse representation of the molecule. The complete specification is shown in Table 1.

**Interaction Layer.** Each of the 24 entries in $\Psi(\mathbf{x})$ is treated as a token corresponding to a single macrophysical property (e.g. MolLogP, acidity, basicity, ring count; Table 1); the cross-attention block thus computes a $24 \times 24$ matrix whose entries directly quantify feature-level solute–solvent coupling. Let the refined representations of the solute and solvent be $\Psi(\mathbf{x}_{\mathcal{A}})$ and $\Psi(\mathbf{x}_{\mathcal{B}})$, respectively. We define linear projection operators to map these representations into a $d$-dimensional relational space ($d = 32$) to get $\mathbf{E}_{\mathcal{A}}$ and $\mathbf{E}_{\mathcal{B}}$. Following standard scaled dot-product attention (Vaswani et al., 2017), DISSOLVR computes the Query ($\mathbf{Q}_{\mathcal{A}}$), Key ($\mathbf{K}_{\mathcal{B}}$), and Value ($\mathbf{V}_{\mathcal{B}}$) matrices through linearly learned transformations of the input vectors: $\mathbf{Q}_{\mathcal{A}} = \mathbf{E}_{\mathcal{A}}\mathbf{W}^Q, \mathbf{K}_{\mathcal{B}} = \mathbf{E}_{\mathcal{B}}\mathbf{W}^K, \mathbf{V}_{\mathcal{B}} = \mathbf{E}_{\mathcal{B}}\mathbf{W}^V$, where $\mathbf{W}^Q, \mathbf{W}^K, \mathbf{W}^V \in \mathbb{R}^{d \times d_k}$ are learnable projection matrices. The parameters ($d = 32, d_k = 8$) in this interaction layer are small and explicitly displayed, and downstream transparency is achieved by exposing the interaction matrix as a sparse heatmap over distilled tokens.

This mechanism can be viewed as solving for a representation that maximises the "solvation affinity" between the solute's features and the solvent's properties. Formally, for a given solute $A$ and solvent $B$, the enriched representation $\mathbf{Z}_{\mathcal{A} \to \mathcal{B}}$ can be framed as:

$$\mathbf{Z}_{\mathcal{A} \to \mathcal{B}} = \sigma\left(\frac{\mathbf{Q}_{\mathcal{A}}\mathbf{K}_{\mathcal{B}}^{\top}}{\sqrt{d_k}}\right)\mathbf{V}_{\mathcal{B}} \qquad (5)$$

where $\sigma$ is the softmax normalization. This yields an interaction matrix that determines the influence of each solvent moiety on the solute's final representation. This provides an approach to dissect the specific strengths of solute-solvent interactions through a visual heatmap (Appendix G).

**Final Prediction.** To produce the final solubility estimate, we fuse the interaction-aware representation with thermodynamic state variables within a physics-constrained framework under an assumption of endothermic dissolution:

**Signed-Interaction Modulus.** Let $\mathbf{z}_k \in \mathbb{R}^d$ denote the $k$-th row of the matrix $\mathbf{Z}_{\mathcal{A}\rightarrow\mathcal{B}}$, corresponding to the weighted representation of the $k$-th refined feature token. To map these representations to physically interpretable energetic contributions, we define a signed-modulus interaction intensity for each token $k$.

$$I_k = \underbrace{\text{sgn}(\bar{z}_k)}_{\text{Direction}} \cdot \underbrace{||\mathbf{z}_k||_2}_{\text{Magnitude}} \cdot \underbrace{\frac{1}{T}}_{\text{Arrhenius Term}} \quad (6)$$

where $\bar{z}_k$ is the scalar mean of the elements in $\mathbf{z}_k$. The sign captures the directionality of the interaction (stabilising *vs* destabilising), while the $L_2$ norm represents its total magnitude. The inverse-temperature scaling, where $T$ is the system temperature defined in Definition 2.2, adopts a phenomenological form inspired by the van't Hoff relation, $\ln S \propto -\Delta H/RT$, which allows $I_k$ to serve as a first-order heuristic for the enthalpy of mixing $\Delta H_{\text{mix}}$ by assuming a linear-reciprocal dependence on temperature.

**Thermodynamic State Augmentation.** We explicitly incorporate thermodynamic state variables to anchor the predictions to physically meaningful regimes. These include the absolute temperature $T$, inverse temperature $(1/T)$, the predicted solute melting point $T_m$ as a proxy for lattice energy, and the reduced temperature $T_r = T/T_{m,\text{pred}}$, derived from corresponding states theory (Leland & Chappelear, 1968). We have now all features for the model. Throughout the paper, the per-molecule descriptor count is 176, and the final Regime-II CatBoost input after solute, solvent, interaction, and thermodynamic terms are included, has 380 features.

**Physics-aware monotonic constraints.** A primary reason we adopt a gradient-boosted decision tree backbone is that it natively supports *hard* per-feature monotonicity at split time,

*Table 1.* Refined Feature Set ($\mathcal{C} \in \mathbb{R}^{24}$)

| Category | Constituent Tokens (Descriptors) |
|---|---|
| **Global Physics** | **Physicochemical:** MolWt, LogP, TPSA, LabuteASA, HallKierAlpha, HeavyAtomCount, AromaticRings, Max/Min PartialCharge, NumRotatableBonds. **H-Bonding:** NumHDonors, NumHAcceptors. **Thermodynamic:** Pred_$T_m$, Abraham A, B, S, E, V. |
| **Functional Super-Groups** | **Acidic:** Carboxyls, Phosphates ($\Sigma$ `fr_COO`...) **Basic:** Amines, Pyridines, Amidines, Lactams **Protic:** Alcohols, Phenols, Thiols **Polar:** Ketones, Aldehydes, Esters, Nitriles, Sulfones **Halogen:** F, Cl, Br, I, Alkyl Halides **Aromatic:** Benzene, Heterocycles (Furan, Thiophene) |

providing a structural guarantee that soft Sobolev penalties (used, e.g., by FastSolv (Attia et al., 2025)) only approximate at substantially higher computational cost (App. H.3). Solubility in general is expected to increase monotonically with absolute temperature $T$ and reduced temperature $T_r$, while exhibiting an inverse monotonic dependence on $1/T$ under a standard assumption of endothermic dissolution. Because tree ensembles are piecewise constant, the following constraints are enforced as coordinate-wise finite monotonicity rather than as differentiability assumptions:

$$\frac{\partial\hat{\Sigma}}{\partial T} \geq 0, \quad \frac{\partial\hat{\Sigma}}{\partial(1/T)} \leq 0, \quad \frac{\partial\hat{\Sigma}}{\partial T_r} \geq 0. \quad (7)$$

**Lemma 3.1** (Closure of Monotone Ensembles). *Let $\mathcal{H}_{\mathcal{M}}$ denote the class of prediction functions that are coordinate-wise nondecreasing in $T$ and $T_r$ and coordinate-wise nonincreasing in $1/T$, holding all other descriptors fixed. If $f, g \in \mathcal{H}_{\mathcal{M}}$ and $\alpha, \beta \geq 0$, then $\alpha f + \beta g \in \mathcal{H}_{\mathcal{M}}$. Consequently, any boosted additive ensemble with nonnegative learning rates remains monotone whenever each base learner is monotone.*

*Proof: See Proof F.1 in Appendix F.*

**Lemma 3.2** (Efficiency of Monotonic Optimization). *Consider training base learners $f_t \in \mathcal{H}_{\mathcal{M}}$ with oblivious trees of depth $L$, where $d_t$ features are subject to monotonic constraints on $N$ samples with $K$ features. Then:*

1. ***Complexity**: Training incurs $\mathcal{O}(NKL)$ cost per tree for split search and $\mathcal{O}(d_t \cdot 2^L)$ for leaf adjustment. For realistic configurations ($L \leq 10$, $d_t \ll K$, $N \geq 10^4$), the adjustment cost is negligible: $\mathcal{O}(d_t \cdot 2^L) \ll \mathcal{O}(NKL)$.*
2. ***Closure under boosting**: The ensemble $F(x) = \sum_{t=1}^{T} \eta f_t(x)$ remains in $\mathcal{H}_{\mathcal{M}}$ for learning rates $\eta \geq 0$, without post-hoc projection.*

*Proof: See Proof F.3 in Appendix F.*

**Theorem 3.1** (Thermodynamic Consistency with Capacity Restriction). *The DISSOLVR framework, when trained with monotonic constraints on temperature features $\{T, 1/T, T_r\}$, satisfies:*

1. ***Physical Consistency**: Under the standard endothermic-dissolution regime, the predicted solubility obeys the coordinate-wise thermodynamic constraints in Equation 7.*
2. ***Capacity Restriction**: The monotone ensemble class is a subset of the corresponding unconstrained ensemble class. Therefore, on any fixed sample, the monotone class cannot have larger empirical Rademacher complexity or greater ability to fit arbitrary noise than the unconstrained class.*
3. ***Computational Feasibility**: Training complexity remains $\mathcal{O}(TNKL)$ for $T$ trees.*

*Proof: See Proof F.4 in Appendix F.*

The final solubility prediction produced by DISSOLVR:

$$\hat{\Sigma} = \text{DISSOLVR}(\mathbf{x}_{\mathcal{A}}, \mathbf{x}_{\mathcal{B}}, \mathbf{I}, f(T)), \qquad (8)$$

where the interaction vector $\mathbf{I} = \{I_1, \ldots, I_{24}\}$ encodes solvent-conditioned physicochemical couplings derived from the interaction layer, and $\mathbf{x}_{\mathcal{A}}, \mathbf{x}_{\mathcal{B}}$ are the full feature sets for the solute and solvent respectively. $f(T)$ are the four temperature terms mentioned: $T_m, T, T_r, \frac{1}{T}$.

## 4. LLM-Augmented Post-Hoc Explainer

Although DISSOLVR is fully transparent at the model level, a fundamental gap remains between statistical transparency and *scientific utility*. For example, DISSOLVR exposes explicit decision tree statistics (*e.g.*, "Shap contribution of TPSA = 0.78 and rotation Between Solute_LogP and Solvent_T_m = 0.98"), but such statistics are chemically abstract. Interpreting such rules requires manual translation by connecting features to the known solubility trend, thereby increasing the cognitive load on domain scientists without adding new insight. To bridge this gap, we use LLMs.

Rather than eliciting a single, end-to-end explanation, the LLM is guided through a six-stage *progressive prompting framework* (bottom ribbon in Fig. 1) in which information is first gathered, then interpreted, and subsequently synthesized into the final narrative. The first three stages are dedicated to structured evidence acquisition, while the final three stages focus on synthesis, validation, and communication. Critically, the LLM is *blinded to prediction quality* until Stage 3. This design choice prevents confirmation bias during early interpretation and ensures that explanations are grounded in model evidence rather than retrospectively rationalizing errors. The individual stages are as follows:

- **Stage 1: Molecular Characterisation:** The LLM, provided only with SMILES representations, characterizes molecular structures through solubility-relevant properties (*e.g.*, polarity, hydrogen bonding, and aromaticity).
- **Stage 2: Evidence Analysis:** The LLM identifies dominant drivers and interprets solute solvent compatibility by processing raw DISSOLVR descriptors, feature level Shapley values (including temperature), and aggregated group level contributions. This stage translates statistical evidence specifically marginal feature contributions into chemically grounded narratives.
- **Stage 3: Decision Logic:** The LLM is exposed to the model's explicit decision logic in the form of DISSOLVR tree paths, together with interaction matrices from the interaction transformer, the experimental temperature, and the predicted solubility value ($\log S$). The LLM maps these symbolic rules and feature interactions to interpretations consistent with chemical intuition.
- **Stage 4: Integration:** The LLM synthesizes all preceding outputs into a structured, chemically grounded explanation comprising four components: **(1) Prediction**

**& Drivers:** Reports the predicted $\log S$ and identifies the 2–3 most influential features. **(2) Compatibility:** Details solute–solvent matching based on interaction rotation patterns. **(3) Mechanism:** Categorises the dominant dissolution regime (e.g., crystal- vs. solvent-limited). (4) **Confidence:** Flags discrepancies between model logic and established literature.

- **Stage 5: Validation:** The LLM validates and cross-references generated explanations against previously used model data (*e.g.*, SHAP values, decision paths) to ensure they remain faithful to our model. Any flaw found is identified and corrected. This attempts to detect and correct logical "drifts" found within the explanation layer.
- **Stage 6: Condensation:** The LLM condenses the detailed solubility explanation produced above into a concise, information-dense summary.

The full prompt architecture and stage-wise outputs are detailed in Appendix J.

## 5. Experiments

We evaluate DISSOLVR across two distinct physical regimes: **Aqueous Solubility** (Regime I) and **Multi-Solvent Solubility** (Regime II). All baselines are trained and evaluated using identical splits to ensure fair comparisons.

### 5.1. Datasets

**Aqueous Solubility Datasets:** We evaluate fixed-solvent performance using ESOL ($N = 1128$) and AqSolDB ($N = 8311$). ESOL benchmarks low-data behaviour, while AqSolDB evaluates large-scale generalisation under experimental noise. See Appendix B for curation procedures.

**Multi-Solvent Datasets and External Validation:** We introduce from Ramani & Karmakar (2024) a benchmark for out-of-distribution (OOD) generalisation using a solvent-holdout split, in which the models are trained on frequent solvents and evaluated on rare solvent environments. We evaluate this using BigSolDB 1.0 and 2.0, which contain solubility measurements of more than 200 organic solvents. We report blind external validation results on the **Second Solubility Challenge dataset** (Llinas et al., 2020) and the **Leeds solubility dataset** (Boobier et al., 2020). These datasets are independent of the training set and test robustness to distribution shift and external variability.

### 5.2. Benchmarking Protocol

**Data Splits and Metrics.** We employ regime-specific splitting strategies designed to rigorously test both interpolation and extrapolation. For aqueous regimes, we perform a random train-test split; for multisolvent regimes, we train on the Top-$K$ solvents and test on the remaining solvents, to access generalisation to data-poor environments. We also test on the Leeds dataset, which contains entirely novel solutes from the training corpus. We report both RMSE and the $R^2$,

*Table 2.* **Performance Benchmarking.** Regime I focuses on aqueous datasets. Regime II evaluates generalization across multi-solvent environments. Reported mean $\pm$ std over 5 seeds. Dark green represents the best model (s), and light green represents the second-best model(s).

*(a)* Regime I Results (Aqueous Only)

| Method | ESOL | | AqSolDB | | Second Solubility Challenge | |
|---|---|---|---|---|---|---|
| | RMSE | $R^2$ | RMSE | $R^2$ | RMSE | $R^2$ |
| GSE(Yalkowsky & Valvani, 1980) | 2.2623 | -0.1669 | 2.5816 | -0.3779 | 2.7283 | -3.6450 |
| ESOL Model(Delaney, 2004) | 0.9802 | 0.7809 | 1.3161 | 0.6419 | 1.0445 | 0.3192 |
| CHEMPROP (Heid et al., 2024) | $0.5749_{\pm 0.0202}$ | $0.9245_{\pm 0.0054}$ | $0.8833_{\pm 0.0234}$ | $0.8386_{\pm 0.0086}$ | $1.0192_{\pm 0.0316}$ | $0.3511_{\pm 0.0400}$ |
| FASTPROP (Burns & Green, 2025) | $0.5485_{\pm 0.0157}$ | $0.9314_{\pm 0.0039}$ | $1.0514_{\pm 0.0711}$ | $0.7704_{\pm 0.0314}$ | $1.0329_{\pm 0.0700}$ | $0.3312_{\pm 0.0919}$ |
| AQSOLPRED (Sorkun et al., 2021) | $0.5033_{\pm 0.0035}$ | $0.9422_{\pm 0.0008}$ | $0.8270_{\pm 0.0060}$ | $0.8437_{\pm 0.0023}$ | $0.9490_{\pm 0.0269}$ | $0.4351_{\pm 0.0322}$ |
| SOLUBNET (Chen et al., 2023) | $0.7749_{\pm 0.0253}$ | $0.8629_{\pm 0.0090}$ | $1.1802_{\pm 0.0044}$ | $0.7120_{\pm 0.0022}$ | $0.9412_{\pm 0.0100}$ | $0.4471_{\pm 0.0117}$ |
| Ulrich et al. (2025) | $0.9823_{\pm 0.0996}$ | $0.7777_{\pm 0.0439}$ | $1.1933_{\pm 0.0516}$ | $0.7051_{\pm 0.0252}$ | $1.2129_{\pm 0.1050}$ | $0.0751_{\pm 0.1612}$ |
| Tayyebi et al. (2023) | $0.6926_{\pm 0.0053}$ | $0.8906_{\pm 0.0017}$ | $0.9064_{\pm 0.0026}$ | $0.8301_{\pm 0.0010}$ | $0.9102_{\pm 0.0063}$ | $0.4829_{\pm 0.0071}$ |
| Random Forest (Breiman, 2001) | $0.5429_{\pm 0.0006}$ | $0.9328_{\pm 0.0001}$ | $0.8902_{\pm 0.0002}$ | $0.8362_{\pm 0.0001}$ | $0.8784_{\pm 0.0006}$ | $0.5184_{\pm 0.0007}$ |
| XGBoost (Chen & Guestrin, 2016) | $0.5005_{\pm 0.0056}$ | $0.9429_{\pm 0.0013}$ | $0.8311_{\pm 0.0029}$ | $0.8572_{\pm 0.0010}$ | $0.9014_{\pm 0.0171}$ | $0.4928_{\pm 0.0193}$ |
| LightGBM (Ke et al., 2017) | $0.4773_{\pm 0.0104}$ | $0.9480_{\pm 0.0022}$ | $0.8457_{\pm 0.0037}$ | $0.8522_{\pm 0.0013}$ | $0.9338_{\pm 0.0118}$ | $0.4558_{\pm 0.0137}$ |
| Artificial Neural Network | $0.8221_{\pm 0.0818}$ | $0.8444_{\pm 0.0304}$ | $1.1322_{\pm 0.0639}$ | $0.7341_{0.0293}$ | $1.0403_{\pm 0.0550}$ | $0.3227_{\pm 0.0733}$ |
| **DISSOLVR** | $0.4631_{\pm 0.0021}$ | $0.9511_{\pm 0.0007}$ | $0.8158_{\pm 0.0030}$ | $0.8624_{\pm 0.0010}$ | $0.9012_{\pm 0.0033}$ | $0.4932_{\pm 0.0037}$ |

*(b)* Regime II Results (Multi-Solvent)

| Method | BigSolDB 1.0 (Top-19 solvents) | | BigSolDB 2.0 (Top-21 solvents) | | Leeds Dataset (Solute Holdout) | |
|---|---|---|---|---|---|---|
| | RMSE | $R^2$ | RMSE | $R^2$ | RMSE | $R^2$ |
| FASTSOLV (Attia et al., 2025) | $0.7799_{\pm 0.0243}$ | $0.5449_{\pm 0.0283}$ | $0.6938_{\pm 0.0041}$ | $0.5844_{\pm 0.0049}$ | $0.8868_{\pm 0.0065}$ | $0.3427_{\pm 0.0096}$ |
| RILOOD (yeyunchen, 2025) | $0.8814_{\pm 0.0509}$ | $0.4201_{\pm 0.0675}$ | $0.8791_{\pm 0.0458}$ | $0.3346_{\pm 0.0710}$ | $1.1566_{\pm 0.0163}$ | $-0.1191_{\pm 0.0316}$ |
| Random Forest (Breiman, 2001) | $0.7276_{\pm 0.0036}$ | $0.6060_{\pm 0.0039}$ | $0.7257_{\pm 0.0030}$ | $0.5477_{\pm 0.0038}$ | $1.0479_{\pm 0.0011}$ | $0.0816_{\pm 0.0019}$ |
| XGBoost (Chen & Guestrin, 2016) | $0.7289_{\pm 0.0051}$ | $0.6047_{\pm 0.0055}$ | $0.6416_{\pm 0.0019}$ | $0.6465_{\pm 0.0021}$ | $0.9442_{\pm 0.0110}$ | $0.2542_{\pm 0.0176}$ |
| LightGBM (Ke et al., 2017) | $0.7109_{\pm 0.0065}$ | $0.6239_{\pm 0.0069}$ | $0.6505_{\pm 0.0036}$ | $0.6366_{\pm 0.0040}$ | $0.9071_{\pm 0.0051}$ | $0.3117_{\pm 0.0077}$ |
| Artificial Neural Network | $0.8447_{\pm 0.0129}$ | $0.4689_{\pm 0.0163}$ | $1.0646_{\pm 0.0295}$ | $0.0259_{0.0539}$ | $6.3069_{\pm 1.7099}$ | $-34.7137_{\pm 18.025}$ |
| GAT (Veličković et al., 2018) | $0.7285_{\pm 0.0387}$ | $0.6040_{\pm 0.0415}$ | $0.6714_{\pm 0.0087}$ | $0.6128_{\pm 0.0100}$ | $1.0851_{\pm 0.0368}$ | $0.0141_{\pm 0.0662}$ |
| GIN (Xu et al., 2019) | $0.7711_{\pm 0.0279}$ | $0.5570_{\pm 0.0323}$ | $0.7398_{\pm 0.0233}$ | $0.5295_{\pm 0.0296}$ | $1.0168_{\pm 0.0346}$ | $0.1344_{\pm 0.0592}$ |
| GCN (Kipf & Welling, 2017) | $0.8180_{\pm 0.0157}$ | $0.5019_{\pm 0.0190}$ | $0.7463_{\pm 0.0258}$ | $0.5211_{\pm 0.0327}$ | $1.2049_{\pm 0.0806}$ | $-0.2196_{\pm 0.1633}$ |
| **DISSOLVR** | $0.6783_{\pm 0.0023}$ | $0.6576_{\pm 0.0023}$ | $0.6305_{\pm 0.0013}$ | $0.6587_{\pm 0.0014}$ | $0.8798_{\pm 0.0012}$ | $0.3527_{\pm 0.0018}$ |

with results averaged over five seeds and reported as mean $\pm$ standard deviation. For details of splits, see Appendix B.

**Baselines.** First, to establish architectural baselines across both regimes, we evaluate standard tree-based learners (Decision Trees, Random Forests, and LightGBM), fundamental GNNs (MPNN, GCN, GAT, and GIN), and a simple ANN. Second, for aqueous solubility (Regime I), we compare against established physicochemical baselines, including the General Solubility Equation and the ESOL model, as well as modern deep learning-based architectures: Chemprop, Fastprop, SolTranNet (Francoeur & Koes, 2021), and SolubNet. We also include recent descriptor-based models by Tayyebi et al. (2023) and tautomer-enhanced augmentation learning by Ulrich et al. (2025). Third, for multi-solvent prediction (Regime II), we evaluate against FastSolv , Chemprop, and OOD generalisation frameworks such as RILOOD. Complete results for all baselines are provided in Appendix C. Tree-based models and ANNs were provided the same features as DISSOLVR, and graph-based models utilised the GraphConvMol Featurizer.

## 5.3. Regime I: Aqueous Solubility

DISSOLVR outperforms all baselines on AqSolDB and maintains strong generalisation on the Second Solubility Challenge, indicating robustness to distribution shift (Table 2a). A wall-clock breakdown against FastSolv is reported in App. E.2, and the eight most influential SHAP features on ESOL are listed in App. H.1.

## 5.4. Regime II: Multi-Solvent Solubility

**Generalisation Across Solvents.** Training on frequent solvents and evaluating on rare or previously unseen solvents constitutes a strict OOD-learning task. DISSOLVR consistently outperforms baselines across all datasets, demonstrating strong generalisation against variability (Table 2b).

While latent-based models (*e.g.*, yeyunchen (2025)) attempt to learn environment-independent representations, we show that DISSOLVR's structural invariance arises from its structural descriptor set derived from fundamental chemical laws, remaining invariant under distribution shifts. The results in Table 2b verify that latent disentanglement in molecular relational learning often captures spurious noise rather than invariant chemical signals. DISSOLVR's superior performance indicates that explicit featurization is a more reliable substrate for OOD generalization.

**Physical Consistency with the Apelblat Equation.** We assessed the model's ability to reproduce temperature-

dependent trends modeled by the empirical Apelblat equation, $\ln S = A + B/T + C \ln T$, on BigSolDB 1.0 (representative parity plots in App. G.4). Predicted curve shapes demonstrated high physical fidelity, with Pearson correlations of 0.9886 (train) / 0.9844 (test) and Cosine similarities of 0.9398 (train) / 0.8201 (test). These results indicate that the model effectively captures underlying thermodynamic behaviors, even when absolute magnitudes vary.

**Hard monotonic constraints.** Beyond the small mean-RMSE improvement, a 5-seed ablation shows the constraint cuts seed-to-seed prediction variance by 44–72% and contributes $\Delta$RMSE $= +0.0205$ on the 18.3% of samples whose solutes have conflicting multi-source training data (vs. $+0.0018$ on clean data), acting as a structural regulariser where unconstrained trees catastrophically fail to extrapolate. Full ablation, stratified analysis, and comparison to soft Sobolev penalties are in App. H.3.

## 5.5. Ablation Studies

**Ablation Analysis: Task-Specific Chemical Logic.** To validate the necessity of each component in our descriptor hierarchy, we conducted an ablation study across two distinct challenges: aqueous interpolation (AqSolDB, Table 4) and solute extrapolation (Leeds, Table 3). In the aqueous regime (Regime I), performance is dominated by bulk Physicochemical descriptors (*e.g.*, MolLogP and TPSA). All feature classes contribute significantly.

The solute-extrapolation task in Regime II (Leeds; Table 3) shows that the model can no longer rely on bulk properties of known scaffolds and instead must prioritize structural identity and solute–solvent interactions. Notably, removing Interaction Terms (**I**) causes a degradation nearly five times the baseline experimental variance, indicating their importance for out-of-distribution generalization. Increased reliance on Compositional and Topological features further suggests that the model leverages local structural motifs to handle molecular novelty. Together, these results motivate the DISSOLVR architecture: although different descriptor layers dominate across tasks, the complete descriptor space is required for consistent performance across chemical regimes. Replacing the cross-attention Interaction Block with direct concatenation of the full solute–solvent descriptors triples the feature dimension, increases training cost by $\sim 9\times$, and degrades OOD RMSE (App. H.5). For complexity analysis, see Appendix E. A comprehensive SHAP mechanistic analysis of the global thermodynamic drivers and interaction fine-tuning is provided in Appendix H.1.

## 5.6. Human Expert Evaluation of Explanation Quality

To validate the utility of our generated explanations, we conducted a blind survey with 7 Professors, 12 PhDs and 3 Masters-level chemists across 15 diverse solubility predic-

*Table 3.* **Feature Ablation on Leeds Dataset (Regime II).** Impact of removing or isolating descriptor groups on a solute extrapolation task. $\sigma_{\text{RMSE}} = 0.0024$ from 5-seed baseline.

| Experiment | $\mathbf{R}^2$ | RMSE | $\Delta$RMSE |
|---|---|---|---|
| **Full Model (380 features)** | **0.3576** | **0.8764** | – |
| REMOVE: $\mathbf{x}_\mathcal{A}, \mathbf{x}_\mathcal{B}$ Entirely | -0.0121 | 1.1001 | +0.2237 |
| REMOVE: Category COMPOSITIONAL | 0.3301 | 0.8950 | +0.0186 |
| REMOVE: Category TOPOLOGICAL | 0.3329 | 0.8931 | +0.0167 |
| REMOVE: Category ENERGETIC | 0.3482 | 0.8828 | +0.0064 |
| REMOVE: Category PHYSICOCHEMICAL | 0.3563 | 0.8773 | +0.0009 |
| REMOVE: **I** (Interaction Terms) | 0.3411 | 0.8876 | +0.0112 |
| REMOVE: $f(\mathbf{T})$ (Engineered State) | 0.3467 | 0.8838 | +0.0074 |
| ONLY: Category COMPOSITIONAL | 0.3456 | 0.8846 | +0.0082 |
| ONLY: Category TOPOLOGICAL | 0.1347 | 1.0172 | +0.1408 |
| ONLY: Category ENERGETIC | 0.2500 | 0.9470 | +0.0706 |
| ONLY: Category PHYSICOCHEMICAL | 0.2666 | 0.9364 | +0.0600 |
| ONLY: **I** (Interaction Terms) | -0.0990 | 1.1463 | +0.2699 |
| ONLY: $f(\mathbf{T})$ (Engineered State) | -0.0974 | 1.1455 | +0.2691 |

*Table 4.* **Feature Ablation on AqSolDB (Regime I).** Single-solvent aqueous ablation. $\sigma_{\text{RMSE}} = 0.0030$ from 5-seed baseline.

| Experiment | $\mathbf{R}^2$ | RMSE | $\Delta$RMSE |
|---|---|---|---|
| **Full Model (176 features)** | **0.8640** | **0.8110** | – |
| REMOVE: COMPOSITIONAL | 0.8614 | 0.8187 | +0.0076 |
| REMOVE: TOPOLOGICAL | 0.8606 | 0.8211 | +0.0101 |
| REMOVE: ENERGETIC | 0.8623 | 0.8163 | +0.0052 |
| REMOVE: PHYSICOCHEMICAL | 0.8344 | 0.8950 | +0.0839 |
| ONLY: COMPOSITIONAL | 0.8283 | 0.9114 | +0.1004 |
| ONLY: TOPOLOGICAL | 0.7359 | 1.1303 | +0.3193 |
| ONLY: ENERGETIC | 0.7914 | 1.0044 | +0.1934 |
| ONLY: PHYSICOCHEMICAL | 0.8577 | 0.8296 | +0.0186 |

tion tasks (330 total observations). The study assessed three critical dimensions: task difficulty, explanation quality, and trust enhancement. Results are summarized in Figure 2 and Table 5. A detailed case study examining a representative prediction is provided in Appendix I.

**Task Difficulty.** In a zero-shot setting, human experts achieved an accuracy of only 17.9% in classifying solubility into the correct log-bin. This performance is statistically indistinguishable from a random baseline of 20.0% (one-sample $t$-test: $t = -0.61$, $p = 0.55$), underscoring the cognitive intractability of solubility prediction and the necessity of computational aids for this domain.

**Explanation Quality.** Despite the complexity of the underlying model, experts rated the generated chemical narratives highly, with a mean quality score of $3.75 \pm 1.10$ (SD) on a 5-point scale. This is significantly above the neutral threshold ($t = 12.41$, $p < 0.001$), with 60.9% of ratings at 4-5, confirms the system produces chemically coherent rationales. A *trained-vs.-untrained* ablation – identical prompts run on a trained DISSOLVR and a randomly-initialised copy, scored by an LLM-as-judge (Zheng et al., 2023) – yields mean agreement $1.83 \pm 0.42$ on a scale of 1–5 ($n = 30$ samples), indicating that the generated narratives are sensitive to model evidence rather than generic chemical priors (App. H.2). An

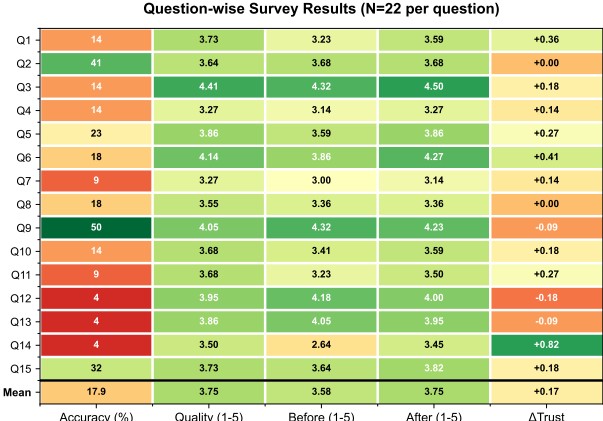

*Figure 2.* Question-wise survey results across 15 solubility prediction tasks (N=22 respondents per question). Colors indicate relative performance: green (high/positive), yellow (moderate), red (low/negative). The Mean row summarizes overall performance.

*Table 5.* Summary of Expert Survey Results

| Metric | Value | Statistical Test |
|---|---|---|
| Respondents/Observations | 22 / 330 | – |
| Human Accuracy | 17.9% | $t = -0.61, p = 0.55$ |
| Random Baseline | 20.0% | (one-sample $t$-test) |
| Explanation Quality (1-5) | $3.75 \pm 1.10$ | $t = 12.41, p < 0.001$ |
| Trust Before Explanation | $3.58 \pm 1.05$ | – |
| Trust After Explanation | $3.75 \pm 1.02$ | – |
| Trust Change ($\Delta$) | $+0.17 \pm 0.72$ | $t = 4.28, p < 0.001$ |
| Effect Size (Cohen's $d$) | 0.235 | (paired $t$-test) |

end-to-end worked example tracing a single prediction from raw features to the final LLM narrative is given in App. K.7.

**Trust Enhancement.** Reading explanations resulted in a statistically significant net increase in expert agreement with the model's prediction ($\Delta = +0.17$, paired $t$-test: $t = 4.28$, $p < 0.001$, Cohen's $d = 0.235$). Trust increased in 23.6% of cases while decreasing in only 10.3%, demonstrating that the explanations are persuasive, bridging the gap between "black box" predictions and expert acceptance.

# 6. Conclusion

This work introduces DISSOLVR, a transparent framework for solubility prediction that also establishes a reproducible benchmark enabling fair comparisons across datasets and OOD regimes. By explicitly encoding moietal identity, topology, and phase-change energetics, the model achieves competitive performance approaching the empirical aleatoric limit ($\sim$0.6–0.8 $\log_{10}(S)$), suggesting that current performance is increasingly constrained by experimental noise rather than model depth. To improve scientific usability, we further introduce an LLM-assisted post-hoc explainer that translates symbolic model evidence into chemically grounded narratives, helping domain experts interpret

predictions in practice. Future work should focus on incorporating inexpensive quantum descriptors for electronically delicate systems and employing active learning to better explore underrepresented physicochemical regimes.

# 7. Limitations

DISSOLVR targets organic small molecules and therefore inherits three principled limitations. (i) **Endothermic-dissolution assumption.** Our monotonic temperature constraints encode the standard van't Hoff expectation that solubility increases with $T$. A source-wise audit shows this holds for $> 96\%$ of solute–solvent pairs in BigSolDB 1.0/2.0 (see Appendix H.4, but truly exothermic systems (e.g. $Ce_2(SO_4)_3$, certain inorganic salts) lie outside the model's hypothesis class. (ii) **2D / stereochemical blindness.** Because the featurizer is intentionally lightweight (CPU-deployable, 176 descriptors), it does not encode stereochemistry or 3D conformation; DISSOLVR will return identical predictions for stereoisomers, which is a hard limit for 3D-sensitive drug compounds. (iii) **Inorganic salts are out of scope.** The training corpora (AqSolDB, BigSolDB, Leeds) are dominated by organic small molecules, and the Abraham parameter reconstruction is calibrated for that class; we make no claim of validity on ionic or polymeric systems.

## Reproducibility Statement

To support the reproducibility of our work, we provide several resources in the paper. The source code for our models is available in our `GitHub Repo`.

## Impact Statement

This paper presents work that prioritises interpretability and theory-grounded efficiency, lowering computational barriers and advancing progress through transparent, chemically insightful predictions for solubility.

## Acknowledgements

This work was partially supported by the CSE Research Acceleration Fund of IIT Delhi.

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

# A. Related Work

The prediction of molecular solubility has evolved from thermodynamic heuristics to high-dimensional representation learning. We trace this progression through four distinct phases: classical physicochemical modeling, data-driven representation learning, physics-informed hybrid architectures, and the recent identification of aleatoric limits that challenge the efficacy of further architectural complexity.

## A.1. Physicochemical Foundations and Descriptor-Based Learning

The early solubility prediction was based on parametric thermodynamic relationships. The General Solubility Equation (GSE) (Yalkowsky & Valvani, 1980) established a foundational baseline linking solubility to melting point and $\log P$ as $\log S = -\log P - 0.01\,(M_{\mathrm{pt}} - 25) + 0.5$, where $\log P$ represents the octanol-water partition coefficient, and $M_{\mathrm{pt}}$ is the melting point. However, this definition neglects complex entropic contributions and solvation-dependent conformational changes of the solutes. More rigorous physics-based methods, such as COSMO-RS (Klamt et al., 2010), utilize quantum chemical calculations to estimate chemical potentials in solution. Although thermodynamically robust, COSMO-RS is computationally prohibitive for large-scale screening and is critically dependent on computationally expensive conformer sampling.(Cordova et al., 2024)

The transition to data-driven methods for solubility prediction began with linear regression on explicit molecular descriptors, exemplified by Delaney's ESOL (Delaney, 2004), which achieved an RMSE of 0.97 $\log S$ units, using only four 2D molecular descriptors: calculated $\log P$, molecular weight, rotatable bond count, and aromatic ring fraction, applied to 2,874 compounds. The subsequent approach Moriwaki et al. (2018) scaled this strategy using a high-dimensional feature set with 1800 descriptors, which demonstrated outperformance over fingerprint-based methods when combined with Random Forests (Tayyebi et al., 2023). Crucially, Boobier et al. (2020) showed that manually curated physicochemical descriptors grounded in thermodynamic and quantum theory matched or exceeded many neural network-based approaches, yielding a simple, interpretable model.

## A.2. The Deep Learning Era: Graphs, Sequences, and Ensembles

The dominant paradigm shifted towards end-to-end representation learning with the introduction of MoleculeNet (Wu et al., 2018), where Graph Convolutional Networks (GCNs) learning directly from topology on the publicly available ESOL dataset achieved an RMSE of ∼0.58. Ulrich et al. (2025) employed a similar approach by using GCNs combined with tautomer-based data augmentation to predict aqueous solubility. However, since GCNs are locality-biased, *i.e.* they primarily capture the first and second-order neighbour information; they cannot entirely capture the long-range electrostatic and global molecular properties which significantly influence solubility behaviour. This was thus refined by using Message Passing Neural Networks (MPNNs) by including learned message and update functions alongside a permutation-invariant readout phase to aggregate local node features into a global molecular representation (Gilmer et al., 2017), and by introducing a self-attention-based MPNN, Tang et al. (2020) achieved an RMSE of 0.635 on ESOL. Lee et al. (2023) developed Multi-Order Graph Attention Networks (MoGAT), which improved accuracy and provided atomic importance scores consistent with chemical intuition.

The shift toward learned representations was popularised by the Chemprop framework (Heid et al., 2024), which implemented a Directed Message-Passing Neural Network (D-MPNN). Unlike many earlier GCNs that aggregate messages from all neighbours, the D-MPNN architecture utilizes directed edges to prevent 'tottering' (cyclical passing of information back to a source node) so it learns more distinct and refined molecular embeddings. Chemprop demonstrates improved performance for chemical property prediction tasks, but it remains largely a 'black-box' approach that prioritizes latent feature optimization over physical interpretability. This limitation extends to attention-based mechanisms more broadly. Although attention has increased model transparency by highlighting salient atoms or substructures, Zhang et al. (2024) note that it remains difficult to infer a model's underlying decision-making logic solely from attention weights. In particular, while attention weights indicate where a model is "focusing," they do not inherently explain the model's decision-making logic or prove that the identified structures are the causal drivers of the predicted property.

In parallel to graph-based methods, sequence-based architectures adapt natural language processing techniques to chemical notation. SolTranNet (Francoeur & Koes, 2021) applied a lightweight Molecular Attention Transformer with only 3393 parameters to SMILES strings, counter-intuitively outperforming larger graph-based architectures on certain splits, challenging assumptions about model complexity. ChemBERTa (Chithrananda et al., 2020) utilized Transformers pre-trained on

a curated corpus of 77 million SMILES strings from PubChem. By utilizing self-supervised masked language modeling, the framework enables transfer learning in which the model first learns a general latent grammar of molecular structures before fine-tuning for downstream tasks, achieving an RMSE of ~0.61 on the ESOL dataset. Despite their scalability, SMILES-based approaches are inherently lossy, as linearized string representations do not explicitly encode 3D geometry and stereochemistry, crucial factors known to influence solubility.(Panapitiya et al., 2022) To further improve predictive performance, recent works have employed stacked ensemble strategies, such as the YZS-Model (Wang et al., 2024a) combining GCNs, Transformers, and LSTM networks, with ablation studies identifying the transformer module as the primary driver for accuracy. Similarly, AqSolPred (Sorkun et al., 2021) also employs a similar approach, combining neural networks, Random Forests, and XGBoost. However, such ensembles trade interpretability for accuracy gains and incur both computational costs and loss of explainability.

### A.3. Interpretability and Physics-Informed Architectures

Although attention mechanisms provide a simple post-hoc interpretation of model focus, Layer-Wise Relevance Propagation (LRP) (Binder et al., 2016) provides a more rigorous framework, back-propagating relevance scores layer-by-layer to assign each input feature a contribution score. Chen et al. (2023) introduced SolubNet, a graph-based framework utilizing Topology Adaptive Graph Convolutional Networks (TAGCN) for aqueous solubility prediction, using LRP to decompose model predictions into atomic-level importance scores. This identified that the model correctly assigned high relevance to polar regions (e.g. as hydroxyl and amine groups) aligning with the chemical intuition that these moieties drive solubility.

Recent work attempts to reconcile data-driven learning with physical chemistry. MolMerger (Ramani & Karmakar, 2024) introduces a physics-informed graph representation explicitly encoding solute-solvent interactions. Rather than representing molecules as isolated graphs, MolMerger adds virtual edges between the most polar and least polar atoms of each molecule, respectively, and then employs AttentiveFP (Xiong et al., 2020). This approach, however, assumes that Gasteiger charges adequately represent all solute-solvent interactions. RILOOD (yeyunchen, 2025) introduces a relational invariant learning framework that models multigranularity solute–solvent interactions and latent environments via a mixup-enhanced conditional VAE and context-aware refinement to improve out-of-distribution robustness in solvation free energy prediction across varying solvents and scaffolds. Solvaformer (Broadbent et al., 2025) combines COSMO-RS calculations with machine learning corrections, achieving state-of-the-art performance on BigSolDB 2.0 (Krasnov et al., 2025), with the caveat of computational overhead worth days for training. Geometric deep learning models, such as ViSNet (Wang et al., 2024b) and OrbNet-Equi (Qiao et al., 2022). VisNet captures complex geometric information (e.g. angles and dihedral angles) by utilising vector-scalar interactive message passing. OrbNet-Equi (Qiao et al., 2022), informed by electronic structure (orbital basis functions), ensures equivariance to isometric basis transformations. While theoretically superior since they explicitly encode physical constraints, these methods introduce dependencies on expensive 3D conformer generation and are computationally expensive.

### A.4. The Aleatoric Limit and Evaluation Crisis

A critical reassessment of the field suggests that architectural complexity often masks fundamental data limitations. Attia et al. (2025) and Palmer & Mitchell (2014) document that the experimental noise floor (aleatoric uncertainty) for solubility lies between 0.5–1.0 log units due to various factors as outlined in Appendix D. This gives a fundamental upper bound on the accuracy of models - and thus models claiming RMSE values below this threshold would most likely rely on aggressive dataset curation, potentially inadvertently removing "hard" cases, but creating an illusion of progress; Llompart et al. (2024) also warns that non-standardised curation might bias models against real-world variability, finding that data leakage and non-standardized curation are pervasive issues within the research community.

## B. Dataset Curation and Split Details

This appendix provides complete details on the dataset curation procedures implemented to ensure data quality and prevent leakage. It also contains information about splits used to test OOD generalisation tasks.

All Regime-I datasets test solute extrapolation with moderate difficulty, where test solutes show a mean maximum Tanimoto similarity of 0.58-0.64 to their nearest training neighbours, with 10-19% of test molecules sharing less than 0.4 similarity to all training compounds. In Regime-II, the BigSol 1.0 and 2.0 datasets evaluate solvent extrapolation under extreme distribution shift, with 100% novel solvents in the test set (mean maximum similarity ~0.39 to training solvents) and 60% of test solvents having similarity below 0.4 to any training solvent. The Leeds dataset tests solute extrapolation in a fixed

solvent space, with zero overlap between the training (709 solutes) and test (1,440 solutes) sets, while maintaining consistent solvents across splits (2 of 3 test solvents present in the training set), yielding a mean maximum solute similarity of 0.48.

## B.1. AqSolDB Curation

To generate the robust training set for Regime I, we processed the raw AqSolDB CSV using the following pipeline:

1. **Standardization:** All SMILES strings were canonicalized using RDKit . Isomeric information was discarded to focus on robust 2D topological descriptors.
2. **Mixture Removal:** We filtered out entries containing the "." character in the SMILES string, effectively removing salts, ionic liquids, and multi-component mixtures which introduce non-additive solubility effects.
3. **Physical Constraints:** We applied rigorous thresholds:
   - **Solubility Range:** $-13.5 \leq \log S \leq 0.5$. Values outside this range were discarded as extremely soluble or extremely unsoluble.
   - **Molecular Weight:** MW $\leq 900$ Da, restricting the domain to drug-like small molecules.
4. **Anti-Leakage Protocol:** We explicitly identified all molecules present in Set 1 (an estimated inter-laboratory reproducibility of ∼0.17 log unit.) of the *Second Solubility Challenge* (SC2) dataset ($N_{SC2} = 100$) and removed them from the training set. This ensures our external validation on SC2 is strictly blind.
5. **Final Aggregation:** Duplicate entries were aggregated by mean solubility. The final curated dataset contains **8,311** unique molecules (reduced from 9,982).

For aqueous solubility (Regime I), we adhere to a standard 90/10 random split for the large-scale AqSolDB dataset. For the substantially smaller ESOL benchmark, we adopt a more conservative 80/20 split to ensure the test set is sufficiently large to yield statistically stable performance metrics.

## B.2. BigSolDB Curation

For the multi-solvent Regime II, we utilized BigSolDB 2.0. The curation pipeline involved:

1. **Solvent Standardization:** We utilized a strict alias map to normalize solvent names (e.g., "THF" → "tetrahydrofuran", "DMS" → "methylthiomethane") and verify them against the `thermo` library database.
2. **Polymer Exclusion:** We explicitly excluded polymeric solvents with variable molecular weights (e.g., PEG-400, Span 80, PEGDME 250) as their molar volume is ill-defined, making accurate unit conversion impossible.
3. **Tautomer Standardization:** We employed `rdMolStandardize.TautomerEnumerator` to canonicalize solutes. This ensures that different tautomeric forms of the same drug (e.g., keto-enol variants) reported by different labs are treated as identical entities.
4. **Quality Control:** We grouped replicates by (Solute, Solvent, Temperature) and calculated the inter-laboratory standard deviation. Groups with $\sigma > 0.7$ LogS units were discarded as unreliable.
5. **External Validation Integrity:** To ensure strictly blind evaluation on the external Leeds Dataset, we cross-referenced the training corpus against the Leeds test set and removed all overlapping solutes prior to model training when attempting to evaluate on Leeds, thereby preventing data leakage.

We train on the Top-$K$ most frequent solvents and evaluate on the remaining rare solvents, utilising a calibrated $K$ to produce an approximate 80:20 partition. This creates a strict transfer task that tests generalisation to data-poor environments.

**BigSolDB version-to-split mapping.** For Regime II we use BigSolDB 2.0 as the primary multi-solvent corpus (Top-21 solvent split) and BigSolDB 1.0 as a smaller cross-check (Top-19 solvent split). The Leeds dataset is used as a strictly held-out solute-extrapolation benchmark, with overlapping solutes removed from the BigSolDB training corpus prior to training (cf. "External Validation Integrity" above). Identical curation, alias maps and quality-control thresholds are applied to both BigSolDB versions.

## B.3. Baseline Featurization

To ensure a like-for-like comparison, all generic tree-based baselines (Decision Tree, Random Forest, XGBoost, LightGBM, CatBoost-from-scratch) and the shallow ANN were given exactly the same 176-dimensional physicochemical descriptor vector that DISSOLVR uses (Table 12). Graph baselines (GCN, GIN, GAT, MPNN) consumed the molecular graph through DeepChem's `GraphConvMol` featurizer with default atom and bond descriptors, matching the standard practice used

in their respective original publications. Specialised baselines (Chemprop, FastProp, SolTranNet, AqSolPred, SolubNet, FastSolv, RILOOD) were run with the featurizers prescribed by their authors. This protocol isolates the contribution of DISSOLVR's architecture: any improvement over the generic tree-based baselines is attributable to the Interaction Block and monotonic constraints rather than to additional input information.

### B.4. Density Calculations for LogS Conversion

Machine learning models require a consistent target variable. While BigSolDB reports solubility in mole fraction ($x$), predictive tasks require molarity ($\log S$, mol/L). We performed this conversion dynamically using the `thermo` library:

$$\log S(\text{mol/L}) = \log_{10}\left( x \cdot \frac{\rho_{\text{solvent}}(T)}{\text{MW}_{\text{solvent}}} \right) \tag{9}$$

where $\rho_{\text{solvent}}(T)$ is the density of the pure solvent (g/L) at the specific experimental temperature $T$, and $\text{MW}_{\text{solvent}}$ is the molecular weight (g/mol). By caching these density lookups, we ensured accurate conversion even for temperature-dependent expansion, rather than assuming standard density at 298K.

## C. Complete Baseline Results

Refer to Tables 6 and 7 for complete baseline results.

*Table 6.* **Regime I Results (Aqueous Only).** Comparison on fixed-solvent datasets. DISSOLVR demonstrates consistent performance across small and large data regimes, outperforming deep learning baselines. Reported mean ± std over seeds. Dark Green represents the best model, and lighter greens represent the second-best model (within variance). Generic tree-based models were provided the same features as DISSOLVR, and graph-based models utilised the GraphConvMol Featurizer.

| Method | ESOL | | AqSolDB | | 2nd Sol. Challenge | |
|---|---|---|---|---|---|---|
| | RMSE | $R^2$ | RMSE | $R^2$ | RMSE | $R^2$ |
| GSE | 2.2623 | -0.1669 | 2.5816 | -0.3779 | 2.7283 | -3.6450 |
| ESOL Model | 0.9802 | 0.7809 | 1.3161 | 0.6419 | 1.0445 | 0.3192 |
| CHEMPROP | $0.5749_{\pm 0.0202}$ | $0.9245_{\pm 0.0054}$ | $0.8833_{\pm 0.0234}$ | $0.8386_{\pm 0.0086}$ | $1.0192_{\pm 0.0316}$ | $0.3511_{\pm 0.0400}$ |
| FASTPROP | $0.5485_{\pm 0.0157}$ | $0.9314_{\pm 0.0039}$ | $1.0514_{\pm 0.0711}$ | $0.7704_{\pm 0.0.0314}$ | $1.0329_{\pm 0.0700}$ | $0.3312_{\pm 0.0919}$ |
| SOLTRANNET | $0.8330_{\pm 0.0732}$ | $0.8406_{\pm 0.0274}$ | $1.5717_{\pm 0.0170}$ | $0.4892_{\pm 0.0111}$ | $1.1852_{\pm 0.1232}$ | $0.1140_{\pm 0.1864}$ |
| AQSOLPRED | $0.5033_{\pm 0.0035}$ | $0.9422_{\pm 0.0008}$ | $0.8270_{\pm 0.0060}$ | $0.8437_{\pm 0.0023}$ | $0.9490_{\pm 0.0269}$ | $0.4351_{\pm 0.0322}$ |
| SOLUBNET | $0.7749_{\pm 0.0253}$ | $0.8629_{\pm 0.0090}$ | $1.1802_{\pm 0.0044}$ | $0.7120_{\pm 0.0022}$ | $0.9412_{\pm 0.0100}$ | $0.4471_{\pm 0.0117}$ |
| Ulrich et al. (2025) | $0.9823_{\pm 0.0996}$ | $0.7777_{\pm 0.0439}$ | $1.1933_{\pm 0.0516}$ | $0.7051_{\pm 0.0252}$ | $1.2129_{\pm 0.1050}$ | $0.0751_{\pm 0.1612}$ |
| Tayyebi et al. (2023) | $0.6926_{\pm 0.0053}$ | $0.8906_{\pm 0.0017}$ | $0.9064_{\pm 0.0026}$ | $0.8301_{\pm 0.0010}$ | $0.9102_{\pm 0.0063}$ | $0.4829_{\pm 0.0071}$ |
| Decision Tree | $0.7745_{\pm 0.0205}$ | $0.8631_{\pm 0.0074}$ | $1.1002_{\pm 0.0097}$ | $0.7498_{\pm 0.0044}$ | $1.1375_{\pm 0.0115}$ | $0.1924_{\pm 0.0164}$ |
| Random Forest | $0.5429_{\pm 0.0006}$ | $0.9328_{\pm 0.0001}$ | $0.8902_{\pm 0.0002}$ | $0.8362_{\pm 0.0001}$ | $0.8784_{\pm 0.0006}$ | $0.5184_{\pm 0.0007}$ |
| XGBoost | $0.5005_{\pm 0.0056}$ | $0.9429_{\pm 0.0013}$ | $0.8311_{\pm 0.0029}$ | $0.8572_{\pm 0.0010}$ | $0.9014_{\pm 0.0171}$ | $0.4928_{\pm 0.0193}$ |
| LightGBM | $0.4773_{\pm 0.0104}$ | $0.9480_{\pm 0.0022}$ | $0.8457_{\pm 0.0037}$ | $0.8522_{\pm 0.0013}$ | $0.9338_{\pm 0.0118}$ | $0.4558_{\pm 0.0137}$ |
| ANN | $0.8221_{\pm 0.0818}$ | $0.8444_{\pm 0.0304}$ | $1.1322_{\pm 0.0639}$ | $0.7341_{\pm 0.0293}$ | $1.0403_{\pm 0.0550}$ | $0.3227_{\pm 0.0733}$ |
| GIN | $0.6512_{\pm 0.0415}$ | $0.9029_{\pm 0.0127}$ | $0.9093_{\pm 0.0191}$ | $0.8284_{\pm 0.0072}$ | $1.0326_{\pm 0.0837}$ | $0.3303_{\pm 0.1063}$ |
| GCN | $0.8551_{\pm 0.0806}$ | $0.8318_{\pm 0.0323}$ | $1.0103_{\pm 0.0198}$ | $0.7882_{\pm 0.0083}$ | $1.2728_{\pm 0.0816}$ | $-0.0151_{\pm 0.1325}$ |
| GAT (Veličković et al., 2018) | $0.7877_{\pm 0.0369}$ | $0.8583_{\pm 0.0132}$ | $1.0118_{\pm 0.0277}$ | $0.7875_{\pm 0.0118}$ | $1.3543_{\pm 0.0960}$ | $-0.1504_{\pm 0.1597}$ |
| MPNN | $0.7788_{\pm 0.0621}$ | $0.8608_{\pm 0.0225}$ | $1.0448_{\pm 0.0205}$ | $0.7735_{\pm 0.0089}$ | $1.3772_{\pm 0.1464}$ | $-0.1970_{\pm 0.2533}$ |
| **DISSOLVR** | $0.4631_{\pm 0.0021}$ | $0.9511_{\pm 0.0007}$ | $0.8158_{\pm 0.0030}$ | $0.8624_{\pm 0.0010}$ | $0.9012_{\pm 0.0033}$ | $0.4932_{\pm 0.0037}$ |

## D. Aleatoric Limit Analysis

To contextualize model performance and establish a fundamental lower bound on achievable prediction accuracy, we analyze the irreducible error (aleatoric uncertainty) inherent to experimental solubility measurements.

*Table 7.* **Regime II Results (Multi-Solvent).** Performance under solvent holdout splits and external validation. DISSOLVR demonstrates superior generalisation in multi-solvent environments compared to traditional GNN and tree-based baselines. Reported mean ± std over seeds. Dark Green represents the best model, and lighter greens represent the second-best model (within variance). Generic tree-based models were provided the same features as DISSOLVR, and graph-based models utilised the GraphConvMol Featurizer.

| Method | BigSolDB 1.0 (Top-19 solvents) | | BigSolDB 2.0 (Top-21 solvents) | | Leeds Dataset (Solute Holdout) | |
|---|---|---|---|---|---|---|
| | RMSE | $R^2$ | RMSE | $R^2$ | RMSE | $R^2$ |
| FASTSOLV | $0.7799_{\pm0.0243}$ | $0.5449_{\pm0.0283}$ | $0.6938_{\pm0.0041}$ | $0.5844_{\pm0.0049}$ | $0.8868_{\pm0.0065}$ | $0.3427_{\pm0.0096}$ |
| RILOOD | $0.8814_{\pm0.0509}$ | $0.4201_{\pm0.0675}$ | $0.8791_{\pm0.0458}$ | $0.3346_{\pm0.0710}$ | $1.1566_{\pm0.0163}$ | $-0.1191_{\pm0.0316}$ |
| Decision Tree | $0.9465_{\pm0.0362}$ | $0.3324_{\pm0.0507}$ | $0.9978_{\pm0.0335}$ | $0.1441_{\pm0.0575}$ | $1.4257_{\pm0.0230}$ | $-0.7004_{\pm0.0553}$ |
| Random Forest | $0.7276_{\pm0.0036}$ | $0.6060_{\pm0.0039}$ | $0.7257_{\pm0.0030}$ | $0.5477_{\pm0.0038}$ | $1.0479_{\pm0.0011}$ | $0.0816_{\pm0.0019}$ |
| XGBoost | $0.7289_{\pm0.0051}$ | $0.6047_{\pm0.0055}$ | $0.6416_{\pm0.0019}$ | $0.6465_{\pm0.0021}$ | $0.9442_{\pm0.0110}$ | $0.2542_{\pm0.0176}$ |
| LightGBM | $0.7109_{\pm0.0065}$ | $0.6239_{\pm0.0069}$ | $0.6505_{\pm0.0036}$ | $0.6366_{\pm0.0040}$ | $0.9071_{\pm0.0051}$ | $0.3117_{\pm0.0077}$ |
| ANN | $0.8447_{\pm0.0129}$ | $0.4689_{\pm0.0163}$ | $1.0646_{\pm0.0295}$ | $0.0259_{\pm0.0539}$ | $6.3069_{\pm1.7099}$ | $-34.7137_{\pm18.025}$ |
| GAT | $0.7285_{\pm0.0387}$ | $0.6040_{\pm0.0415}$ | $0.6714_{\pm0.0087}$ | $0.6128_{\pm0.0100}$ | $1.0851_{\pm0.0368}$ | $0.0141_{\pm0.0662}$ |
| GIN | $0.7711_{\pm0.0279}$ | $0.5570_{\pm0.0323}$ | $0.7398_{\pm0.0233}$ | $0.5295_{\pm0.0296}$ | $1.0168_{\pm0.0346}$ | $0.1344_{\pm0.0592}$ |
| MPNN | $0.7723_{\pm0.0635}$ | $0.5532_{\pm0.0759}$ | $0.7382_{\pm0.0333}$ | $0.5311_{\pm0.0436}$ | $1.0860_{\pm0.0293}$ | $0.0129_{\pm0.0527}$ |
| GCN | $0.8180_{\pm0.0157}$ | $0.5019_{\pm0.0190}$ | $0.7463_{\pm0.0258}$ | $0.5211_{\pm0.0327}$ | $1.2049_{\pm0.0806}$ | $-0.2196_{\pm0.1633}$ |
| DISSOLVR | $0.6783_{\pm0.0023}$ | $0.6576_{\pm0.0023}$ | $0.6305_{\pm0.0013}$ | $0.6587_{\pm0.0014}$ | $0.8798_{\pm0.0012}$ | $0.3527_{\pm0.0018}$ |

### D.1. Formal Derivation of the Aleatoric Limit

To contextualize model performance and establish a fundamental lower bound on achievable prediction accuracy, we analyze the irreducible error (aleatoric uncertainty) inherent to experimental solubility measurements. Let the experimentally observed solubility $Y$ for a chemical system $\mathbf{x}$ be modeled as a deterministic physical relationship $f(\mathbf{x})$ perturbed by experimental random noise $\delta$:

$$Y = f(\mathbf{x}) + \delta, \quad \delta \sim \mathcal{N}(0, \epsilon_{\text{noise}}^2) \tag{10}$$

where $\epsilon_{\text{noise}}^2$ denotes the variance arising from experimental uncertainty and inter-laboratory disagreement. Under this formulation, the expected Mean Squared Error (MSE) of any predictive model $M(\mathbf{x})$ admits the standard bias–variance–noise decomposition:

$$\mathbb{E}[(Y - M(\mathbf{x}))^2] = \text{Bias}[M(\mathbf{x})]^2 + \text{Var}[M(\mathbf{x})] + \epsilon_{\text{noise}}^2 \tag{11}$$

Even in the idealized limit of a perfectly specified model with vanishing bias and variance, prediction error is lower-bounded by the conditional variance:

$$\mathbb{E}_{\mathbf{x},Y}[(Y - M(\mathbf{x}))^2] \geq \mathbb{E}_{\mathbf{x}}[\text{Var}(Y|\mathbf{x})] = \epsilon_{\text{noise}}^2 \tag{12}$$

We therefore define the standard deviation $\epsilon_{\text{noise}}$ as the *aleatoric limit* of solubility prediction. This root-mean-square error (RMSE) lower bound has been estimated to lie between $\sim 0.6$ and $0.8 \log_{10}(S)$ units (this metric, however, is dataset-dependent). Any model reporting RMSE $< \epsilon_{\text{noise}}$ should be interpreted cautiously and may indicate either improved data harmonization (lower experimental variance) or potential leakage/correlated measurement noise.

We note that Equation 10 models the experimental uncertainty as a homoscedastic perturbation with a constant variance $\epsilon_{\text{noise}}^2$. Although true solubility measurement noise is heteroscedastic across different compounds and experimental conditions, this single-constant approximation remains mathematically valid for establishing the expected lower bound of the conditional variance over the entire dataset distribution.

### D.2. Noise Benchmarks

Accurate measurement of solubility is intrinsically challenging, and reported values are well known to exhibit substantial experimental uncertainty. Unlike many physicochemical properties, solubility measurements are highly sensitive to subtle variations in experimental protocol, sample preparation, and material form. In practice, organic compounds are frequently measured as amorphous solids, hydrates, polymorphs, solvates, or impure cocrystals rather than as a single well-defined crystalline phase, leading to systematic discrepancies between nominally identical experiments (Jorgensen & Duffy, 2002).

Additional sources of variability arise from differences in equilibration time, temperature control, analytical technique, and post-processing or data interpretation.

As a result, solubility datasets compiled from the literature inherently reflect inter-laboratory disagreement rather than a single well-defined ground truth. Multiple large-scale studies have quantified this variability and consistently report substantial experimental noise floors:

- Katritzky et al. (1998) analyzed solubility measurements for 411 compounds and reported an average inter-laboratory standard deviation of approximately **0.58** $\log S$ units.
- Independent assessments by Palmer & Mitchell (2014) and Hughes et al. (2008) observed experimental standard deviations consistently in the range of **0.60–0.70** $\log S$ units across diverse compound sets.
- In the Second Solubility Challenge, Llinas et al. (2020) demonstrated that inter-laboratory measurements of the same solute–solvent system could differ by as much as **0.86 to 1.56** $\log S$ units, highlighting the magnitude of real-world experimental variability.
- Andersson et al. (2016) showed that differences in data analysis and interpretation alone can induce high inter-laboratory variability (with CVs exceeding 100% for poorly soluble compounds like aprepitant), which can be significantly reduced through standardized data processing protocols.

Taken together, these findings consistently indicate that the aleatoric limit for solubility prediction lies in the range of **0.6–0.8** $\log S$ **units**. This range represents a practical hard ceiling on achievable predictive accuracy when training on heterogeneous literature data; improvements beyond this threshold would require higher-fidelity experimental measurements rather than more sophisticated modeling techniques.

**Inter- vs. intra-laboratory variance: an upper bound.** We emphasise that the $0.6$–$0.8$ $\log S$ range cited above is a measure of *inter-laboratory* disagreement: it aggregates across protocol, sample preparation, equilibration, and analytical-method differences between labs. Single-lab (*intra-laboratory*) reproducibility is typically substantially smaller – often $\lesssim 0.2 \log S$ for well-controlled crystalline phases (Andersson et al., 2016; Palmer & Mitchell, 2014) – so the irreducible noise floor faced by any model trained on harmonised single-lab data is correspondingly lower. Consequently, the $0.6$–$0.8$ figure should be interpreted as an *upper bound* on the true aleatoric limit relevant to literature-aggregated benchmarks like BigSolDB and AqSolDB, rather than a fundamental lower bound on every solubility-prediction problem.

### D.3. Model Evaluation in the Context of the Aleatoric Limit

This experimental context is essential for interpreting DISSOLVR's performance. When trained on BigSolDB 1.0 and evaluated on the external Leeds dataset for solute extrapolation, our model achieves a test RMSE of $\sim$**0.90** $\log S$ units.

Given the literature-established experimental noise floor of $\sigma_{\exp} \approx 0.6 - 0.8$, the remaining gap between observed performance and the theoretical minimum is relatively small. This suggests that DISSOLVR captures the majority of the learnable signal in the data and is approaching the limits imposed by experimental reproducibility. Further improvements in RMSE are therefore more likely to arise from advances in dataset curation and experimental standardization (as exemplified by newer resources such as BigSolDB 2.0) rather than from increased model complexity.

## E. Computational Complexity Analysis

We analyse the computational efficiency of the DISSOLVR framework from a theoretical and empirical perspective. Our goal is to clarify why the proposed architecture enables fast training on commodity hardware without relying on specialised accelerators, and is competitive with deep learning architectures trained on GPU accelerators.

### E.1. Theoretical Complexity

DISSOLVR is built around Gradient Boosted Decision Trees (GBDTs) operating on fixed-length molecular descriptors. This design choice yields favourable scaling properties in both training and inference.

- **Training Complexity.** The training cost of a GBDT model scales as $\mathcal{O}(T \cdot N \cdot K \cdot L)$, where $T$ is the number of trees, $K$ is the number of input features, $N$ is the number of training samples, and $L$ is the tree depth. CatBoost employs histogram-based split finding with oblivious trees, reducing the effective cost of tree construction to linear scaling in $N$ rather than $N \log N$. As a result, training remains efficient even for large solubility datasets with $N > 10^5$ samples.

- **Inference Complexity.** For a trained ensemble, inference for a single input scales as $\mathcal{O}(T \cdot L)$, where $L$ denotes the tree depth. Importantly, this cost is independent of the molecular size or graph structure. Once molecular descriptors are computed, prediction latency is effectively constant-time with respect to the number of atoms and bonds.

Taken together, these theoretical considerations explain why DISSOLVR exhibits substantially lower computational overhead than contemporary deep geometric learning approaches. The empirical training times and inference throughputs reported in Tables 9, 10, and 11 are a direct consequence of this architectural simplicity, enabling efficient deployment on standard consumer-grade CPUs. The model AqSolPred (Sorkun et al., 2021) is primarily a combination of tree-based models and a shallow MLP, and thus it does not benefit from GPU acceleration. The model introduced by Tayyebi et al. (2023) cannot be run on GPUs because it is tree-based. All generic baselines (tree-based and GNN-based models) were run on commodity CPUs by choice.

### E.2. Empirical Benchmarks

**Parameter count and inference throughput.** As detailed in Table 8, the full DISSOLVR pipeline (Regime II Interaction Block plus the final CatBoost regressor) operates with an $\sim 81\times$ reduction in trainable parameters compared to FastSolv (Attia et al., 2025). Consequently, the complete pipeline runs end-to-end in under one second on commodity laptops without GPU acceleration, achieving up to a $7.6\times$ speedup during inference.

*Table 8.* Comparison of parameter counts and end-to-end inference throughput on an Apple M2 CPU (mean over five runs). DISSOLVR achieves an $81\times$ reduction in parameters and sub-second inference across all evaluated datasets.

| Model | Parameters | Inference Time (s) | | |
| --- | --- | --- | --- | --- |
| | | **BigSolDB 1.0** | **BigSolDB 2.0** | **Leeds** |
| FastSolv | 74,760,004 | 2.169 | 4.115 | 0.393 |
| DISSOLVR (Ours) | **921,089** | **0.396** | **0.539** | **0.224** |
| *Improvement* | $81\times$ fewer | $5.5\times$ faster | $7.6\times$ faster | $1.7\times$ faster |

Experiments were conducted on a variety of hardware. Models were first run on commodity hardware (Apple M2 chip, 8GB RAM, 8 cores). However, some deep models were trained on T4 GPUs due to slow performance on commodity hardware. Tables 9, 10, and 11 compare performance for Regime I (aqueous solubility) and Regime II (multi-solvent prediction).

*Table 9.* **Computational Efficiency (Regime I: AqSolDB).** Training time in seconds per seed. (a) Classical, shallow, and graph-based models.

| Metric | DT | RF | XGB | LGBM | ANN | GCN | GIN | GAT | MPNN |
| --- | --- | --- | --- | --- | --- | --- | --- | --- | --- |
| Train Time (s) | $\sim 0.5$ | $\sim 415$ | $\sim 23$ | $\sim 8$ | $\sim 13$ | $\sim 48$ | $\sim 58$ | $\sim 63$ | $\sim 141$ |
| Hardware | M2 CPU | M2 CPU | M2 CPU | M2 CPU | M2 CPU | M2 CPU | M2 CPU | M2 CPU | M2 CPU |

*Table 10.* **Computational Efficiency (Regime I: AqSolDB).** Training time in seconds per seed. (b) Deep chemistry and literature baselines.

| Metric | Chemprop | FastProp | SolTranNet | AqSolPred | SolubNet | Ulrich et al. (2025) | Tayyebi et al. (2023) | **DISSOLVR** |
| --- | --- | --- | --- | --- | --- | --- | --- | --- |
| Train Time (s) | $\sim 137$ | $\sim 376$ | $\sim 604$ | $\sim 30$ | $\sim 768$ | $\sim 3332$ | $\sim 322$ | $\sim 198$ |
| Hardware | T4 GPU | M2 CPU | T4 GPU | M2 CPU | T4 GPU | M2 CPU | M2 CPU | **M2 CPU** |

*Table 11.* **Computational Efficiency (Regime II: BigSolDB 1.0).** Training time in seconds per seed.

| Metric | FastSolv | RILOOD | ChemProp | DT | RF | XGB | LGBM | ANN | GCN | GIN | GAT | MPNN | **DISSOLVR** |
| --- | --- | --- | --- | --- | --- | --- | --- | --- | --- | --- | --- | --- | --- |
| Train Time (s) | $\sim 525$ | $\sim 1658$ | $\sim 1917$ | $\sim 1.5$ | $\sim 40$ | $\sim 37$ | $\sim 16$ | $\sim 95$ | $\sim 271$ | $\sim 271$ | $\sim 422$ | $\sim 1034$ | $\sim 540$ |
| Hardware | T4 GPU | T4 GPU | T4 GPU | M2 CPU | M2 CPU | M2 CPU | M2 CPU | M2 CPU | M2 CPU | M2 CPU | M2 CPU | M2 CPU | **M2 CPU** |

# F. Theorems and Proofs

This appendix gives the formal support for Lemmata 3.1 and 3.2 and Theorem 3.1. Because boosted trees are piecewise constant, the derivative notation in Equation 7 should be read as coordinate-wise monotonicity: increasing $T$ or $T_r$ while holding all other descriptors fixed cannot decrease the prediction, and increasing $1/T$ while holding all other descriptors fixed cannot increase it.

**Lemma F.1** (Closure of Monotone Ensembles). *Let $\mathcal{H}_{\mathcal{M}}$ denote the class of prediction functions that are coordinate-wise nondecreasing in $T$ and $T_r$ and coordinate-wise nonincreasing in $1/T$, holding all other descriptors fixed. If $f, g \in \mathcal{H}_{\mathcal{M}}$ and $\alpha, \beta \geq 0$, then $\alpha f + \beta g \in \mathcal{H}_{\mathcal{M}}$. Consequently, any boosted additive ensemble with nonnegative learning rates remains monotone whenever each base learner is monotone.*

## F.1. Proof of Lemma 3.1

We prove the claim for a positively constrained coordinate $z \in \{T, T_r\}$; the proof for the negatively constrained coordinate $1/T$ is identical after reversing the inequality. Let $x$ and $x'$ be two inputs that differ only in $z$, with $z(x') \geq z(x)$. Since $f, g \in \mathcal{H}_{\mathcal{M}}$,

$$f(x') \geq f(x), \qquad g(x') \geq g(x). \tag{13}$$

For $\alpha, \beta \geq 0$,

$$(\alpha f + \beta g)(x') - (\alpha f + \beta g)(x) = \alpha\big(f(x') - f(x)\big) + \beta\big(g(x') - g(x)\big) \geq 0. \tag{14}$$

Thus $\alpha f + \beta g$ is nondecreasing in $z$. For $1/T$, if $(1/T)(x') \geq (1/T)(x)$, then $f(x') \leq f(x)$ and $g(x') \leq g(x)$, so the same nonnegative linear combination is nonincreasing. Applying this argument to every constrained coordinate proves closure.

For a boosted ensemble $F_m = F_0 + \sum_{t=1}^{m} \eta_t f_t$, the initialization $F_0$ is constant and hence monotone. If every $f_t \in \mathcal{H}_{\mathcal{M}}$ and every learning rate $\eta_t \geq 0$, induction using the closure argument gives $F_m \in \mathcal{H}_{\mathcal{M}}$ for all $m$. $\square$

**Proposition F.1** (Capacity Restriction Under Monotonicity). *Let $\mathcal{H}$ be an unconstrained tree-ensemble class and let $\mathcal{H}_{\mathcal{M}} = \mathcal{H} \cap \mathcal{M}$ be the subclass satisfying the monotonicity constraints above. For any fixed sample $S = \{x_i\}_{i=1}^{N}$,*

$$\widehat{\mathfrak{R}}_S(\mathcal{H}_{\mathcal{M}}) \leq \widehat{\mathfrak{R}}_S(\mathcal{H}), \tag{15}$$

*where $\widehat{\mathfrak{R}}_S$ denotes empirical Rademacher complexity. Moreover, for any vector of residual noise $\epsilon \in \mathbb{R}^N$,*

$$\inf_{h \in \mathcal{H}_{\mathcal{M}}} \sum_{i=1}^{N} \big(\epsilon_i - h(x_i)\big)^2 \geq \inf_{h \in \mathcal{H}} \sum_{i=1}^{N} \big(\epsilon_i - h(x_i)\big)^2. \tag{16}$$

*Thus monotonicity cannot increase the class's ability to fit arbitrary noise; it can only leave that ability unchanged or reduce it.*

## F.2. Proof of Proposition F.1

The first inequality follows directly from set inclusion. By definition,

$$\widehat{\mathfrak{R}}_S(\mathcal{A}) = \frac{1}{N}\mathbb{E}_\sigma\left[\sup_{h \in \mathcal{A}} \sum_{i=1}^{N} \sigma_i h(x_i)\right], \tag{17}$$

where $\sigma_i$ are independent Rademacher signs. Since $\mathcal{H}_{\mathcal{M}} \subseteq \mathcal{H}$, the supremum over $\mathcal{H}_{\mathcal{M}}$ is bounded above by the supremum over $\mathcal{H}$ for every realization of $\sigma$, and the expectation preserves the inequality.

The squared-error statement is the same inclusion argument applied to empirical risk minimization. Optimizing over the smaller class $\mathcal{H}_{\mathcal{M}}$ cannot achieve a smaller residual sum of squares than optimizing over the larger class $\mathcal{H}$. This is the formal sense in which hard monotonic constraints restrict noise-fitting capacity. The strictness and practical magnitude of this restriction are data-dependent; in DISSOLVR, they are measured empirically in Appendix H.3. $\square$

**Lemma F.2** (Efficiency of Monotonic Optimization). *Consider training monotone oblivious-tree base learners $f_t \in \mathcal{H}_{\mathcal{M}}$ of depth $L$, where $d_t$ features are subject to monotonic constraints on $N$ samples with $K$ features. Then:*

1. **Complexity**: *Training incurs $O(NKL)$ cost per tree for split search and $O(d_t \cdot 2^L)$ for leaf adjustment. For realistic configurations ($L \leq 10$, $d_t \ll K$, $N \geq 10^4$), the adjustment cost is negligible: $O(d_t \cdot 2^L) \ll O(NKL)$.*

2. **Closure under boosting**: *The ensemble $F(x) = \sum_{t=1}^{T} \eta f_t(x)$ remains in $\mathcal{H}_{\mathcal{M}}$ for learning rates $\eta \geq 0$, without post-hoc projection.*

### F.3. Proof of Lemma 3.2

**Split Search Complexity:** CatBoost employs **oblivious decision trees**, where all nodes at depth $\ell$ split on the same feature. The split-search phase proceeds level-by-level.

At each level $\ell \in \{0, \ldots, L-1\}$:

- Evaluate all $K$ candidate features via histogram aggregation: $O(NK)$.
- For constrained features $x_j \in \mathcal{C}$, monotonicity checking adds constant-time feasibility checks during gain computation.
- Select the split maximizing the constrained gain criterion.

Total split-search cost is $\mathcal{O}(NKL)$ per tree, matching the unconstrained order.

**Leaf Value Adjustment:** After fixing the tree structure, leaf values are computed subject to the monotone ordering induced by constrained splits:

$$\boldsymbol{\ell}^* = \arg\min_{\boldsymbol{\ell} \in \mathbb{R}^{2^L}} \sum_{i=1}^{N} \mathcal{L}(y_i, F_{t-1}(x_i) + \ell_{r(x_i)}), \tag{18}$$

where $r(x_i) \in \{1, \ldots, 2^L\}$ is the leaf index for sample $x_i$. The monotonic constraints impose partial-order inequalities on the $2^L$ leaves. For a constrained coordinate, any two leaves that differ by increasing that coordinate must have ordered leaf values.

The adjustment is over the leaves of a single tree, not over all $N$ samples. Enforcing multiple interacting monotonic constraints requires projecting the leaf values onto a partially ordered monotone cone. While this cannot be decomposed into $d_t$ independent one-dimensional isotonic regressions, the complexity of this finite-poset projection depends purely on the number of leaves (a function of $2^L$) rather than the number of training samples $N$. Thus the additional constrained-optimization cost is lower order relative to split search for shallow trees.

**Dominance Analysis:** For typical solubility prediction tasks:

- Tree depth: $L \in [6, 10] \Rightarrow 2^L \in [64, 1024]$.
- Constrained features: $d_t \leq 4$ (e.g., $T, 1/T, T_r, T_m$).
- Dataset size: $N \geq 10^4$ (AqSolDB), $N \geq 10^5$ (BigSolDB).
- Total features: $K \in [176, 380]$.

The representative cost ratio is:

$$\frac{d_t \cdot 2^L}{NKL} \approx \frac{4 \times 1024}{10^4 \times 176 \times 8} \approx 3 \times 10^{-4} < 0.03\%. \tag{19}$$

Therefore, the leaf-adjustment overhead is negligible in this regime, and the dominant term remains $\mathcal{O}(TNKL)$ for an ensemble of $T$ trees.

**Closure under Boosting:** By construction, each base learner $f_t$ satisfies the specified monotonicity constraints. Lemma F.1 shows that a nonnegative additive combination of monotone learners is monotone. Since the boosting update has the form

$$F_{t+1}(x) = F_t(x) + \eta f_t(x), \quad \eta > 0, \tag{20}$$

and $F_0$ is constant, induction gives $F_T \in \mathcal{H}_{\mathcal{M}}$ for all $T$. This avoids any need for post-hoc projection of the final ensemble. $\qquad \square$

**Theorem F.1** (Thermodynamically Monotone and Capacity-Restricted Prediction). *The DISSOLVR framework, when trained with monotonic constraints on temperature features $\{T, 1/T, T_r\}$, satisfies:*

1. *Physical Consistency: Under the standard endothermic-dissolution regime, the predicted solubility obeys the coordinate-wise thermodynamic constraints in Equation 7.*
2. *Capacity Restriction: The monotone ensemble class is a subset of the corresponding unconstrained ensemble class. Therefore, on any fixed sample, the monotone class cannot have larger empirical Rademacher complexity or greater ability to fit arbitrary noise than the unconstrained class.*

3. **Computational Feasibility**: *Training complexity remains $\mathcal{O}(TNKL)$ for $T$ trees in the shallow-tree regime used by* DISSOLVR.

## F.4. Proof of Theorem 3.1

**Physical Consistency:** The monotonic constraints are enforced at the base-learner level. By Lemma F.1, the boosted additive ensemble preserves coordinate-wise monotonicity for nonnegative learning rates. Thus, within the endothermic-dissolution regime,

$$\hat{\Sigma}(T') \geq \hat{\Sigma}(T) \text{ for } T' \geq T, \qquad \hat{\Sigma}(T'_r) \geq \hat{\Sigma}(T_r) \text{ for } T'_r \geq T_r, \tag{21}$$

with all other descriptors fixed, and

$$\hat{\Sigma}((1/T)') \leq \hat{\Sigma}(1/T) \text{ for } (1/T)' \geq 1/T. \tag{22}$$

This is the finite-difference version of Equation 7, appropriate for tree ensembles.

**Capacity Restriction:** The constrained learner is optimized over $\mathcal{H}_{\mathcal{M}} = \mathcal{H} \cap \mathcal{M}$, a subset of the unconstrained ensemble class $\mathcal{H}$. Proposition F.1 therefore implies that the constrained class cannot have higher empirical Rademacher complexity or lower best-fit squared error on arbitrary noise than the unconstrained class. The theorem does not require a universal isotonic-regression rate such as $O(N^{1/3})$; the observed reduction in seed variance and failure-mode suppression is reported empirically in Appendix H.3.

**Computational Feasibility:** The monotonicity constraints are enforced during tree construction and leaf-value adjustment. Lemma F.2 shows that the additional leaf-level cost is lower order for the shallow trees and large sample sizes used in DISSOLVR, so the ensemble training complexity remains $\mathcal{O}(TNKL)$. $\qquad\square$

## G. DISSOLVR Methodology

### G.1. Physicochemical Featurizer

Table 12 enumerates the complete set of physicochemical descriptors used by the DISSOLVR featurizer. The features are organized by abstraction level, from explicit compositional counts to higher-order topological motifs and thermodynamic proxies. For each feature group, we report the specific descriptors and their intended physical interpretation, providing a precise accounting of the model's input space.

### G.2. Regime I: Single-Solvent Prediction

For Regime I experiments, solubility was predicted using a CatBoost regression model operating on the full physicochemical feature vector described in Table 12. All descriptors were deterministically pre-computed prior to training and held fixed throughout optimization.

The model was trained to minimize the root mean squared error,

$$\mathcal{L} = \sqrt{\frac{1}{N} \sum_{i=1}^{N} (\Sigma_i - \hat{\Sigma}_i)^2}, \tag{23}$$

with predictions given by

$$\hat{\Sigma} = \text{CatBoost}(\mathbf{x}_{\mathcal{A}}). \tag{24}$$

Training employed a slow-learning configuration (learning rate $\eta = 0.02$, 10,000 boosting iterations, tree depth 8) to stabilize optimization in the high-dimensional feature space. All runs were executed on an Apple M2 CPU over 5 seeds for reproducibility, and results are reported in Table 6.

### G.3. Regime II: Multi-Solvent Interaction Learning (Implementation Details)

This section documents the concrete instantiation of the Interaction Layer used for multi-solvent solubility prediction. All conceptual motivation and high-level design choices are described in the main text; here, we specify the exact representations, transformations, and optimization procedures.

*Table 12.* Complete DISSOLVR Feature Inventory (176 features per molecule)

| Category | Feature Group | Specific Descriptors & Rationale |
|---|---|---|
| **1. Compositional (109)** | *Molecular Weight* | `MolWt`: Total molecular weight as a size descriptor. |
| | *Atom/Ring Counts* | **Structural Census (23):** `HeavyAtomCount`, `NumHeteroatoms`, `NumValenceElectrons`, `RingCount`, `NumAromaticRings`, `NumAliphaticRings`, `NumSaturatedRings`, `NumHAcceptors`, `NumHDonors`, `NumRotatableBonds`, `NHOHCount`, `NOCount`, plus 11 additional RDKit ring/atom counts. |
| | *Functional Fragments* | **RDKit Fragment Counts (All 85 `fr_*` features):** Census of functional groups including: 
 • `fr_ketone`, `fr_aldehyde`, `fr_ester`, `fr_ether` (Polarity) 
 • `fr_aniline`, `fr_sulfonamd`, `fr_amide`, `fr_phenol` (H-Bonding) 
 • `fr_halogen`, `fr_benzene`, `fr_pyridine` (Lipophilicity/Aromaticity) |
| **2. Topological (19)** | *Graph Indices (6)* | **Connectivity Descriptors:** 
 `BalabanJ`, `BertzCT`, `Kappa1`, `Kappa2`, `HallKierAlpha`, `Phi`. |
| | *MoSE Motifs (13)* | **Steric & Packing Constraints:** 
 • **Cycles:** Ring sizes `mose_cyc3` through `mose_cyc8`, `mose_benzene`. 
 • **Complexity:** Fused rings (`mose_fused`), branching (`mose_branched_4`, `mose_star_5`). 
 • **Paths:** `mose_path3`, `mose_path4`, `mose_path5`. |
| **3. Energetic (6)** | *Lattice Energy Proxy* | **Joback Melting Point (`pred_Tm`):** 
 Calculated via group contribution ($T_m \approx 122.5 + \sum N_i T_{c,i}$) to proxy $\Delta G_{\mathrm{fus}}$. |
| | *Abraham Proxies (5)* | **Solvation Parameters:** 
 • `abraham_A` (Acidity), `abraham_B` (Basicity): H-bond donor/acceptor. 
 • `abraham_S` (Polarity): Heteroatom and aromatic density. 
 • `abraham_E` (Polarizability), `abraham_V` (McGowan Volume). |
| **4. Physicochemical (42)** | *Electronic State (8)* | **Charge Distribution (8):** `MaxEStateIndex`, `MinEStateIndex`, `MaxPartialCharge`, `MinPartialCharge`, and their 4 absolute counterparts (`MaxAbs...`). |
| | *Surface Area (28)* | **VSA Distributions (28):** `EState_VSA1-11`, `PEOE_VSA1-14`, `LabuteASA`, `TPSA` (Polar Surface Area), plus 1 dynamically extracted surface proxy. |
| | *Global Properties (6)* | **Global Properties (6):** `MolLogP` (Lipophilicity), `FractionCSP3`, `qed` (Drug-likeness), `AvgIpc`, `SPS`, and remaining dynamically extracted bulk properties. |

### G.3.1. FEATURE TOKENIZATION

Each refined representational vector (Table 1) $\mathbf{c} \in \mathbb{R}^{24}$ is converted into a sequence of feature tokens. For feature index $i$, the embedding is defined as

$$\mathbf{e}_i = (c_i \cdot \mathbf{w}_i + \mathbf{b}_i) + \mathbf{t}_i, \tag{25}$$

where $c_i$ is the scalar feature value, $\mathbf{w}_i \in \mathbb{R}^d$ is a learnable projection vector, $\mathbf{b}_i$ is a bias term, and $\mathbf{t}_i$ is a fixed type embedding. This produces sequence tensors

$$\mathbf{E}_{\mathcal{A}}, \mathbf{E}_{\mathcal{B}} \in \mathbb{R}^{24 \times d},$$

with $d = 32$, for the solute and solvent, respectively.

### G.3.2. CROSS-COUPLING INTERACTION LAYER

Interactions are modeled using multi-head coupling, with the solute sequence querying the solvent sequence. For each head $h \in \{1, \dots, H\}$ (with $H = 4$), linear projections are applied:

$$\mathbf{Q}_h = \mathbf{E}_{\mathcal{A}} \mathbf{W}_h^Q, \tag{26}$$

$$\mathbf{K}_h = \mathbf{E}_{\mathcal{B}} \mathbf{W}_h^K, \tag{27}$$

$$\mathbf{V}_h = \mathbf{E}_{\mathcal{B}} \mathbf{W}_h^V, \tag{28}$$

where $\mathbf{W}_h^Q, \mathbf{W}_h^K, \mathbf{W}_h^V \in \mathbb{R}^{d \times d_k}$ and $d_k = 8$.

Outputs are computed as

$$\text{head}_h = \text{softmax}\left(\frac{\mathbf{Q}_h \mathbf{K}_h^\top}{\sqrt{d_k}}\right) \mathbf{V}_h. \tag{29}$$

The heads are concatenated and linearly projected to form the interaction-aware representation

$$\mathbf{Z}_{\mathcal{A} \to \mathcal{B}} = \text{Concat}(\text{head}_1, \ldots, \text{head}_H)\mathbf{W}^O. \tag{30}$$

**Training & Explainability:** The model was trained using 5-Fold Cross-Validation to minimize Mean Squared Error (MSE) Loss and prevent data leakage:

$$\mathcal{L} = \frac{1}{N} \sum_{i=1}^N (\Sigma_i - \hat{\Sigma}_i)^2 \tag{31}$$

We utilize the Adam optimizer ($LR = 1 \times 10^{-3}$, batch size 64) for 20 epochs per fold. Crucially, the weights matrix $\mathbf{A} \in \mathbb{R}^{24 \times 24}$ provides intrinsic explainability. By averaging these weights (Figure 3), we reveal which solute-solvent feature pairs drive prediction, offering a direct window into the learned chemical logic.

### G.3.3. FINAL PREDICTION AND CONSTRAINTS

Solubility is predicted using a CatBoost regressor:

$$\hat{\Sigma} = \text{CatBoost}\left[\mathbf{x}_{\mathcal{A}}^{\text{full}} \parallel \mathbf{x}_{\mathcal{B}}^{\text{full}} \parallel \mathbf{I} \parallel f(T)\right], \tag{32}$$

where $f(T)$ includes temperature, inverse temperature, predicted melting point, and reduced temperature, and $\mathbf{I}$ is the enthalpic state vector. We note for consistency with the main text that the symbols $\mathbf{x}_{\mathcal{A}}$ and $\mathbf{x}_{\mathcal{B}}$ in the abbreviated equation $\hat{\Sigma} = \text{CatBoost}(\mathbf{x}_{\mathcal{A}}, \mathbf{x}_{\mathcal{B}}, \mathbf{I}, f(T))$ refer to the *full* 176-dimensional physicochemical descriptors $\mathbf{x}_{\mathcal{A}}^{\text{full}}$ and $\mathbf{x}_{\mathcal{B}}^{\text{full}}$ used here, concatenated as inputs to the regressor, and the two formulations are equivalent.

Prior to training, low-variance features were removed using a variance threshold of $10^{-4}$. The model was trained with 3,000 trees (depth = 8) using a slow learning rate of $0.02$ and $L_2$ leaf regularization of 5, with early stopping after 100 rounds based on a 95/5 train–validation split. To enforce thermodynamic consistency, monotonic constraints were applied to temperature-derived features ($T_r, T, 1/T$), while all other features were left unconstrained. Training was performed on an Apple M2 CPU over 5 seeds and results are reported in Table 7. We report both RMSE and $R^2$. RMSE is the primary metric, as it provides a physically interpretable error in $\log S$ units, while $R^2$ enables normalised comparison across datasets with differing variance.

### G.4. Per-Solvent Results on BigSolDB 1.0

We provide a granular breakdown of DISSOLVR's performance across the top 10 solvents in the BigSolDB 1.0 test set. It is important to note that this evaluation follows the **Solvent-Holdout Split** protocol: the model was trained on the top 19 most frequent solvents and evaluated on the remaining "rare" solvents. This constitutes a zero-shot transfer task that tests the model's ability to generalize to new chemical environments.

The figures in 4 illustrate the correlation between experimental and predicted solubility ($\log S$) for the 10 most represented solvents in the holdout set. The dashed line represents the identity ($x = y$) line.

As evidenced by the parity plots, DISSOLVR achieves consistent performance ($RMSE \approx 0.64$ to $0.90$) across structurally distinct solvents such as Benzene ($R^2 = 0.58$) and 1-Butanol ($R^2 = 0.92$). We also tested the Apelblat curve fitted to the experimental data against the predicted values, we found a high pearson correlation of 0.9886 (train) / 0.9844 (test) and Cosine similarities of 0.9398 (train) / 0.8201 (test) (illustrated in Figure 5).

## H. Additional Empirical Results and Ablation Studies

This section provides extended empirical validation for DISSOLVR, including a granular decomposition of learned feature importances, stress-tests of the LLM-explainer's faithfulness, and specific architectural ablations justifying our thermodynamic constraints and interaction layer.

## H.1. Mechanistic Feature Importance Analysis

We performed a SHAP (SHapley Additive exPlanations) analysis on a random subset of the test set predictions of size 500 on the BigSolDB 1.0 trained model for Regime II. This analysis decomposes the model's decision-making process, allowing us to map statistical feature importance directly to the four physicochemical pillars defined in our hypothesis.

**Unified Mechanistic Validation (Multi-Solvent).**   Figure 6 presents the complete decomposition of the model's learned physics, illustrating both the macroscopic thermodynamic drivers and the microscopic fine-tuning.

**Tier 1: Global Thermodynamic Drivers.** As shown in Figure 6a and 6b, the top-ranked features map one-to-one with our hypothesized requirements:

- **Phase-Change Energetics (Pillar 4):** The dominance of `T_inv` ($1/T$) and `T_act` ($T$) confirms the model effectively learned the Van't Hoff equation ($\ln S \propto 1/T$).
- **Topological Context (Pillar 2):** `Solute_TPSA` (Topological Polar Surface Area) appears as the single most important feature. Physically, high TPSA correlates with strong intermolecular hydrogen bonding in the solid state, serving as a robust proxy for Crystal Lattice Energy ($\Delta G_{lattice}$). MACCS keys features also appear as highly-ranked by the model, along with shape-based descriptors like Autocorrelation.
- **Latent Physicochemical State (Pillar 3):** `Solute_MolLogP` (Rank 3) captures the global hydrophobicity/lipophilicity balance, essential for determining the "Like Dissolves Like" baseline in organic solvents.
- **Moietal Identity (Pillar 1):** Descriptors such as `Solute_SlogP_VSA1` and `Solvent_PEOE_VSA8` encode the electronic contributions of specific functional fragments, and explicit functional group counts also appear in Hierarchical importance (Figure 6b).

**Tier 2: Fine-Tuning via Interaction Terms.** While global descriptors set the baseline solubility regime, Figure 6c reveals the critical role of the Interaction Transformer. The learned interaction terms (highlighted in red) do not dominate the global variance but are crucial for correcting predictions based on specific solute-solvent coupling (e.g., dipole-dipole alignment), effectively closing the gap between the baseline approximation and the experimental aleatoric limit.

**Top-Feature SHAP Importances (Aqueous ESOL).**   For a comparative baseline in a single-solvent aqueous regime, Table 13 reports the eight features with the largest mean absolute SHAP values on the ESOL test set. The ranking recapitulates classical solubility chemistry: lipophilicity (`MolLogP`) and partial-charge surface area (`PEOE_VSA6`) dominate, followed by hydrogen-bond/polarisability terms (`Abraham E`, `partial charge`) and structural complexity descriptors (`BertzCT`, `MolWt`, `MoSE_path5`, `NOCount`). All eight remain among the 24 distilled tokens of $\Psi$ (Table 1), confirming that the refined feature set retains the model's most informative dimensions.

*Table 13.* Top-8 mean absolute SHAP values on ESOL.

| Feature | Category | Mean\|SHAP\| |
|---|---|---|
| `MolLogP` | Physicochemical | 0.677 |
| `PEOE_VSA6` | Physicochemical | 0.104 |
| `Abraham E` (polarisability) | Energetic | 0.093 |
| `PartialCharge` | Physicochemical | 0.077 |
| `BertzCT` (complexity) | Topological | 0.069 |
| `MolWt` | Compositional | 0.060 |
| `MoSE_path5` | Topological | 0.045 |
| `NOCount` | Compositional | 0.044 |

## H.2. LLM-Explainer Faithfulness: Trained vs. Untrained Ablation

To validate the faithfulness of our post-hoc explanation pipeline, we conduct a quantitative ablation comparing narratives generated from a fully trained DISSOLVR model against those from a randomly-initialised baseline (Table 14). This ablation serves as a rigorous stress test to ensure the LLM is not simply acting as a salesman, that is, relying on its internal parametric priors, to hallucinate plausible chemical narratives, but is instead strictly conditioning its output on the provided model evidence.

By randomising the underlying predictor, we drastically alter the symbolic artifacts (i.e., SHAP values, decision paths, and the interaction matrix) while holding the prompt template constant. If the LLM were merely falling back on generic chemical knowledge, the explanations for the trained and untrained models would remain highly similar. Instead, the remarkably low agreement scores across all measured dimensions demonstrate that the generated narrative is highly sensitive, and strictly faithful, to the specific numerical evidence supplied by the model. The exact prompts and per-sample raw scores are released alongside the source code in our `GitHub Repo`.

*Table 14.* Agreement between explanations generated for a trained and randomly-initialised DISSOLVR on identical prompts ($n = 30$). Lower is better: low agreement implies that the LLM relies on the specific structural evidence of the trained model rather than its own parametric priors.

| Dimension | Trained vs. Untrained ($n$=30) |
|---|---|
| Conclusion Agreement | $1.37 \pm 0.96$ |
| Reasoning Alignment | $1.53 \pm 0.57$ |
| Causal Attribution | $1.90 \pm 0.76$ |
| Completeness Overlap | $2.53 \pm 0.68$ |
| Overall Agreement | $1.63 \pm 0.76$ |
| **Mean of 4 dimensions** | $\mathbf{1.83 \pm 0.42}$ |

## H.3. Hard Monotonicity Ablation

Table 15 reports the full constrained-vs.-unconstrained sweep across five random seeds. The constraint marginally reduces *mean* RMSE on every dataset, but its dominant effect is on *stability*: seed-to-seed variance collapses by $44$–$72\%$. This matters operationally because a chemist receives one prediction per query, not an average over seeds.

*Table 15.* **Effect of hard monotonic temperature constraints** (mean $\pm$ std over 5 seeds). Variance reduction listed in parentheses.

| Dataset | Constrained | Unconstrained | $\Delta$RMSE |
|---|---|---|---|
| BigSolDB 1.0 | $0.6783 \pm 0.0023$ | $0.6805 \pm 0.0062$ ($-63\%$) | $+0.0022$ |
| BigSolDB 2.0 | $0.6305 \pm 0.0013$ | $0.6335 \pm 0.0023$ ($-44\%$) | $+0.0030$ |
| Leeds (OOD) | $0.8798 \pm 0.0012$ | $0.8846 \pm 0.0043$ ($-72\%$) | $+0.0048$ |

**Stratified analysis.** We further partition the test set by the consistency of the training data available for each solute. For the $18.3\%$ of test samples whose solutes possess multi-source training data with *explicitly conflicting* temperature–solubility trends, the constraint produces a $\Delta$RMSE $= +0.0205$ ($\sim 10\times$ the global average); on the remaining "clean" samples it remains effectively dormant ($\Delta$RMSE $= +0.0018$). Decision trees are piecewise-constant and cannot extrapolate: when high-temperature samples are sparse and contradictory, an unconstrained ensemble reverts to the nearest leaf and produces flat or thermodynamically inverted predictions. Hard monotonicity blocks this failure mode *within every split* at negligible cost (Lemma 3.2).

**Comparison to soft Sobolev penalties.** FastSolv (Attia et al., 2025) enforces temperature consistency through a Sobolev penalty: a soft regulariser that requires three passes per training step (forward, backward with graph retention, and a second backward through the retained graph for the Hessian-vector product $\partial^2 \hat{y}/\partial\theta\partial T$), roughly tripling per-step compute and doubling GPU memory. Furthermore, because the target derivatives for this penalty must be approximated via finite differences, FastSolv imposes a strict dataset curation bottleneck: its training procedure inherently requires experimental measurements at multiple temperatures for the exact same solute-solvent pair. DISSOLVR's constraints (Lemma 3.2), in contrast, are imposed purely at split time. They add no gradient computation, require no multi-temperature paired data, and provide a hard guarantee that soft Sobolev penalties structurally cannot.

**Failure mode of the unconstrained ensemble.** Because CatBoost trees are piecewise-constant, in regions of the input space where high-temperature training points are sparse and mutually inconsistent the unconstrained ensemble defaults to the nearest leaf of the most recently fit base learner. The resulting prediction surface is either flat in $T$ (the leaf is shared

across the local temperature range) or, when conflicting leaves are averaged across boosting rounds, thermodynamically inverted – predicted solubility decreasing with increasing temperature for endothermic dissolution. The hard monotonic constraints described in Lemma 3.2 eliminate this regime *within every leaf* at no asymptotic cost, which is what the $10\times$ stratified RMSE gap on the conflicting subset reflects.

### H.4. Endothermic Assumption Audit

The monotonic constraint we applied assumes a standard endothermic-dissolution regime. In order to quantify how often it is correct, we audited temperature-solubility curves under two criteria: (i) any non-monotonic dip, and (ii) strictly decreasing solubility with temperature. Table 16 shows that strict exothermic behavior is rare in the organic solute–solvent benchmarks used here; this supports the limitation in Section 7 that DISSOLVR is appropriate for organic small molecules but not for inorganic salts or exothermic systems.

Table 16. Frequency of endothermic-assumption violations in the evaluated datasets.

| Dataset | Any violation | Strictly decreasing |
|---|---|---|
| BigSolDB 1.0 | 3.21% | 0.45% |
| BigSolDB 2.0 | 2.67% | 0.51% |
| Leeds | 0.00% | 0.00% |

Under the stricter criterion, fewer than $0.6\%$ of pairs exhibit genuinely decreasing solubility with temperature, and these cases may also include inter-laboratory measurement noise. We therefore keep the hard monotonic constraint for the target domain, while explicitly excluding salts and inorganic systems as shown in Section 7.

### H.5. Direct-Concatenation vs. Interaction Block

A natural baseline to our cross-attention Interaction Block is to simply concatenate the full solute and solvent descriptors and let CatBoost handle the cross-feature interactions internally. We ran this experiment on the OOD Leeds Test with all other hyperparameters held fixed.

Table 17. Cross-attention Interaction Block vs. direct concatenation of full feature vectors (Leeds Test).

| Metric | Interaction Block | Direct Concat. |
|---|---|---|
| Raw feature dim. | 380 | 1,124 |
| Pruned feature dim. | 313 | 1,057 |
| Fit time (s) | 191.9 | 1714.6 |
| Predict time (s) | 0.039 | 0.084 |
| RMSE | 0.8823 | 0.8870 |
| $R^2$ | 0.3489 | 0.3419 |

Direct concatenation expands the feature space by $\sim 3\times$ and training time by $\sim 9\times$, while *degrading* OOD performance. The Interaction Block is therefore not merely a stylistic choice: by collapsing the $D \times D$ cross-product into a $24 \times 24$ structured matrix, it both sharpens generalisation and keeps inference compatible with consumer-grade CPUs.

## I. Case Study: Bezafibrate in 3-Methylbutan-1-ol at 323 K

### I.1. Abstract

We trace a single DISSOLVR prediction end-to-end: bezafibrate (a PPAR-$\alpha$ agonist used to treat hyperlipidemia (Staels et al., 1998)) dissolved in 3-methylbutan-1-ol (isoamyl alcohol) at 323 K. DISSOLVR returns $\log S = -1.0426$ ($\sim 0.091$ mol/L, $\sim 33$ mg/mL, *moderately soluble*), against an experimental label of $-1.1366$ (absolute error 0.094). The exported evidence is dominated by the thermodynamic group ($+0.2922$, $46.6\%$ of the total signal), led by temperature ($\phi = +0.1506$), its inverse ($\phi = +0.1427$) and solute lipophilicity (MolLogP = 3.55, $\phi = +0.1254$), while solute-side penalties from topolog-

ical complexity ($\mathtt{BertzCT} = 739.02$, $\phi = -0.1845$) and polar surface area ($\mathtt{TPSA} = 75.63$, $\phi = -0.0878$) partially offset those gains. The dominant cross-attention rotation ($\mathtt{NumRotatableBonds} \leftrightarrow \mathtt{TPSA}$, $A = 0.5876$) couples conformational flexibility to the solute's polar surface area; Cross-attention weights of 0.5876 for $\mathtt{NumRotatableBonds} \rightarrow \mathtt{TPSA}$ and 0.2611 for $\mathtt{Acidic} \rightarrow \mathtt{abraham\_S}$ anchor the compatibility narrative. Literature lacks a direct measurement of bezafibrate in 3-methylbutan-1-ol, but isopropanol and $n$-propanol proxies place the expected solubility in the same $10^{-2}$–$10^{-1}$ mole-fraction band at 283–323 K, consistent with the prediction. Tree-path variability (std 71.5 over 3,000 trees) and the exported $\mathtt{pred\_Tm}$ (300.65 K vs. experimental $\sim 459$ K) flag residual uncertainty; the Stage 5 verdict was $\mathtt{supported}$ and Stage 5b revision was not required.

## I.2. Introduction

Bezafibrate ($\mathtt{2-[4-[2-[(4-chlorobenzoyl)amino]ethyl]phenoxy]-2-methylpropanoic\ acid}$) is a BCS Class II drug with very low aqueous solubility ($\sim 1.55\,\mu$g/mL at 298 K (Sun et al., 2020)), making its dissolution in organic solvents central to crystallisation, nanosuspension and lipid-based delivery workflows. Solubility in a branched primary alcohol at elevated temperature is a stringent test for a predictor: it requires correctly trading lattice/complexity penalties against thermal activation and H-bond stabilisation. We use this single sample as the worked example for the LLM explainer (App. K.7).

## I.3. Molecular Properties

Bezafibrate (SMILES: $\mathtt{CC(C)(Oc1ccc(CCNC(=O)c2ccc(Cl)cc2)cc1)C(=O)O}$) couples a 4-chlorobenzamide head to a 2-methylphenoxy-2-methylpropanoic-acid tail through an ethylene linker. Key properties:

- **MolWt** 361.8 g/mol; **LogP** 3.46–3.97 (model $\mathtt{Solute\_MolLogP} = 3.5545$; XLogP3 = 3.8).
- **TPSA** 75.6 Å$^2$ (carboxyl, amide, ether donors/ acceptors).
- **Melting point** 457–459 K (high lattice energy).
- **Rotatable bonds** 7; **BertzCT** 739.02; **Cactvs complexity** 452.
- **pK$_a$** $\sim 3.6$ (partial ionisation at neutral pH).

3-Methylbutan-1-ol is a branched primary alcohol with HSP profile ($\delta_D \approx 15$–16, $\delta_P \approx 5$, $\delta_H$ from OH) MPa$^{1/2}$ and LogP $\approx 1.16$; the explainer describes it as a relatively small, branched alcohol ($\mathtt{Solvent\_TPSA} = 20.23$), whose hydroxyl group can stabilise bezafibrate's acid/amide head while its alkyl branching reduces packing efficiency relative to linear alcohols.

## I.4. Solubility Prediction and Model Explanation

DISSOLVR predicts $\log S = -1.0426$ at 323 K (experimental label $-1.1366$; absolute error 0.094). The explainer ingests the Stage 1 molecule descriptions verbatim (App. K.1): bezafibrate as two *para*-substituted rings linked by an amidoethyl bridge with a chlorophenyl head and branched isobutyric-acid ether tail; the solvent as a small branched primary alcohol (isoamyl skeleton, $\mathtt{Solvent\_TPSA} = 20.23$). The exported evidence is dominated by thermodynamic and solute-side terms:

- **Group contributions:** Thermo $+0.2922$ (46.6% of total signal); Solute $-0.2736$; Solvent $-0.0594$; Interaction $+0.0024$. No unusual feature values were flagged.
- **Positive per-feature contributors:** temperature ($\phi = +0.1506$), inverse temperature ($\phi = +0.1427$), and solute lipophilicity ($\mathtt{Solute\_MolLogP} = 3.5545$, $\phi = +0.1254$), followed by smaller positive terms from $\mathtt{Solute\_PEOE\_VSA9}$ ($+0.0366$), $\mathtt{Solute\_abraham\_V}$ ($+0.0279$), and $\mathtt{Solute\_fr\_Al\_COO}$ ($+0.0208$).
- **Negative per-feature contributors:** topological complexity ($\mathtt{Solute\_BertzCT} = 739.02$, $\phi = -0.1845$), $\mathtt{Solute\_TPSA} = 75.63$ ($\phi = -0.0878$), and smaller penalties from $\mathtt{Solute\_BalabanJ}$ ($-0.0664$), $\mathtt{Solvent\_PEOE\_VSA8}$ ($-0.0505$), and $\mathtt{Solute\_NumHDonors}$ ($-0.0219$).
- **Cross-attention (Stage 2 input):** the top rotation is $\mathtt{NumRotatableBonds} \rightarrow \mathtt{TPSA}$ ($A = 0.5876$), followed by $\mathtt{MaxPartialCharge} \rightarrow \mathtt{HeavyAtomCount}$ (0.2852), $\mathtt{Acidic} \rightarrow \mathtt{abraham\_S}$ (0.2611), $\mathtt{abraham\_E} \rightarrow \mathtt{abraham\_B}$ (0.2574), and $\mathtt{abraham\_B} \rightarrow \mathtt{abraham\_S}$ (0.2458).
- **Stage 5 audit ($\mathtt{supported}$):** the validator returned $\mathtt{needs\_revision = false}$ with no unsupported claims and no generic phrasings flagged for removal; Stage 5b revision was skipped and the Stage 4 integrated explanation passed directly to condensation.

- **Stage 6 condensation (verbatim output):** four dense paragraphs reporting moderate $\log S = -1.0426$ at $323.00$ K; thermodynamic ($+0.2922$) vs. solute-side ($-0.2736$) competition; dominant attention on solute flexibility vs. solvent polar surface area ($0.5876$) and solute acidity vs. dipolarity ($0.2611$); a *solute-limited* dissolution regime; and moderate confidence (mean leaf $165.9$, std $71.5$ over $3{,}000$ trees) with conflicting lipophilicity vs. complexity/polarity signals within the solute (full text in App. K.6).

## I.5. Literature Validation

No direct measurement of bezafibrate in 3-methylbutan-1-ol exists, but proxy alcohols bracket the prediction:

- Aqueous solubility is very low ($1.55$ $\mu$g/mL at $298$ K; $34.3$ mg/L at $310$ K).
- In neat ethanol $\sim 3$ mg/mL at room temperature; in methanol / ethanol / $n$-propanol / isopropanol the mole-fraction solubility lies in $10^{-2}$–$10^{-1}$ at $283$–$323$ K and increases with $T$ via van 't Hoff behaviour (Liu et al., 2020). Isopropanol – the closest branched proxy – yields values comparable to $n$-propanol, supporting $\sim 10$–$100$ mg/mL at $323$ K.
- Predictive PLS/MLR baselines that combine $T_m$, LogP and TPSA report RMSE $\sim 0.5$–$1$ log units on lipid solvents; cocrystals, inclusion complexes (HP-$\beta$-CD, L-proline) and PVA nanosuspensions (Zhu et al., 2024) report $1$–$10\times$ enhancement, consistent with moderate intrinsic solubility in branched alcohols.

Against the held-out label ($\log S = -1.1366$), the prediction error of $0.094$ log units falls inside the moderate-solubility band implied by these proxies.

## I.6. Discussion

The prediction is qualitatively consistent with proxy data: the thermodynamic group ($+0.2922$, $46.6\%$ of total signal) and solute lipophilicity ($+0.1254$) push $\log S$ upward, while solute-side penalties from `BertzCT` ($-0.1845$) and `TPSA` ($-0.0878$) partially offset those gains. Three caveats follow directly from the explainer trace. First, the exported `pred_Tm` ($300.65$ K) sits $\sim 158$ K below the experimental melting point ($\sim 459$ K), so the combined positive thermal SHAP from `T` ($+0.1506$) and `T_inv` ($+0.1427$) should be read as the model's internal temperature attribution rather than as independent thermodynamic validation. Second, with Gemini 3.1 Pro the validator returned `supported` without triggering Stage 5b, meaning the Stage 4 narrative—including cross-attention couplings read as solute-flexibility vs. solvent-polarity ($0.5876$) and solute-acidity vs. dipolarity ($0.2611$)—passed unchanged into condensation. Third, tree-path variability (std $71.5$ over $3{,}000$ trees, mean leaf $165.9$) shows that the moderate-solubility verdict is reached through diverse split sequences rather than a single canonical rule. All three observations are surfaced by the model's own evidence channels—SHAP attributions, group aggregates, cross-attention weights, tree statistics, and the Stage 5/6 confidence blocks (App. K.7).

## I.7. Conclusion

The case validates DISSOLVR's moderate-solubility prediction for bezafibrate in 3-methylbutan-1-ol at $323$ K ($\log S = -1.0426$ vs. label $-1.1366$; absolute error $0.094$) against literature proxies, and demonstrates that the explainer pipeline (i) identifies the dominant thermodynamic and solute-side drivers with explicit SHAP and group aggregates, (ii) validates the Stage 4 narrative in a single pass (`supported`; Stage 5b skipped), and (iii) condenses it into a four-paragraph Stage 6 summary that flags *solute-limited* dissolution and mixed tree-path confidence. The same machinery and verbatim prompt I/O (Gemini 3.1 Pro) are reproduced in App. K.7.

# J. LLM Explanation Pipeline: Technical Specification

This appendix provides complete technical details of the multi-stage LLM explanation pipeline, including mathematical formulations, full prompt templates, and implementation specifications.

## J.1. Symbolic Extraction: Mathematical Formulation

### J.1.1. SHAP FEATURE ATTRIBUTION

For each test prediction $\hat{\Sigma}_i$, we compute Shapley values for all $380$ features ($\phi_j(i)$ for the $j_{th}$ feature, using the `get_feature_importance` function of the CatBoost model, which calculates SHAP feature attribution using the method discussed in Lundberg et al. (2020) This quantifies each feature's marginal contribution to the deviation from the baseline prediction:

$$\hat{\Sigma}_i = \mathbb{E}[\hat{\Sigma}] + \sum_{j=1}^{380} \phi_j(i) \tag{33}$$

We aggregate attributions into semantic groups: (1) Solute Descriptors, (2) Solvent Descriptors, (3) Interaction Terms, (4) Thermodynamic State, to identify which class of physics dominates each prediction.

### J.1.2. INTERACTION LAYER COEFFICIENTS

From the Interaction Layer (Section 3.2), we extract the interaction matrix from each head $\mathbf{A} \in \mathbb{R}^{24 \times 24}$:

$$A_{ij} = \mathrm{softmax}\left(\frac{\mathbf{Q}_i \mathbf{K}_j^\top}{\sqrt{d_k}}\right) \tag{34}$$

We identify the top-$k$ highest-magnitude interactions:

$$\mathcal{I}_{\mathrm{top}} = \mathrm{argsort}_{(i,j)} \left(|A_{ij}|\right) [:k] \tag{35}$$

These reveal which specific solute-solvent feature pairs drove the interaction prediction (e.g., "Solute H-Bond Donors attend strongly to Solvent Carbonyl Acceptors").

### J.1.3. DECISION PATH SIGNATURE

CatBoost exposes the leaf index traversed by each sample in every tree. For sample $i$, we extract:

$$\mathbf{L}_i = [\ell_1^{(i)}, \ell_2^{(i)}, \ldots, \ell_T^{(i)}] \in \mathbb{Z}^T \tag{36}$$

where $\ell_t^{(i)}$ is the terminal leaf in tree $t$. This acts as a fingerprint of decision logic: samples with similar leaf paths have traversed similar feature splits and share analogous chemical reasoning.

### J.2. Multi-Stage LLM Reasoning Protocol

#### J.2.1. ARCHITECTURE OVERVIEW

The explanation pipeline employs a **progressive refinement strategy** with six stages:

1. **Stage 1 (Molecular Characterization)**: Generate unbiased structural descriptions from SMILES
2. **Stage 2 (Evidence Synthesis)**: Analyze SHAP values, identify patterns, detect anomalies
3. **Stage 3 (Decision Logic Analysis)**: Interpret decision splits and coupling effects from the Interaction Layer
4. **Stage 4 (Mechanistic Integration)**: Synthesize all evidence into grounded explanation
5. **Stage 5 (Validation)**: Cross-references the integrated explanation against the model evidence supplied in earlier stages, flags unsupported claims, and rewrites the narrative using only supported evidence.
6. **Stage 6 (Condensation)**: Condenses the detailed solubility explanation produced above into a concise, information-dense summary.

**Critical Design Principle**: The LLM is **blinded to prediction quality** (true $\log S$ values and absolute errors) until **Stage 3**. Stages 1–2 receive only structural inputs and SHAP evidence; the predicted $\log S$ is first revealed at Stage 3 alongside the decision-path and interaction-matrix evidence. This staged disclosure reduces confirmation bias: the explainer must first summarize the available model evidence before it can rationalize the final numerical prediction.

**SMILES handling**: Stage 1 uses SMILES only to elicit a coarse molecular description; it is not treated as the sole structural authority. Stages 2–5 then ground the explanation in deterministic RDKit descriptors, SHAP attributions, decision paths, and interaction matrices. Thus a possible SMILES-parsing error in Stage 1 can be overridden by model evidence in later stages, and the Stage 1 component can be swapped for a chemistry-specialized parser without changing the rest of the pipeline.

Full system and user-prompt templates for all six stages, together with a worked end-to-end walkthrough, are provided in App. K.

## J.3. Enhanced SHAP Analysis

The evidence synthesis stage performs pattern recognition on SHAP values

**Anomaly Detection:** For each feature $f_j$ with value $x_j$, flag as unusual if:

$$x_j \notin [\mu_j - 2\sigma_j, \mu_j + 2\sigma_j] \tag{37}$$

where $\mu_j$ and $\sigma_j$ are computed from the training distribution.

## J.4. Sample Selection Protocol

To validate the explainer across the full accuracy spectrum, we select samples using a stratified error-based approach:

- **Good Predictions**: $|y_{\text{true}} - y_{\text{pred}}| < 0.30$ (8 samples with lowest error)
- **Bad Predictions**: $|y_{\text{true}} - y_{\text{pred}}| > 1.00$ (7 samples with highest error)

This ensures the explainer is evaluated on both successful and failed predictions, testing whether it remains faithful to model reasoning even when the model is wrong.

## J.5. Implementation Details

### API Configuration:

- Model: `gemini-3.1-pro-preview` (Google Generative Language API).
- Temperature schedule: 0.2 (Molecule Description), 0.3 (Evidence analysis & Decision analysis), 0.4 (Integration), 0.1 (Validation), 0.2 (Revision, if needed), 0.2 (Condensation).
- Max tokens: 8192 per stage
- Rate limiting: 1-second delay between calls
- Key rotation: Automatic failover across multiple API keys on quota exhaustion

### Computational Cost:

- Processing time: $\sim$24 seconds per sample (8 sequential LLM calls).
- Throughput: $\sim$150 molecules/hour on commodity hardware.
- Scalability: parallelisable across samples; supports batch processing.

# K. Full LLM Prompt Templates and Worked Walkthrough

This appendix provides the prompt templates for every stage of the six-stage explainer (Sec. 4, App. J), the exact categories of evidence supplied at each stage, and an end-to-end evidence-audited walkthrough on the bezafibrate / 3-methylbutan-1-ol pair at 323 K (App. I). The prompts shown below are reproduced *byte-for-byte* from the production prompt bank (`prompts.json`, eight keys: `molecule_description`, `evidence_summary`, `decision_analysis`, `integration`, `validation`, `revision`, `condensation`, `agreement`); the worked example reports the actual inputs assembled by the explainer pipeline and the actual outputs returned by `gemini-3.1-pro` for that sample (test row 715 in the held-out set: true $\log S = -1.1366$, predicted $\log S = -1.0426$, absolute error 0.094). All stages use `gemini-3.1-pro` with the temperature schedule $(0.2, 0.2, 0.3, 0.3, 0.4, 0.1, 0.2, 0.2)$ across (solute description, solvent description, evidence analysis, decision analysis, integration, validation, revision, condensation) and an 8,192-token cap per call. Placeholders are written as `{name}`; everything outside braces is sent verbatim.

## K.1. Stage 1 – Molecular Characterisation

**Inputs supplied.** Solute SMILES and solvent SMILES only. No descriptors, no SHAP values, no model output. The stage is invoked twice – once for the solute and once for the solvent – so that the downstream stages receive structural priors

uncontaminated by solubility expectations.

**System prompt.**

```
You are a chemist describing molecular structures based on SMILES.

Focus on:
- Core structure (rings, chains, branching)
- Key functional groups and their positions
- Polarity, hydrogen bonding capacity, size

Be direct and specific.  No solubility predictions.
```

**User prompt template.**

```
Describe the following molecule based ONLY on its SMILES structure.

SMILES: {smiles}

Cover structure, functional groups, and physicochemical character
in 3-4 sentences.  Be specific about what makes this molecule
distinctive.
```

**Worked example – solute call.   Input.** `smiles =` `CC(C)(Oc1ccc(CCNC(=O)c2ccc(Cl)cc2)cc1)C(=O)O` (bezafibrate).

**Output.**

*The molecule features two para-substituted benzene rings connected by an amidoethyl linker, terminating in a chlorophenyl group on one end and a branched isobutyric acid ether on the other. Key functional groups include a terminal ionizable carboxylic acid, a central secondary amide, an ether linkage, and an electron-withdrawing aryl chloride. This architecture creates a distinct amphiphilic character, pairing a highly polar, hydrogen-bonding headgroup with a bulky, lipophilic halogenated aromatic tail. The combination of rigid aromatic rings and flexible aliphatic linkers allows this moderately large molecule to adopt extended conformations while participating in multiple hydrogen bond donor and acceptor interactions.*

**Worked example – solvent call.   Input.** `smiles = CC(C)CCO` (3-methylbutan-1-ol).

**Output.**

*The molecule is a small, branched aliphatic alcohol consisting of a four-carbon backbone with a primary hydroxyl group at C1 and a methyl branch at C3 (forming an isopentyl or isoamyl skeleton). The terminal hydroxyl group acts as both a strong hydrogen bond donor and acceptor, conferring localized polarity to the head of the molecule. In contrast, the branched hydrocarbon tail provides a nonpolar, sterically bulky region. This clear separation between a highly accessible polar head and a branched hydrophobic tail gives the molecule a distinctively asymmetric, amphiphilic character.*

### K.2. Stage 2 – Evidence Analysis

**Inputs supplied.**   The top-20 per-feature SHAP attributions $\phi_j$ (Eq. 33) split into positive and negative contributions, the four group-level aggregates (Solute, Solvent, Interaction, Thermodynamic) with the dominant group flagged, the top-15 structural descriptor values, and a list of any values that fall outside the training $\pm 2\sigma$ band. The predicted $\log S$ is **not** revealed at this stage.

**System prompt.**

```
You are interpreting machine learning model data.

Your job:  Find patterns and signal in the numbers, not just list them.

Key skills:
- Identify which feature groups dominate (positive vs negative)
- Spot unusual values or contradictions
- Compare solute vs solvent contributions
- Note when features work together or oppose each other

Be analytical, not just descriptive.
```

**User prompt template.**

```
Analyze the model evidence for this solute-solvent pair.

=== SHAP Feature Contributions (Top 20) ===
{shap_features}

=== Group Contributions ===
{group_contributions}

=== Structural Feature Values ===
{structural_features}

Identify:
1.  What's driving the prediction?  (cite top 3-5 features with values)
2.  Are there competing effects?  (positive vs negative contributions)
3.  Does the solute or solvent dominate, or is it balanced?
4.  Any unusual feature values that stand out?

Write 4-6 sentences of analysis.  Focus on insights, not lists.
```

**Worked example – inputs assembled by the pipeline.**

```
=== SHAP Feature Contributions (Top 20) ===
POSITIVE CONTRIBUTIONS (increasing solubility):
T: +0.1506
T_inv:  +0.1427
Solute_MolLogP: +0.1254
Solute_PEOE_VSA9:  +0.0366
Solute_abraham_V: +0.0279
Solute_PEOE_VSA3:  +0.0219
Solute_fr_Al_COO: +0.0208
Solute_NumHeterocycles:  +0.0185

NEGATIVE CONTRIBUTIONS (decreasing solubility):
Solute_BertzCT: -0.1845
Solute_TPSA: -0.0878
Solute_BalabanJ: -0.0664
Solute_PEOE_VSA7:  -0.0613
Solvent_PEOE_VSA8:  -0.0505
Solute_fr_NH1:  -0.0488
Solute_PEOE_VSA8:  -0.0326
Solute_abraham_E: -0.0243
Solute_abraham_S: -0.0223
Solute_NumHDonors:  -0.0219

=== Group Contributions ===
Solute:  -0.2736
Solvent:  -0.0594
Interact:  +0.0024
Thermo:  +0.2922

Dominant group:  Thermo (46.6% of total signal)

=== Structural Feature Values ===
Solute_BertzCT: 739.0243
Solute_MolWt:  361.8250
T: 323.0000
pred_Tm:  300.6500
Solute_LabuteASA: 151.1270
Solvent_MolWt:  88.1500
Solute_TPSA: 75.6300
Solute_HeavyAtomCount:  25.0000
Solvent_TPSA: 20.2300
Solute_NumRotatableBonds:  7.0000
Solute_MolLogP: 3.5545
T_inv:  3.0960
Solute_NumHAcceptors:  3.0000
```

```
Solute_HallKierAlpha:  -2.5300
Solute_NumHDonors:  2.0000

UNUSUAL VALUES:
No unusual values detected
```

**Worked example – output.**

*The prediction is defined by a strong tug-of-war between thermodynamic conditions and solute complexity. Elevated temperature (T = 323 K) acts as the primary driver for increasing solubility, with thermodynamic features contributing +0.2922 overall. However, this thermal advantage is heavily offset by the solute's structural properties (-0.2736), specifically its high complexity (BertzCT = 739.02) and polar surface area (TPSA = 75.63), which strongly hinder dissolution. Interestingly, there is a competing effect within the solute itself, as its relatively high lipophilicity (MolLogP = 3.55) provides a strong positive contribution (+0.1254) that fights against its own negative structural features. The solvent plays a minimal role in this system, leaving the solubility to be determined almost entirely by the balance between the applied heat and the solute's intrinsic properties.*

### K.3. Stage 3 – Decision Logic (prediction revealed)

**Inputs supplied.** The predicted $\log S$ and temperature, the top-10 cross-attention rotations from the $24 \times 24$ Interaction-Layer matrix $A_{ij}$ (Eq. 34), summary statistics over the CatBoost per-tree leaf signature $\mathbf{L}_i$ (Eq. 36; mean and standard deviation over the 3,000 trees), and the top-10 key decision features with sign and magnitude.

**System prompt.**

```
You are a machine learning interpretability expert analyzing
decision trees.

Look for:
- Which feature types the model prioritized
- Patterns in the cross-attention (which properties interact)
- Consistency or conflicts in the decision path

This is about understanding model behavior through the data.
```

**User prompt template.**

```
Analyze the model's reasoning for this prediction.

=== Prediction Info ===
Predicted LogS: {y_pred:.4f}
Temperature (K): {temperature:.2f}

=== Cross-Attention Weights (Solute→Solvent Interactions) ===
The transformer uses cross-attention from solute features (query) to solvent
features (key/value).
These weights show which solvent properties the model attends to for each solute
property.

**Top Cross-Attention Interactions:**
{cross_attention_summary}

=== Leaf Path Statistics ===
Number of trees:  {num_trees}
Path variability:  {path_stats}

=== Key Decision Features ===
{top_features}

Write 4-5 sentences of analysis addressing:
- Which feature interactions received most attention?  What does that suggest about
the compatibility?
- What pattern of features led to this prediction value?
- Is the decision path consistent or mixed?

Be specific about attention weights and feature contributions.
If citing any value, specify the feature / molecule whose value you are citing.
```

**Worked example – inputs assembled by the pipeline.**

```
=== Prediction Info ===
Predicted LogS: -1.0426
Temperature (K): 323.00

=== Cross-Attention Weights (Solute->Solvent Interactions) ===
**Top Cross-Attention Interactions:**
NumRotatableBonds -> TPSA: 0.5876
MaxPartialCharge -> HeavyAtomCount:  0.2852
Acidic -> abraham_S: 0.2611
abraham_E -> abraham_B: 0.2574
abraham_B -> abraham_S: 0.2458
TPSA -> abraham_B: 0.2396
NumAromaticRings -> abraham_B: 0.2140
MolWt -> pred_Tm:  0.1994
pred_Tm -> TPSA: 0.1824
abraham_A -> pred_Tm:  0.1767

=== Leaf Path Statistics ===
Number of trees:  3000
Path variability:  Mean leaf:  165.9, Std:  71.5

=== Key Decision Features ===
Solute_BertzCT: -0.1845 (negative contribution)
T: +0.1506 (positive contribution)
T_inv:  +0.1427 (positive contribution)
Solute_MolLogP: +0.1254 (positive contribution)
Solute_TPSA: -0.0878 (negative contribution)
Solute_BalabanJ: -0.0664 (negative contribution)
Solute_PEOE_VSA7:  -0.0613 (negative contribution)
Solvent_PEOE_VSA8:  -0.0505 (negative contribution)
Solute_fr_NH1:  -0.0488 (negative contribution)
Solute_PEOE_VSA9:  +0.0366 (positive contribution)
```

**Worked example – output.**

*The model's cross-attention heavily prioritizes the interaction between the solute's flexibility and the solvent's polarity, highlighted by the dominant `NumRotatableBonds` to `TPSA` weight (0.5876), alongside attention from solute acidity to solvent dipolarity (`Acidic` to `abraham\{}_S`: 0.2611). This suggests the model is primarily evaluating compatibility based on how well the solvent's polar surface area and dipolarity can accommodate the conformational changes and acidic nature of the solute. The final prediction of -1.0426 results from a competing pattern of features where thermodynamic and hydrophobic factors increase solubility, while structural complexity decreases it. Specifically, the elevated temperature (`T`: +0.1506, `T\{}_inv`: +0.1427) and solute lipophilicity (`Solute\{}_MolLogP`: +0.1254) strongly drive the prediction upward, but are counteracted by penalties from solute complexity and polarity (`Solute\{}_BertzCT`: -0.1845, `Solute\{}_TPSA`: -0.0878). Consequently, the decision path is relatively mixed, as reflected by the moderate path variability (Std: 71.5 against a Mean leaf of 165.9), illustrating a balancing act between favorable thermal conditions and restrictive solute structural properties.*

### K.4. Stage 4 – Mechanistic Integration

**Inputs supplied.** The Stage 1 solute and solvent descriptions, the Stage 2 evidence summary, the Stage 3 decision analysis, the predicted $\log S$, and the temperature.

**System prompt.**

```
You generate model explanations grounded in evidence.

Core rules:
1.  Every claim must cite specific values from the provided data.
Also specify the feature / molecule whose value you are citing.
2.  Synthesize insights - don't just repeat what's already stated
3.  Identify mechanistic patterns
(e.g., "high TPSA + low HBA indicates...")
```

```
4.  Note uncertainties or conflicts in the evidence
5.  Be concise - eliminate filler words
6.  IMPORTANT: Do NOT use raw feature names like "Solute_MACCS_105",
"Solvent_Morgan_283", "Interact_MolLogP", "BertzCT", etc.  in your
explanation.  Instead, translate them to chemical concepts (e.g.,
"structural fingerprint features", "topological complexity",
"hydrogen bonding capacity", "molecular interaction patterns").
Say "the model's analysis suggests..." rather than
"Solute_MACCS_105 indicates..."
7.  Keep these boundaries in mind while explaining the prediction:
Very highly soluble :  LogS >= 0.0
Highly soluble :  0.0 > LogS >= -1.0
Moderately soluble :  -1.0 > LogS >= -2.5
Poorly soluble :  -2.5 > LogS >= -4.0
Highly insoluble :  LogS < -4.0

Your goal:  Explain WHY the model predicted this value based on the
evidence, in plain chemical language.
```

**User prompt template.**

```
Generate a comprehensive explanation for this solubility prediction.

=== Molecule Descriptions ===
**Solute ({solute_smiles}):**
{solute_description}

**Solvent ({solvent_smiles}):**
{solvent_description}

=== Evidence Summary (from Stage 1) ===
{evidence_summary}

=== Decision Analysis (from Stage 2) ===
{decision_analysis}

=== Prediction ===
- Predicted LogS: {y_pred:.4f}
- Temperature:  {temperature:.2f} K

Structure your explanation as:

## Prediction & Key Drivers
State the predicted value.  Identify the 2-3 dominant factors with
specific values.

## Solute-Solvent Compatibility
Based on cross-attention and interaction features, what compatibility
or mismatch drives the result?  Cite specific feature pairs and their
contributions.

## Mechanistic Interpretation
What does the pattern of contributions suggest about the dissolution
process?  Connect molecular properties to the prediction.

## Confidence & Caveats
Based on evidence consistency and magnitude, how reliable is this
explanation?  Note any conflicting signals or unusual patterns.

--

IMPORTANT FORMATTING RULES:
- Do NOT use raw feature names (e.g., Solute_MACCS_105, BertzCT,
Interact_MolLogP). Translate them to plain chemical language.
- Write in plain text format, avoiding markdown headers or
formatting.
- Keep each section to 2-4 sentences.  Cite specific values.  Focus
on insights that connect the dots.
```

**Worked example – output.**

> ***Prediction \{}& Key Drivers.*** *The model predicts a moderately soluble LogS of -1.0426 at an elevated temperature of 323.00 K. The prediction is driven by a strong tug-of-war between favorable thermodynamic conditions, which increase the solubility score by +0.2922, and restrictive solute structural properties, which decrease it by -0.2736. Specifically, the solute's high topological complexity (739.02) and polar surface area (75.63) hinder dissolution, while its intrinsic lipophilicity (3.55) and the applied heat strongly promote it.*

> ***Solute-Solvent Compatibility.*** *The model's attention mechanism heavily prioritizes the interplay between the solute's structural flexibility and the solvent's polar surface area, assigning this interaction a dominant weight of 0.5876. Additionally, the compatibility between the solute's acidic carboxylic acid group and the solvent's dipolarity is a significant focus, with an attention weight of 0.2611. Despite these specific interaction pathways, the solvent's overall contribution to the prediction is minimal, indicating that the isoamyl alcohol's amphiphilic nature accommodates the solute but does not actively drive the solubility outcome compared to intrinsic solute properties.*

> ***Mechanistic Interpretation.*** *The dissolution process is characterized by competing hydrophobic and structural forces. The solute's bulky, rigid aromatic rings and high complexity impose a significant energetic penalty (-0.1845) for cavity formation in the solvent. However, the solute's substantial lipophilicity (+0.1254) interacts favorably with the solvent's branched hydrocarbon tail, while the elevated temperature (+0.1506) provides the necessary kinetic energy to overcome the cohesive forces of the solute's hydrogen-bonding networks, which otherwise apply a polar surface area penalty of -0.0878.*

> ***Confidence \{}& Caveats.*** *The prediction is moderately confident, reflecting a mixed decision path with moderate variability, shown by a standard deviation of 71.5 against a mean leaf value of 165.9. This variability stems from conflicting signals within the solute itself, where its lipophilic character promotes solubility while its structural complexity and polarity strongly resist it. The near-total reliance on temperature and solute properties, rather than specific solvent interactions, suggests the model views this system primarily through a thermodynamic and structural lens rather than relying on strong chemical synergies.*

## K.5. Stage 5 – Validation & Correction

**Inputs supplied.** The Stage 4 explanation, both molecule descriptions, the top-20 SHAP contributions, the list of unusual features, the top-10 key decision features, the group contributions, the top-10 cross-attention interactions, the structural feature table, the tree statistics, the predicted $\log S$ and the temperature. No new external information.

**System prompt.**

```
You are a strict validator for chemistry explanations grounded in model evidence.
Your job is not only factual checking, but also preventing generic chemistry prose
that could sound plausible even when the grounding is weak or uninformative.
Your task:  Verify that every claim in the explanation is grounded in the provided
model evidence.

Check for:
1.  **Structural accuracy**:  Do the molecule descriptions used in the explanation
matches the actual SMILES?
2.  **Numerical accuracy**:  Do cited values match the source data exactly?
4.  **Magnitude claims**:  Are statements about "dominant", "major", "minor"
contributions accurate?
5.  **Logical consistency**:  Do the conclusions follow from the evidence?
6.  **Chemical Plausibility**:  Does the mechanistic explanation violate
fundamental laws of chemistry or thermodynamics?  (e.g., claiming a highly
polar group inherently decreases solubility in water without a valid steric or
electronic reason).
7.  Ensure that there are no unbacked claims in the explanation.

You must reject explanations that:

- rely on broad textbook solubility statements without tying them to the supplied
evidence
- could have been written for many molecules with only minor wording changes
- mention polarity, hydrogen bonding, aromaticity, size, hydrophobicity, or
rigidity in a generic way without connecting them to specific supplied features
- smooth over the actual model evidence into a vague narrative
- violate fundamental laws of chemistry or thermodynamics (e.g., claiming a highly
polar group inherently decreases solubility in water without a valid steric or
electronic reason)

You should prefer explanations that :
```

– explicitly track the supplied SHAP features, grouped contributions,
cross-attention interactions, unusual values, and decision evidence
– make claims that would materially change if the supplied evidence changed
– are specific enough that an explanation from an untrained grounding model would
differ substantially in content
– describe the data exactly as it was provided to you
– ensure there are no unbacked claims in the explanation

These are the boundaries defined by us:
Very highly soluble :  LogS >= 0.0
Highly soluble :  0.0 > LogS >= -1.0
Moderately soluble :  -1.0 > LogS >= -2.5
Poorly soluble :  -2.5 > LogS >= -4.0
Highly insoluble :  LogS < -4.0

Flag as hallucination:
– Descriptions that contradict the SMILES structure (e.g.  claiming a ring in a
linear chain)
– Invented feature values or contributions
– Features mentioned that aren't in top 20 SHAP values
– Incorrect percentages or group contributions
– Unsupported mechanistic claims
– Mechanistic claims that violate fundamental laws of chemistry or thermodynamics,
EVEN IF the SHAP data implies it.  If the model evidence forces an absurd chemical
conclusion, you must flag the explanation as physically ungrounded.
– Any claim that is not backed by the evidence & comes from internal knowledge of
the LLM. We want all claims to be a natural consequence of input data.

Output JSON format:
{
"verdict":  "supported" | "partially_supported" | "unsupported",
"unsupported_claims":  ["..."],
"supported_points":  ["..."],
"generic_claims_to_remove":  ["..."],
"evidence_specific_points_to_keep":  ["..."],
"correction_instructions":  ["..."],
"needs_revision":  true
}

Be strict but fair.  Minor rounding differences are acceptable.  Focus on factual
grounding.
IMPORTANT: Keep the `corrected_explanation` concise (approx 4 paragraphs).  Do not
unnecessarily expand it.
Set "needs_revision" to false only if the explanation is both well grounded and
clearly discriminative with respect to the supplied evidence.

## User prompt template.

Validate the following explanation against the source model evidence.

=== EXPLANATION TO VALIDATE ===
{explanation}

=== SOURCE EVIDENCE ===

**Molecule Information:**
– Solute SMILES: {solute_smiles}
(Description used in explanation:  "{solute_description}")
– Solvent SMILES: {solvent_smiles}
(Description used in explanation:  "{solvent_description}")

**Top 20 SHAP Contributions:**
{shap_features}

**Unusual Features:**
{unusual_features}

**Key Decision Features (from Decision Analysis):**
{decision_features}

```
**Group Contributions:**
{group_contributions}

**Cross-Attention Summary (Top 10 Interactions):**
{cross_attention_summary}

**Structural Features:**
{structural_features}

**Tree Statistics:**
{tree_stats}

**Prediction:**
- Predicted LogS: {y_pred:.4f}
- Temperature:  {temperature:.2f} K

=== VALIDATION TASK ===

Check every numerical claim, feature reference, and mechanistic statement in the
explanation.
Verify that the qualitative descriptions match the actual chemical structure
defined by the SMILES.

For each issue found:
1.  Quote the problematic statement
2.  Explain why it's incorrect or unsupported (or contradicts the SMILES)
3.  Provide the correct information from source data

Validation objective:
- Keep only explanation content that is strongly anchored to the supplied evidence
- Favor explanation wording that is evidence-specific, sparse, and discriminative
yet still explains the solubility
- Frame arguments in the format:  [Data given by the user] followed by [what it
implies in terms of solubility]
- Remove broad textbook statements about polarity/H-bonding if they don't
explicitly cite the SHAP feature or Interaction node providing that signal

If issues are found, you MUST provide a "corrected_explanation" in the JSON that:
- Retains the structure and tone of the original
- Fixes all factual errors using source data
- Removes unsupported claims
- Takes care to NOT introduce new hallucinations
- Keep only explanation content that is strongly anchored to the supplied evidence
- Favor explanation wording that is evidence-specific, sparse, and discriminative
yet still explains the solubility.

Output your findings in JSON format as specified in the system prompt.
```

**Worked example – validator output (JSON).**

```
  "verdict": "supported", "unsupported_claims": [], "supported_points": [
"LogS prediction of -1.0426 at 323.00 K", "Thermo group contribution of +0.2922
and Solute group contribution of -0.2736", "BertzCT value of 739.02 and SHAP
of -0.1845", "TPSA value of 75.63 and SHAP of -0.0878", "MolLogP value of 3.55
and SHAP of +0.1254", "Cross-attention weights for NumRotatableBonds -> TPSA
(0.5876) and Acidic -> abraham_S (0.2611)", "Tree statistics (mean leaf 165.9,
std 71.5)", "Solvent group contribution of -0.0594" ], "generic_claims_to_remove":
[], "evidence_specific_points_to_keep": [ "The tug-of-war between thermodynamic
conditions and solute structural properties.", "The specific mention of
the solute's bulky, rigid aromatic rings and high complexity imposing an
energetic penalty.", "The interpretation of the tree statistics reflecting
a mixed decision path.", "The observation that the model relies heavily on
temperature and solute properties rather than specific solvent interactions." ],
"correction_instructions": [ "Minor adjustment: Remove the assumption that 'TPSA'
in the cross-attention specifically refers to the solvent, as the feature lacks a
prefix in the evidence." ], "needs_revision":  false
```

**Stage 5b – Revision (separate prompt in `prompts.json`).** Because `needs_revision = false`, the validator returned `supported` and Stage 5b revision was *skipped*; the Stage 4 integrated explanation passed directly to Stage 6 condensation. The revision prompt below is still defined in `prompts.json` and would run

when validation sets `needs_revision = true`. The revision stage uses its own system and user prompts from `prompts.json["revision"]` (the system prompt is identical to the integration system prompt; the user template threads the validator's `evidence_specific_points_to_keep`, `generic_claims_to_remove` and `correction_instructions` back into the LLM).

```
Revise the explanation so every claim is grounded in the supplied
evidence.
The revised explanation should be maximally evidence-specific and
should avoid generic chemistry language.
It should read differently when the grounding evidence changes.

Molecule SMILES: {solute} in {solvent}
Predicted LogS: {y_pred:.4f}

Original explanation:
{explanation}

Evidence-specific points to keep:
{evidence_points}

Generic claims to remove:
{generic_points}

Revision instructions:
{instructions}

Constraints:
- Do not use broad filler like 'this affects solubility' unless
tied to explicit evidence.
- Prefer naming concrete supplied drivers over giving a smooth
general narrative.
- If evidence is weak or noisy, say so rather than inventing a
polished explanation.

Return only the corrected explanation.
```

**Worked example – corrected explanation (Stage 5b output).**

[Stage 5b not invoked for this sample — validation returned `supported`; see Stage 4 worked-example output above.]

### K.6. Stage 6 – Condensation

**Inputs supplied.** The validated (and, if necessary, corrected) explanation from Stage 5.

**System prompt.**

```
You are an expert scientific editor.  Your task is to condense a detailed
solubility explanation into a concise, information-dense summary.

Rules:
1.  PRESERVE all numerical values (LogS, TPSA, MolLogP, temperatures, percentages,
contribution values)
2.  REMOVE redundancy, filler phrases, and repetitive statements
3.  MAINTAIN the scientific accuracy and causal reasoning
4.  Use precise, technical language appropriate for chemistry researchers
5.  Do NOT add information not present in the original
6.  Do NOT use markdown formatting – output plain text only
7.  Do NOT use raw feature names – keep the chemical language from the input
8.  End Sentences with a period.
9.  Ensure that the explanation is complete.
```

**User prompt template.**

```
Condense the following solubility explanation into exactly 4 dense
paragraphs:
```

```
=== Original Explanation ===
{original_explanation}

=== Required Structure ===

Paragraph 1 - PREDICTION SUMMARY:
State the LogS prediction and temperature.  Identify the dominant
driver category (solute/solvent/interaction) with its percentage.
Highlight the 2-3 most influential molecular properties with their
values.

Paragraph 2 - SOLUTE-SOLVENT DYNAMICS:
Describe the key cross-attention interactions and what they reveal
about compatibility.  Quantify the solvent's contribution and
explain why it helps or hinders dissolution.

Paragraph 3 - MECHANISTIC INTERPRETATION:
Explain the dissolution mechanism in chemical terms.  Identify
whether dissolution is solute-limited, solvent-limited, or
interaction-limited.  Connect specific molecular features to the
predicted outcome.

Paragraph 4 - CONFIDENCE & UNCERTAINTY:
Assess prediction reliability based on model statistics.  Note any
conflicting signals between feature groups.  Indicate if the
prediction lies within expected model behavior.

Output ONLY the 4 paragraphs with no headers, labels, or markdown.
Each paragraph should be 3-5 sentences of dense, information-rich
text.
```

**Worked example – output.**

*The model predicts a moderately soluble LogS of -1.0426 at an elevated temperature of 323.00 K. The prediction is primarily driven by favorable thermodynamic conditions increasing the score by +0.2922, which compete against restrictive solute structural properties decreasing it by -0.2736. Specifically, the solute topological complexity of 739.02 and polar surface area of 75.63 hinder dissolution, whereas an intrinsic lipophilicity of 3.55 and applied heat strongly promote solubility.*

*The model prioritizes the interplay between solute structural flexibility and solvent polar surface area, assigning this interaction a dominant attention weight of 0.5876. Additionally, compatibility between the solute acidic carboxylic acid group and solvent dipolarity receives a significant attention weight of 0.2611. Despite these specific interaction pathways, the overall solvent contribution to the prediction remains minimal. This indicates that the amphiphilic nature of isoamyl alcohol accommodates the solute but does not actively drive the solubility outcome compared to intrinsic solute properties.*

*The dissolution process is solute-limited and characterized by competing hydrophobic and structural forces. Bulky, rigid aromatic rings and high complexity impose a significant energetic penalty of -0.1845 for cavity formation within the solvent. Conversely, substantial solute lipophilicity contributes +0.1254 by interacting favorably with the branched hydrocarbon tail of the solvent. Furthermore, the elevated temperature provides a +0.1506 kinetic energy boost necessary to overcome cohesive forces within the solute hydrogen-bonding networks, which otherwise apply a polar surface area penalty of -0.0878.*

*The prediction is moderately confident, reflecting a mixed decision path with moderate variability indicated by a standard deviation of 71.5 against a mean leaf value of 165.9. This variability stems from conflicting signals within the solute itself, where lipophilic character promotes solubility while structural complexity and polarity strongly resist it. The near-total reliance on temperature and solute properties rather than specific solvent interactions suggests the model views this system primarily through a thermodynamic and structural lens. Consequently, the prediction lies within expected model behavior for systems lacking strong chemical synergies.*

### K.7. End-to-End Worked Walkthrough

We illustrate the full pipeline on the case-study pair from App. I: solute = bezafibrate (`CC(C)(Oc1ccc(CCNC(=O)c2ccc(Cl)cc2)cc1)C(=O)O`), solvent = 3-methylbutan-1-ol (`CC(C)CCO`), temperature $T = 323$ K. DISSOLVR predicts $\log S = -1.0426$ against an experimental value of $-1.1366$, giving an absolute error of $0.094$.

**(a) Model-evidence excerpt.** Table 18 summarises the dominant SHAP attributions and group-level aggregates assembled by the explainer pipeline for this sample, and Table 19 shows the corresponding top-10 entries of the cross-attention matrix $A_{ij} \in \mathbb{R}^{24 \times 24}$. Together they constitute the quantitative grounding used by Stages 2–6.

*Table 18.* Evidence excerpt for bezafibrate / 3-methylbutan-1-ol at 323 K. Positive $\phi_j$ pushes $\log S$ upward (more soluble); the row in parentheses gives the raw descriptor value where it appears in the top-15 structural list.

| Evidence item | Reported value | Role in explanation |
|---|---|---|
| Prediction | $\log S = -1.0426$ at 323 K | Moderate organic-solvent solubility |
| Group contribution – Thermo | $+0.2922$ (46.6% of total signal) | Dominant positive driver |
| Group contribution – Solute | $-0.2736$ | Dominant negative driver |
| Group contribution – Solvent | $-0.0594$ | Mild penalty |
| Group contribution – Interaction | $+0.0024$ | Negligible |
| T (Temperature) | $\phi = +0.1506$ | Thermal activation |
| T_inv ($1000/T$) | $\phi = +0.1427$ | Boltzmann-style boost |
| Solute_MolLogP | $3.55, \phi = +0.1254$ | Lipophilic compatibility with alcohol |
| Solute_BertzCT | $739.02, \phi = -0.1845$ | Lattice / cavity / complexity penalty |
| Solute_TPSA | $75.63, \phi = -0.0878$ | Polar-surface mismatch |
| Solute_BalabanJ | $\phi = -0.0664$ | Topological penalty |
| Solvent_PEOE_VSA8 | $\phi = -0.0505$ | Largest negative solvent driver |
| Solute_NumHDonors | $2.0, \phi = -0.0219$ | H-bond donor mismatch with solvent |
| pred_Tm | $300.65$ K (vs. experimental $\sim 459$ K) | Thermal benefit may be over-credited |
| Tree path statistics | 3000 trees, mean leaf 165.9, std 71.5 | Mixed / diverse decision path |

*Table 19.* Top-10 cross-attention rotations $A_{ij}$ for the walkthrough sample. Solute tokens (rows) attend to solvent tokens (columns).

| Solute token $i$ | Solvent token $j$ | $A_{ij}$ |
|---|---|---|
| NumRotatableBonds | TPSA | 0.5876 |
| MaxPartialCharge | HeavyAtomCount | 0.2852 |
| Acidic | abraham_S | 0.2611 |
| abraham_E | abraham_B | 0.2574 |
| abraham_B | abraham_S | 0.2458 |
| TPSA | abraham_B | 0.2396 |
| NumAromaticRings | abraham_B | 0.2140 |
| MolWt | pred_Tm | 0.1994 |
| pred_Tm | TPSA | 0.1824 |
| abraham_A | pred_Tm | 0.1767 |

**(b) Stage 4 narrative (post-validation).** The validator returned `supported` with `needs_revision = false`; Stage 5b was skipped. The verbatim Stage 4 integrated explanation is reproduced as the worked-example output in App. K.4; we do not duplicate it here.

**(c) Stage 6 summary.** The verbatim Stage 6 condensed output (four dense paragraphs) for this sample is reproduced as the worked-example output in App. K.6; we do not duplicate it here.

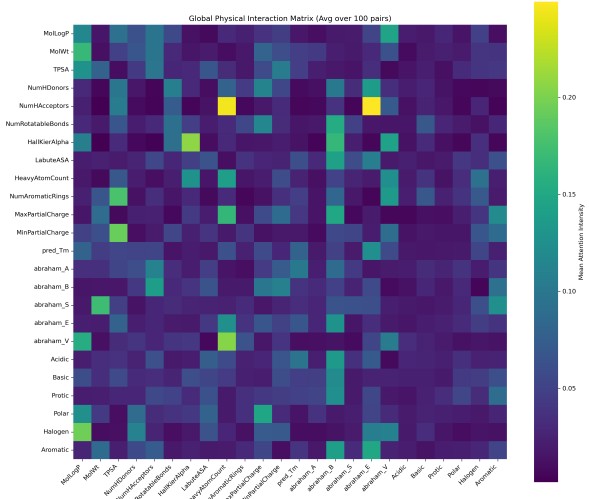
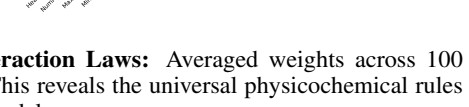

*(a)* **Global Interaction Laws:** Averaged weights across 100 random pairs. This reveals the universal physicochemical rules learned by the model.

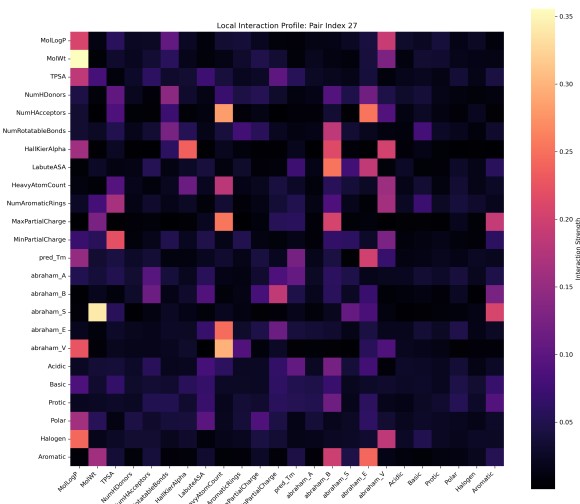

*(b)* **Case Study 1 (H-Bond Driven):** Interaction profile for Solute-Solvent Pair 27. Note the intense activation (bright spots) indicating the important features for prediction.

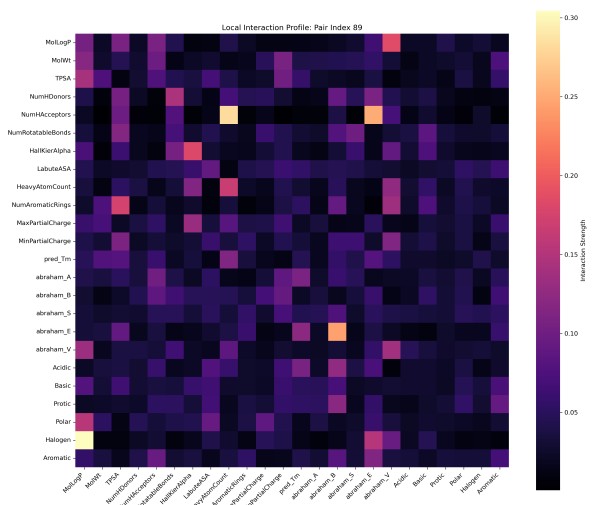

*(c)* **Case Study 2 (Mixed Mode):** Interaction profile for Pair 89. A diffuse pattern indicates a complex dissolution process in which multiple features interact.

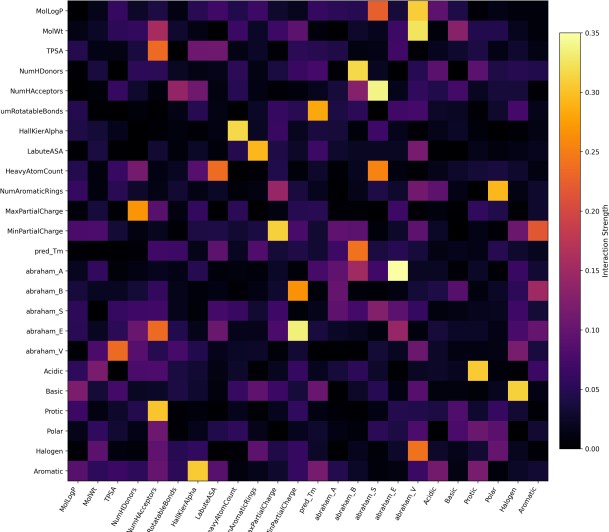

*(d)* **Case Study 3 (Mixed Mode):** Interaction profile for Pair 17. A variety of solute features successfully query the solvent indicating important features for solubility.

*Figure 3.* **Visualizing the "Chemical Intuition" of the Interaction Layer.** Panel (a) demonstrates the model's grasp of global LSER (Linear Solvation Energy Relationships). Panels (b-d) reveal the dynamic nature of the mechanism, which adapts its focus based on the specific solute-solvent context.

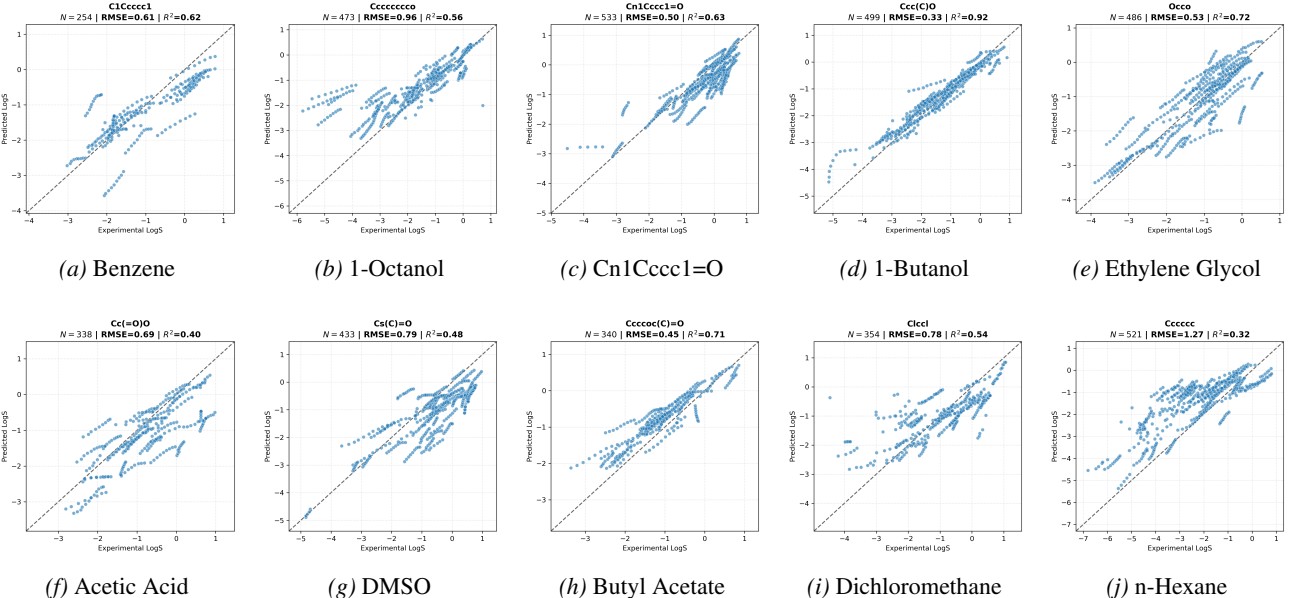

*Figure 4.* **Per-Solvent Parity Plots on BigSolDB 1.0.** These plots demonstrate the model's predictive accuracy across diverse solvent classes, ranging from non-polar alkanes to highly polar protic media. Despite being trained on only the top 19 solvents, the model maintains a tight correlation on these holdout environments.

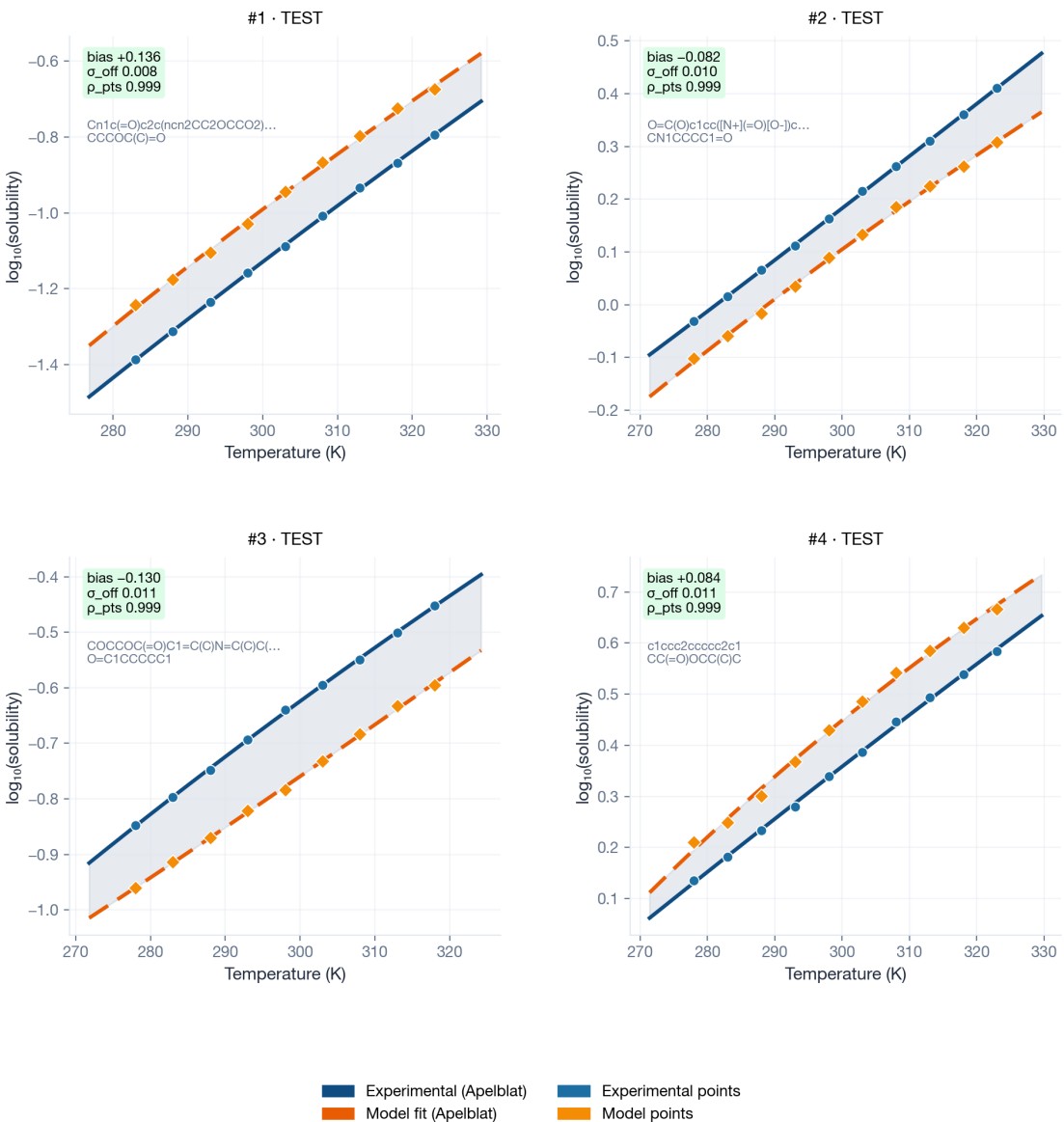

*Figure 5.* This figure shows the high Pearson correlation between the fitted apelblat curve to the experimental data against the predicted data

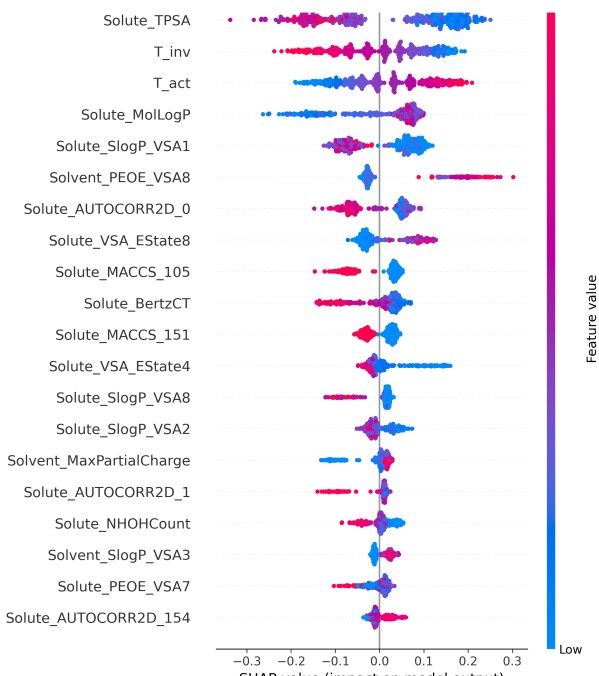

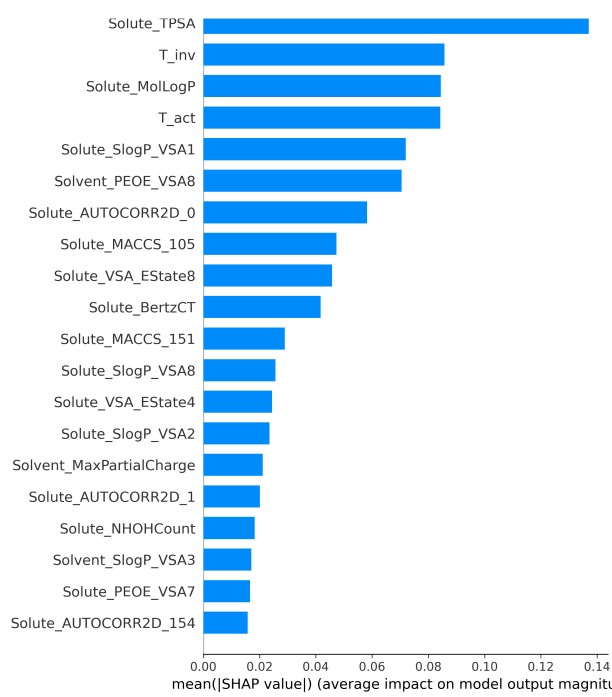

*(a)* **Thermodynamic Drivers.** `Solute_TPSA` and `T_inv` show clear monotonic impact, confirming rediscovery of lattice energy and Van't Hoff laws.

*(b)* **Hierarchical Importance.** Feature contribution drops sharply after the top thermodynamic drivers, establishing the macroscopic baseline.

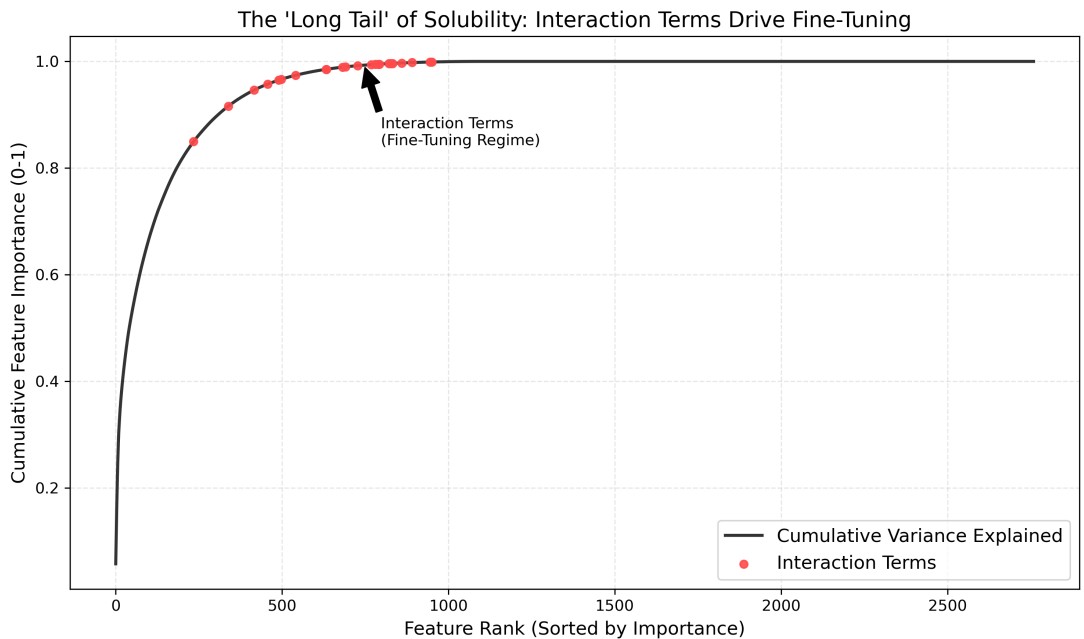

*(c)* **Fine-Tuning via Interactions.** While global descriptors (black line) explain the variance baseline, the Interaction Terms (red dots) cluster in the "shoulder" (Ranks 200–1000), providing the fine-tuning required to reach the aleatoric limit.

*Figure 6.* **Mechanistic Validation of DISSOLVR.** (a-b) The model prioritizes Tier 1 global descriptors that map to thermodynamic laws. (c) It utilizes Tier 2 interaction terms to resolve specific solute-solvent compatibilities that global descriptors cannot capture.

