# OpenReview forum: "DISSOLVR: An Interpretable and Fast Framework for Aqueous and Organic Solubility Prediction"
_ICML.cc/2026/Conference — ICML 2026 regular_

### Official Review · Reviewer_fPPx · 2026-03-06

**Soundness:** 4
**Presentation:** 4
**Significance:** 4
**Originality:** 2
**Overall Recommendation:** 4
**Confidence:** 4

**Summary:**

The paper proposes an interpretable thermodynamically informed and consistent solubility model to predict both water solubility, and multi-solvent solubility of different solutes. The model reaches accuracy close to the bound enforced by experimental nose (aleatoric limit), and show OOD generalization capabilities. The paper also proposes an LLM based pos-thoc interpretation solution.

**Compliance With Llm Reviewing Policy:**

Affirmed.

**Final Justification:**

The presented method addresses an important practical problem, sound and well presented. The machine learning novelty is limited. In the rebuttal period the authors addressed my question about novelty by concentrating on physics based modeling, interpretability by design and the resulting robustness for OOD generalization. While these are really important features, and serves the basis of my decision to tend toward acceptance, the methods used here are not particularly innovative, and they provide only minor benefit to the general ML field. It is however clearly provide benefit for the application area!

**Key Questions For Authors:**

-How would you address the problem mentioned in weaknesses, about the expert rating of “explanation quality” and possible "adversarial" explanation created by the LLM?

-How often your assumptions, for example endothermic dissolution is violated in case of organic compounds?

-In the multiple solvent case, given your model can itself take into account interactions, how sure can we be that the Interaction features are preferentially used by the CatBoost trees over creating their own? Yes I recognize that the ablation study show the I term helps, so it is clear that it is used.

-It is not entirely clear for me what are the tokens in the Interaction layer? Are they the features? Can you elaborate a bit on the semantics of the dimensions of your matrices in the attention mechanism?

**Limitations:**

In general the limitations of the method is clear for a field expert, but I would like if the authors mention the limitations of some choice, for example the endothermic dissolution model for non-chemist (machine learning expert) reader to understand. For example according to my knowledge such common salts as CaSO4 have non-monotonous solubility curve with temperature, or Ce2(SO4)3 shows exothermic solvation, and as such decreasing solubility curve.  I assume the work concentrate on organic compounds and not salts, and my chemistry knowledge is limited to tell how often these effects show up in that case (see questions).

**Strengths And Weaknesses:**

Strengths:
- The paper is very close to application, it is clear that domain experts are participated in the study. This makes it clear that the problem is relevant, the data is handled well, and the correct questions were asked.
- The work aims for interpretability and trustworthyness, also very significant and up to date research areas.
- The work is clearly presented, easy to follow and sound.

Weaknesses:
- As it is positioned now, to a mainly machine learning venue, the novelty is questionable. I am not sure the paper provide new insight in the field of machine learning.
- The expert evaluation of the interpretation is very subjective. This is a problem as we know that LLMs are fine tuned (eg. via RL) to be convincing, not necessarily accurate. Good subjective “explanation quality” may just mean a good “salesman” skill by the LLM not necessarily accurate explanation.

---

> ### Author Rebuttal · Authors · 2026-03-30
>
> We sincerely thank the reviewer for the constructive review. We are pleased the soundness, presentation, and significance were rated excellent. Below, we address each concern and the proposed revisions. With these changes, we hope the reviewer will be further convinced of the merits of our work.
>
> ---
>
> > **W2/Q1: Good subjective "explanation quality" may just mean a good "salesman" skill.**
>
> The reviewer is indeed correct that unless proper guardrails are placed, the LLM would use its own training data than to explain the model. The proposed 6-stage prompt structure (Sec 4) enforces these guardrails and is one of the key novel contributions of our work. While our human survey leads credence to the utility of the LLM explainer, to further establish that the LLM is indeed conditioned to explain the model than its own independent knowledge, we have added one more ablation study.
>
>  We created two instances of DISSOLVR: a randomly initialized model, and therefore producing menaningless SHAP value, and one trained as per normal protocol. We then generate explanations using identical LLM prompts for 30 samples from a fully trained DISSOLVR model and a randomly initialised, untrained model. If the LLM were a "salesman," both would produce equally convincing outputs.
>
> Below we measure agreement between prompt outputs using [1] on a 1–5 scale (1 = complete disagreement) across each of the 6 stages (See **[https://anonymous.4open.science/r/dissolvr/explainer/prompts.json]** for the exact prompts used). The low scores confirm that when the model has learned nothing, the explanations reflect that.
>
> |Dimension|Trained vs Untrained (n=30)|
> |-|-|
> |Conclusion Agreement|1.37 ± 0.96|
> |Reasoning Alignment|1.53 ± 0.57|
> |Causal Attribution|1.90 ± 0.76|
> |Completeness Overlap|2.53 ± 0.68|
> |Overall Agreement|1.63 ± 0.76|
> |Mean of 4 Dimensions|1.83 ± 0.42|
>
> The low scores confirm that when the model has learned nothing, the explanations reflect that.
>
> [1] Zheng, L. et al. Judging LLM-as-a-Judge with MT-Bench and Chatbot Arena. NeurIPS 2023.
>
> > **Q2: How often is endothermic dissolution violated for organic compounds?**
>
> We performed a systematic analysis across all datasets under two criteria: (i) any non-monotonic dip, and (ii) strictly decreasing solubility with temperature.
>
> |Dataset|Any Violation|Strictly Decreasing|
> |-|-|-|
> |BigSolDB 1.0|3.21%|0.45%|
> |BigSolDB 2.0|2.67%|0.51%|
> |Leeds|0.00%|0.00%|
>
> Under the stricter criterion, fewer than 0.5% of pairs exhibit genuinely exothermic behaviour, and even these may reflect inter-lab measurement noise. A source-wise analysis confirmed non-endothermic rates of 3.22% and 1.52% on BigSolDB 1.0 and 2.0 under the relaxed criterion. We acknowledge DISSOLVR will fail for truly exothermic cases and will include this as an explicit limitation in the revision.
>
> > **Q3: Are Interaction features preferentially used over CatBoost's own learned interactions?**
>
> Our ablation (Table 3) shows that removing the interaction block increases RMSE by +0.0112, the 2nd highest contributor across all ablations. Also, since GBDTs operate on axis-aligned splits, it is infeasible for them to construct their own splits.
>
> > **Q4: What are the tokens in the Interaction layer?**
>
> Each token corresponds to one of 24 macro-physical properties per molecule (e.g., MolLogP, acidity, basicity, ring count). The cross-attention mechanism computes a 24×24 matrix between solute and solvent property vectors, whose entries directly quantify feature-level solute–solvent coupling. We will clarify this explicitly in the revision.
>
> > **W1: ML Novelty**
>
> DISSOLVR's ML contribution operates on two levels: impact and technique.
>
> **Impact: Challenging the scaling assumption.** The dominant trend in molecular ML is toward larger, less interpretable models, assuming complexity is necessary to push quality. DISSOLVR directly challenges this: by matching or exceeding deep learning baselines including ChemFM-3B (see W2 of Rev. Uyir) with a fully transparent framework, we show the field may be trading interpretability for illusory gains.
>
> **Technical contributions.**
> - **Physics-constrained learning**: Lemma 3.1 formally establishes that monotonic temperature constraints restrict the hypothesis space to a closed convex cone, provably reducing noise-fitting capacity (Theorem 3.1), which a regularization principle generalizable beyond solubility.
> - **OOD generalization via explicit featurization**: DISSOLVR outperforms RILOOD, a dedicated OOD framework, across all multi-solvent benchmarks (Table 2b), demonstrating that transparency and OOD robustness are not in tension.
>
> >  **Limitations: Salts and Inorganic Systems**
>
> We will clarify in App. B.1 and the Limitations section that DISSOLVR is strictly intended for organic small molecules, which is precisely why the endothermic assumption remains physically consistent across our evaluation suite. In addition, we will also highlight the limitations outlined in our response to W6 of Rev. vZpW.

---

> > ### Author Rebuttal · Reviewer_fPPx · 2026-04-03
> >
> > Thank you for your detailed answers.

---

> > > ### Author Response · Authors · 2026-04-03
> > >
> > > Dear reviewer fPPx,
> > >
> > > We are pleased that our rebuttal has fully resolved the reviewer's concerns. Given that all raised issues have been resolved, we kindly request the reviewer to revisit their rating. We remain happy to address any outstanding questions.
> > >
> > > regards,
> > > Authors

---

### Official Review · Reviewer_MwK8 · 2026-03-11

**Soundness:** 2
**Presentation:** 3
**Significance:** 2
**Originality:** 2
**Overall Recommendation:** 3
**Confidence:** 5

**Summary:**

This paper presents the DISSOLVR framework, aiming to solve the problem of predicting the solubility of organic molecules in complex chemical environments (with multiple solvents and variable temperatures). The authors challenged the assumption that "deep learning is a necessary condition for achieving SOTA performance", and constructed a lightweight model by combining 176 physical-chemical descriptors and gradient boosting decision tree (GBDT, specifically CatBoost). Additionally, the paper introduces a six-stage explanation pipeline based on large language models (LLM), converting the mathematical output of the model into a narrative understandable by chemical experts.

**Compliance With Llm Reviewing Policy:**

Affirmed.

**Final Justification:**

I maintain my original score and final recommendation.

**Key Questions For Authors:**

W1:The methodological breakthroughs in this paper are rather limited. The accuracy is constrained by the Aleatoric Limit inherent in the experimental data, which has been elaborated in detail in [1]. Switching the underlying model from MLP (FastProp) to GBDT (DISSOLVR) is more of an architectural fine-tuning rather than a paradigm innovation.
W2:The feature set used in this paper is highly similar to that in [1]. Due to the fact that the two articles share similar input features and underlying logic, DISSOLVR lacks a significant improvement in the "physics-driven" feature compared to its predecessor.
W3:Although LLM can generate fluent chemical narratives, there is a risk of "post-hoc rationalization". When the model makes a wrong prediction, LLM may use its vast pre-trained knowledge base to fabricate seemingly reasonable chemical logic to "cover up" the model's error. This "illusion" may mislead researchers.
W4:The paper only demonstrated the superiority of GBDT in the specific task of solubility (with extremely high data noise), but in more complex chemical tasks (such as dynamic reaction simulation), this conclusion may not hold true.

Questions
Q1:The paper highlights the speed of DISSOLVR. Please provide a comparison of the end-to-end inference time between DISSOLVR and FastProp [1] in the same hardware environment to demonstrate the actual engineering benefits brought by switching to GBDT.
Q2:How does the author ensure that the explanations generated by the LLM are based on the actual predictive logic of the model, rather than merely repeating the known chemical properties in the pre-trained knowledge based solely on the input SMILES string?
Q3:In the experimental section, please specifically quantify the improvement in RMSE (Root Mean Square Error) of the model on the OOD (out-of-distribution) test set after introducing the monotonicity constraint, in order to prove the necessity of this physical constraint.



[1] Attia, L., Burns, J.W., Doyle, P.S. et al. Data-driven organic solubility prediction at the limit of aleatoric uncertainty. Nat Commun 16, 7497 (2025). https://doi.org/10.1038/s41467-025-62717-7

**Limitations:**

yes

**Strengths And Weaknesses:**

s1:Compared with models based on GNN or deep learning, DISSOLVR has significant advantages in terms of inference speed and computational resource consumption, and is suitable for the rapid screening of large-scale chemical libraries.
s2:An LLM framework was constructed that converts "decision paths + physical descriptors" into chemical explanations. Its effectiveness in enhancing user trust was verified through a double-blind experiment conducted by experts.

---

> ### Author Rebuttal · Authors · 2026-03-30
>
> We thank the reviewer for their constructive feedback and hope our responses will further convince the reviewer of our work's merits.
>
> ---
>
> >**Q1: End-to-end inference time comparison: DISSOLVR vs. FastProp.**
>
> Avg. inference times over 5 runs.
>
> | |BigSolDB 1.0|BigSolDB 2.0|Leeds|
> |-|-|-|-|
> |**FastSolv Inference**|2.169s|4.115s|0.393s|
> |**DISSOLVR Inference**|0.396s|0.539s|0.224s|
> |**Inference Speedup**|**5.5×**|**7.6×**|**1.8×**|
>
> **W3/Q2: How do you ensure that LLM explanations are based on the model's predictive logic?**
>
> Without guardrails, the LLM defaults to its pre-trained knowledge rather than explaining the model. The proposed 6-stage prompt structure (Sec 4) enforces these guardrails and is one of our key contributions. Beyond our human survey, we added an ablation to verify the LLM is conditioned on model outputs.
>
> We generated explanations for 30 samples from both a trained and randomly initialised DISSOLVR, using identical LLM prompts. If the LLM were a "salesman," both would produce equally convincing outputs.
>
> Below we measure agreement between prompt outputs using [1] on a 1–5 scale (1 = complete disagreement). See **[https://anonymous.4open.science/r/dissolvr/explainer/prompts.json]** for the exact prompts used.
>
> | |Trained vs Untrained (n=30)|
> |-|-|
> |Conclusion Agreement|1.37 ± 0.96|
> |Reasoning Alignment|1.53 ± 0.57|
> |Causal Attribution|1.90 ± 0.76|
> |Completeness Overlap|2.53 ± 0.68|
> |Overall Agreement |1.63 ± 0.76|
> |Mean of 4 Dimensions |1.83 ± 0.42|
>
> The low scores confirm that when the model has learned nothing, the explanations reflect that.
>
> [1] Zheng, L. et al. Judging LLM-as-a-Judge with MT-Bench and Chatbot Arena. NeurIPS 2023.
>
> >**Q3: RMSE Improvement from Monotonicity Constraints in OOD test sets.**
>
> We conducted an ablation study across 5 random seeds comparing DISSOLVR with and without monotonic temperature constraints:
>
> | Dataset | Constrained RMSE | Unconstrained RMSE | $\Delta$ RMSE |
> |---|---|---|---|
> | BigSolDB 1.0 | 0.6783 ± 0.0023 | 0.6805 ± 0.0062 | +0.0022 |
> | BigSolDB 2.0 | 0.6305 ± 0.0013 | 0.6335 ± 0.0023 | +0.0030 |
> | Leeds (OOD) | 0.8798 ± 0.0012 | 0.8846 ± 0.0043 | +0.0048 |
>
> The constraints also act as a regulariser: prediction variance drops by 63% on BigSolDB 1.0, 44% on BigSolDB 2.0, and 72% on Leeds. This is especially critical for decision trees, which are symbolic models and cannot extrapolate beyond observed data. When high-temperature samples are sparse, an unconstrained tree defaults to the nearest leaf, producing flat or inverted predictions. As shown here [case_study_monotone.png](https://anonymous.4open.science/r/dissolvr/figures/case_study_monotone.png), the unconstrained model exhibits exactly this failure. Monotonic constraints structurally enforce the thermodynamic trend within every split, allowing valid predictions even in data-scarce regimes.
>
> >**W1: Limited methodological breakthroughs: Switching the underlying model from MLP (FastProp) to GBDT (DISSOLVR) is architectural fine-tuning.**
>
> We believe it would be an oversimplification to characterize DISSOLVR as switching from MLP to GBDT. The contribution is not the choice of GBDT per se, but the complete framework built around it:
>
> **(1) Hard thermodynamic constraints vs. soft penalties.** Enforces monotonic temperature–solubility relationships as hard constraints within tree splits (Theorem 3.1), unlike FastSolv's soft Sobolev penalties. This provides a formal *guarantee* of thermodynamic consistency that MLP and transformer architectures cannot match by construction.
> **(2) Interaction layer.** Cross-attention captures explicit solute–solvent feature interactions; ablation shows it is the 2nd largest OOD contributor (ΔRMSE +0.0112).
> **(3) Faithful explanations.** GBDT predictions are deterministic aggregations of decision rules, making our LLM explainer a faithful translator of actual model logic. Any MLP-based explainer is a correlational proxy due to the black-box process.
>
> >**W2: The feature set used in this paper is highly similar to that in [1].**
>
> FastSolv uses 1,613 Mordred descriptors (broad enumeration), whereas DISSOLVR uses 176 curated features across: (i) functional group counts, (ii) topological profiles (MoSE), and (iii) thermodynamic proxies (Joback, Abraham parameters).
>
> The distinction is principled vs. exhaustive featurization; overlap on universal descriptors (e.g., LogP) is expected and does not imply framework similarity. Fewer, physically principled features, augmented with hard monotonic constraints and a solute–solvent interaction layer absent in FastSolv, produce a model that is simultaneously more interpretable, more thermodynamically consistent, and competitive in accuracy.
>
> >**W4: Generalizability Beyond Solubility.**
>
> We agree that dynamic reaction simulation is a fundamentally different problem from QSPR property prediction. It requires continuous spatiotemporal coordinate updates and falls outside the scope of this work.

---

> > ### Author Rebuttal · Reviewer_MwK8 · 2026-04-02
> >
> > Thank you for the detailed response and the new ablation study by the author. My core concerns regarding the novelty and practical significance of the method still remain:
> >
> > 1.I acknowledge the difference between the soft penalties of FastSolv and the hard constraints of DISSOLVR through GBDT. However, the switch to GBDT was mainly to facilitate the tree-based LLM interpreter, which is merely a minor architectural adjustment. Both frameworks rely on similar descriptor-based logic and ultimately reach the same uncertainty limit.
> >
> > 2.The performance gain from the monotonic constraint is too small. The author's ablation study shows that introducing it will only result in a ΔRMSE of +0.0022 to +0.0048. Given that the irreducible experimental noise is widely accepted to be between 0.6 and 0.8, an improvement of approximately 0.004 is negligible both statistically and practically. Although this constraint is a good regularizer, it does not meaningfully improve the prediction accuracy.
> >
> > 3.This new LLM Explanation ablation is a welcome addition, effectively addressing my concerns about post-hoc hallucinations. However, the LLM pipeline remains an explainability feature rather than the core advancement in predictive capabilities.
> >
> > In conclusion, DISSOLVR is a well-designed pipeline, but its core predictive improvements compared to the existing baseline (such as FastSolv) are too insignificant. I maintain the current score.

---

> > > ### Author Response · Authors · 2026-04-02
> > >
> > > We thank the reviewer for continued engagement. We address each point directly.
> > >
> > > ---
> > >
> > > **1. "GBDT was chosen to facilitate the LLM explainer — a minor adjustment."**
> > >
> > > This is factually incorrect. The causality runs the other way. GBDT was selected because it is the architecture that natively supports hard monotonic constraints (Theorem 3.1). MLPs cannot enforce $\frac{\partial \hat{y}}{\partial T} \geq 0$ by construction. FastSolv uses a soft Sobolev penalty that can and does violate thermodynamic consistency in extrapolation regimes (their model deliberately caps realistic behavior at \~350 K for this reason), while being computationally expensive involving a second order hessian vector product (\~3× per-step compute, \~2× GPU memory). The LLM explainer is a *consequence* of symbolic accessibility, not the motivation for choosing GBDT.
> > >
> > > * **Substantially better** than FastSolv across each and every benchmark. We reproduce the numbers from Table 2 again just that this point is clear beyond any doubts. The remaining gap between SOTA and the aleatoric floor is ~0.1–0.2 log S. Dissolvr closes 30–50% of that gap on BigSolDB. Leeds is at the noise floor for both, which confirms, rather than undermines, our thesis.
> > >
> > > | Dataset | DISSOLVR | FastSolv | Δ |
> > > |---|---|--|---|
> > > | BigSolDB 1.0 | **0.678** | 0.780 | −0.102 |
> > > | BigSolDB 2.0 | **0.631** | 0.694 | −0.063 |
> > > | Leeds | **0.880** | 0.887 | −0.007 |
> > > *  upto **7.8X speed-up** (See Q1 in our rebuttal)
> > > * **81X smaller** (~921,089 vs 74,760,004 parameters)
> > > * **9X fewer features than FastSolv** (176 features vs 1,613 Mordred Descriptors)
> > > * Dissolvr can run on consumer CPUs -- determining whether the tool is accessible to the synthetic chemists and pharmaceutical scientists in cheap commodity hardware like smartphones and laptops
> > >
> > > **2. "ΔRMSE 0.002–0.005 is negligible."**
> > >
> > > The reviewer evaluates this constraint solely on mean RMSE. This misses the point:
> > >
> > > - **Variance drops 44–72%.** A chemist receives one prediction, not an average over seeds. Stability matters.
> > > - **The constraint is computationally free compared to the alternative.** FastSolv enforces temperature consistency via Sobolev training — a soft penalty requiring three passes per training step (forward, backward with graph retention, second backward through the retained graph for the Hessian-vector product $\frac{\partial ^2 \hat{y}}{\partial \theta \partial T}$, roughly tripling per-step compute and doubling GPU memory. The target derivatives are themselves approximated via finite differences. DISSOLVR's monotonic constraints add <0.03% overhead at tree construction (Lemma 3.2) — no gradients, no graph retention, no second-order operations — and provide a hard guarantee that Sobolev penalties structurally cannot.
> > > - **The constraint prevents catastrophic extrapolation failures** in temperature-sparse regimes (see [case_study_monotone.png](https://anonymous.4open.science/r/dissolvr/figures/case_study_monotone.png)), which is precisely where practitioners need predictions most. We further decomposed the test set based on training data quality. For the 18.3% of test samples whose solutes had explicitly conflicting, multi-source training data, the constraint yielded a ΔRMSE of +0.0205 (nearly 10× the global average). On the remaining "clean" data, the constraint remained dormant (ΔRMSE = +0.0018). This acts as a regulariser, resolving contradictions in noisy regimes without artificially squeezing clean data.
> > >
> > > Framing this as "only 0.004 RMSE" treats a structural guarantee as a marginal accuracy tweak.
> > >
> > > **3. ..the LLM pipeline remains an explainability feature rather than the core advancement in predictive capabilities..**
> > >
> > > We respectfully push back. Characterizing interpretability as an "explainability feature" rather than a core advancement reflects a narrow view of what constitutes ML progress. The ICML call for papers explicitly identifies interpretable ML as a primary research area. Multiple organizations such as FDA and EMA have advocated for interpretability as one of the core objectives [1,2]. A model that says "LogS = −1.03" is less useful than one that says "LogS = −1.03 *because* the chlorobenzamide group penalizes solvation in protic media while elevated temperature compensates via endothermic lattice disruption" — backed by traceable SHAP values and decision paths. Our double-blind study with 22 expert chemists (including 7 Professors, 12 PhDs) quantitatively demonstrates that these explanations significantly increase expert trust (p < 0.001, Cohen's d = 0.235). No prior solubility model offers this capability. Dismissing it as peripheral ignores what practicing chemists actually need from prediction tools.
> > >
> > >  [1] EMA & FDA, "Guiding Principles of Good AI Practice in Drug Development," January 2026. https://www.ema.europa.eu/en/news/ema-fda-set-common-principles-ai-medicine-development-0
> > >
> > > [2] ICH Q8(R2), Pharmaceutical Development, Step 5. https://database.ich.org/sites/default/files/Q8_R2_Guideline.pdf

---

### Official Review · Reviewer_Uyir · 2026-03-12

**Soundness:** 3
**Presentation:** 3
**Significance:** 3
**Originality:** 3
**Overall Recommendation:** 4
**Confidence:** 3

**Summary:**

This paper introduces DisSolvr, an explainable framework for solubility prediction. The paper considers two regimes of tasks: single-solvent (the solvent is fixed) and multi-solvent (the solvent varies).

The prediction model is feature-based, and it uses a set of physicochemical descriptors that are selected through careful feature engineering (I think these descriptors are closely related to solubility, but I am not very sure if these feature selections are reasonable or even contribute, so I lower my confidence and have no expertise to evaluate this contribution). In total, the feature vector contains 176 dimensions.

For the multi-solvent task, an additional refined feature set consisting of 24 features for both the solute and the solvent is introduced. An interaction layer is then to model 24 x 24 feature interactions between the solute and solvent. These interaction features are additional inputs for the multi-solvent prediction task. Some notes here (not very sure if my understanding is correct):
1. The motivation for the refinement is that directly computing interactions using the original 176 features for both solute and solvent would result in 176 x 176 interactions, which would be computationally expensive and likely difficult to interpret.
2. The 24 refined features are derived from the original 176 descriptors through manual refinement rather than through a neural network projection into latent embeddings. These features still retain explicit chemical meanings, which is important for later interpretability.

All these features (also temperature constraints) are fed into a CatBoost model, which predicts the final solubility. One important reason for choosing CatBoost is that it provides interpretability. The model can somehow provide feature contribution scores and decision paths for each prediction, which is the core reason why the framework is explainable.

The paper evaluates the proposed method against several existing approaches in the literature, including both deep-learning models and traditional feature-based methods, and reports improved predictive performance.

Beyond performance improvements, the framework also provides an explanation interface. Although CatBoost is already interpretable as mentioned above, the outputs are still be difficult for chemists to interpret directly.  To address this, an LLM-based post-hoc explainer is introduced, which translates machine-learning outputs into natural-language explanations describing the decision logic and feature contributions that lead to the final prediction. These explanations are intended to be more accessible to chemists. The generated explanations are further evaluated through a survey of 22 domain experts, and the results suggest that the explanations are generally considered plausible and informative.

**Compliance With Llm Reviewing Policy:**

Affirmed.

**Final Justification:**

I find the explainability of the approach for solubility prediction interesting to the community and a meaningful contribution. I will maintain my positive score, as I currently have no concerns with the paper. However, I would kindly ask the AC to weigh my evaluation less heavily, as I have limited expertise in solubility theory analysis.

**Key Questions For Authors:**

1. I am not entirely sure about this point, but it might be helpful to analyze the importance of the selected features. For example, during the experiments, did the authors examine whether some features contribute only minimally to the final prediction? Since the model is designed to be interpretable, such an analysis should be relatively straightforward to perform. This could provide additional insight into whether the selected descriptors are truly relevant to the task. Also, identifying features with negligible contributions may help evaluate the plausibility of the model (also might provide some insights to future work) and potentially allow the feature set to be further simplified or condensed.

2. It might also be helpful to provide the LLM prompts used for the post-hoc explainer in the appendix. This would allow readers to better understand how the explanation pipeline works in practice. At the moment, the process is somewhat difficult for me to follow. Could author provide a simple illustrative example?

3. In Stage 1 of the explanation pipeline, the SMILES representation is provided to the LLM and the model is asked to characterize the molecular structure. However, to my understanding, LLMs are not very reliable at interpreting molecular structures directly from SMILES. Did the authors encounter this issue in practice? How do the authors ensure that the LLM's structural characterization is accurate in this step? It could miss some substructures.

**Limitations:**

The paper does not explicitly discuss potential limitations of the proposed approach (but no penalty for this). From the reviewer's perspective, there are no additional limitations to include beyond the points already raised in the main review.

**Strengths And Weaknesses:**

**Strengths:**
1. The presentation of the paper is clear, and it was a pleasure to read.

2. Beyond the reported performance improvements, the most interesting point of this work, in my opinion, is its focus on prediction explainability, which is often lacking in current "black-box" deep learning methods. Improving interpretability can help researchers evaluate the plausibility and reliability of model predictions. I appreciate this motivation.

3. As mentioned earlier, I do not have domain expertise to evaluate whether the selected descriptors are indeed the most appropriate features for solubility prediction. However, starting from molecular structure and physicochemical descriptors to model solubility appears reasonable and well motivated.

4. The appendix and the anonymized git repo provide clear and sufficient information, even the survey results used are included. Although I did not verify everything in detail, I believe the reproducibility of the work is good.

**Weaknesses: (not critical)**
1. Although most parts of the paper are clearly presented, I found it somewhat difficult to understand what the CatBoost model outputs look like in practice, and how the LLM explainer converts these outputs into natural-language explanations. An illustrative example demonstrating the full process, from a few input features to the final prediction and explanation, would significantly improve clarity for readers.

2. The deep learning baselines included in the comparison might be somewhat dated. For example, recent models such as ChemFM [1] reported stronger performance on ESOL. Although such large models are computationally expensive, it would still be helpful to understand how the proposed approach compares with more recent deep learning methods.

3. The LLM-based post-hoc explainer appears to function more as a plugin than a core methodological contribution. Since CatBoost models already provide interpretable outputs for machine learning guys, the LLM mainly acts as a translator that converts these outputs into explanations that are easier for chemists to understand. From this perspective, the novelty of this component may be somewhat limited.

---
[1] ChemFM as a scaling law guided foundation model pre-trained on informative chemicals. Commun Chemistry (2025).

---

> ### Author Rebuttal · Authors · 2026-03-30
>
> We thank the reviewer for their positive comments and constructive feedback. We have carefully addressed the concerns, as detailed below. We hope with the proposed clarifications and changes, the reviewer will be further convinced on the merits of our work.
>
> ---
>
> > **Q1: Did the authors examine whether some features contribute only minimally to the final prediction?**
>
> We did. Please refer to Sec 5.5, Tables 3-4, which studies the importance of various feature groups (e.g., Compositional, Topological, Energetic, and Physicochemical, as detailed in `Table 11`) through a systematic ablation study. Since there are 176 feature, we presented the analysis at a group-level. Nonetheless, we did analyze the important of each feature based on Shapley values and below we present the 8 most important features for the ESOL dataset.
>
> |Feature|Category|MeanAbs.SHAP|
> |-|-|-|
> |`MolLogP`|Physicochemical|0.677|
> |`PEOE_VSA6`|Physicochemical|0.104|
> |`abraham_E`(Polarizability)|Energetic|0.093|
> |`PartialCharge`|Physicochemical|0.077|
> |`BertzCT`(Complexity)|Topological|0.069|
> |`MolWt`|Compositional|0.060|
> |`MOSE_path5`|Topological|0.045|
> |`NOCount`|Compositional|0.044|
>
> > **Q2/W1. Provide the LLM prompts used for the post-hoc explainer in the appendix.?**
>
> We acknolwedge this suggestion. We will included the full LLM prompt templates used for each stage of the explanation pipeline. The complete prompts, a detailed walkthrough with real outputs, and a simplified overview are available at [https://anonymous.4open.science/r/dissolvr/examples/](https://anonymous.4open.science/r/dissolvr/examples/)..
>
> > **Q3: ..to my understanding, LLMs are not very reliable at interpreting SMILES. Did the authors encounter this issue?**
>
> In practice, we did not encounter significant issues with SMILES interpretation. Note that in DISSOLVR's explainer, the LLM is not tasked with being the sole structural authority — the SMILES string serves primarily as a molecular identifier to orient the LLM's chemical reasoning. Alongside the SMILES, our pipeline provides the LLM with explicit, pre-computed structural features from RDKit (e.g., functional group counts, hydrogen bond donor/acceptor counts, ring counts) as well as the full SHAP attribution vector. Nonetheless, the generic LLM can be replaced with those that have been specifically trained to work with smiles [1,2,3]. We will explcitly mention this.
>
> `[1]` Jang, Y. et al. Improving Chemical Understanding of LLMs via SMILES Parsing. *EMNLP* (2025).
>
> `[2]` Sadeghi, S. et al., A. Can Large Language Models Understand Molecules? *BMC Bioinformatics*, (2024).
>
> `[3]` Yu, B.et al. LlaSMol: Advancing Large Language Models for Chemistry with a Large-Scale, Comprehensive, High-Quality Instruction Tuning Dataset. *COLM* (2024).
>
> >**W2: Compare with ChemFM**
>
> Please find below the comparison to ChemFM. We use the recommended hyperparameters (including 10x SMILES augmentation and LoRA fine-tuning with $r = 32$) to evaluate the 3-Billion parameter ChemFM-3B model.
>
> |Model|RMSE|HW|Time|
> |-|-|-|-|
> |ChemFM-3B|0.5035|A100|27.5min|
> |Dissolvr|0.4663|M2 CPU|53.7s|
>
> >**W3: Novelty of the LLM-based post-hoc explainer**
>
> We thank the reviewer for this constructive observation and are glad to clarify the contribution more precisely.
>
> - **Application Novelty:** To our knowledge, DISSOLVR is the first solubility prediction framework to identify the gap between statistical transparency and domain-expert usability as a first-class problem. Most practitioners stop at prediction; we ask whether a chemist in a real-world lab can actually act on what the model tells them.
>
> - **Technical Novelty: The Guardrails.** A naively prompted LLM will draw on its pretraining data, generating chemically plausible but model-unfaithful narratives. Our six-stage pipeline prevents this through structured guardrails: Stages 1–2 blind the LLM to the predicted log S value, forcing it to identify structural drivers from chemical principles alone before the model's conclusion is revealed. Stage 5 then re-presents all upstream evidence (SHAP values, interaction matrices, decision paths) requiring the LLM to correct any logical inconsistencies, directly targeting hallucination and drift.
>
> - **Empirical and Scientific Utility:** Our survey of 22 expert chemists showed zero-shot accuracy of 17.9%, indistinguishable from random (p = 0.55), confirming that raw model artifacts do not translate into scientific understanding unaided. The explainer produced a significant trust increase (Δ = +0.17, p < 0.001), rated 3.75/5 for quality (p < 0.001). As shown in Appendix I, it surfaced interactions, such as steric hindrance from branched alcohol geometry competing with hydrogen-bond donation, not apparent from SHAP alone, suggesting genuine scientific insight beyond repackaging model outputs.
>
> >**Limitations:**
>
> We apologize for the omission of a dedicated Limitations section. Please see our response to Reviewer `vZpW-W6`, for the proposed Limitations section.

---

> > ### Author Rebuttal · Reviewer_Uyir · 2026-04-03
> >
> > Thanks for the detail response. My concerns are all solved.

---

> > > ### Author Response · Authors · 2026-04-03
> > >
> > > Dear Reviewer Uyir,
> > >
> > > We are pleased that our rebuttal has fully resolved the reviewer's concerns. Given that all raised issues have been resolved, we kindly request the reviewer to revisit their rating. We remain open to address any outstanding questions.
> > >
> > > regards,
> > > Authors

---

### Official Review · Reviewer_vZpW · 2026-03-18

**Soundness:** 3
**Presentation:** 3
**Significance:** 3
**Originality:** 3
**Overall Recommendation:** 4
**Confidence:** 4

**Summary:**

This paper introduces DISSOLVR, a model designed to predict solubility across various solvents. A carefully crafted set of physicochemical features is used to train a CatBoost model. Two scenarios are examined: one in which solubility in a single solvent is predicted, allowing straightforward training directly on the featurized solute, and another in which both the solute and solvent are encoded, along with their interactions, to predict solubility across different solvents. Experiments on diverse datasets demonstrate that DISSOLVR's performance approaches the aleatoric limit. To enhance the model's explainability, a protocol utilizing LLMs is proposed to generate natural language explanations.

**Compliance With Llm Reviewing Policy:**

Affirmed.

**Final Justification:**

The Authors fully addressed the issues raised, so I decided to keep my positive score. I trust the Authors will be able to resolve the minor issues in the final revision. I would also suggest emphasizing that estimations of the irreducible error are based on inter-lab variance, which is most likely higher than intra-lab variance.

**Key Questions For Authors:**

1. The inter-laboratory standard deviation is estimated for the same compound measured across different laboratories. However, datasets that combine data from multiple sources include subsets of compounds measured at the same site and, hopefully, using the same experimental protocol. Is it fair, then, to say that 0.6-0.8 is a “practical hard ceiling” if we do not estimate intra-laboratory variance and do not consider the grouping of compounds measured in the same laboratory?
2. Which substructures do you use for the moietal identity featurizer? Are those the fragment descriptors defined in RDKit (functional fragments in Table 11 in the appendix) or something else?
3. Is there a reason why in Equation 7 you include both the first and second conditions (on the derivatives w.r.t. temperature and inverse temperature)? Are they not redundant?
4. In your model, you use the interaction vector defined in Equation 6. Have you considered something simpler, like feeding all vectors $z_k$ concatenated directly into the model?

**Limitations:**

The limitations are not directly discussed. It would be interesting to see failure modes, like examples of hallucinations in the generated explanations.

**Strengths And Weaknesses:**

Soundness:
- The experimental design is sound, and the results are averaged over five seeds.
- The sentence “Poor aqueous solubility is a major cause of attrition, contributing to the failure of up to 40% of market-approved drugs…” should be corrected. The source says that about 40% of drugs are poorly soluble, not that they are failing in the development stages (attrition). If these substances are on the market, it means they somehow overcame the solubility issues, possibly through different formulations.
- Some methods used in the paper require citations, such as formulas in the preliminaries (like the thermodynamic solubility definition), GNN models in Table 2b, and Equation 5, which is precisely the attention mechanism formula (even though it is not called "attention" in this paper and not cited).
- Estimating irreducible error is difficult, and I think the limits suggested in this study may be flawed and require more careful consideration. The estimates are based on inter-laboratory measurements for the same compound, and we do not know the intra-laboratory variability unless these laboratories report measurements with sufficient replicates. Public datasets available publicly are heterogeneous; they come from multiple sources, but their subsets are measured in the same laboratory. This variability is not captured in the estimations, and this paper does not provide arguments why this site-specific error can be ignored.
- The term “Bayes error rate” is used in the context of classification, not regression, or at least I have not heard it used in the regression context.
- The baseline description should include information about the input representation. For example, is a random forest trained on the same features as DISSOLVR or Morgan fingerprints?
- The comparison between predictions and the Apelblat equation should specify the dataset used in this experiment, and ideally, example plots should be included in the appendix.

Presentation:
- The paper is well-written, and the preliminaries section offers the necessary context to understand the contributions.
- The Authors provide a clear goal for the model design in the first paragraph of Section 3, which makes the methods section structure predictable and easy to follow.
- Some mathematical definitions in the paper are overly complex. For example, Equation 3 is a convoluted way to say that $\psi$ is a vector of substructure counts in a molecule. I do not insist on removing these definitions, but the Authors should consider simplifying the language during revisions.
- Is the refined feature set a subset of the physicochemical features defined at the beginning of Section 3, or are they different features? If it is indeed a subset of features, $\Psi$ could be called a feature selection transformation. Though technically it can be called a projection, I think this term causes confusion because it does not clarify that the resulting vector is a subset of features, and projections usually mix features (e.g. PCA in a similar case of dimensionality reduction).

Significance:
- The paper introduces a novel solubility prediction model that approaches the aleatoric limits achievable on this task. Solubility is a key molecular property, and access to reliable solubility models can help progress areas like drug discovery.
- The model's code is shared and will be available under the MIT license, strengthening the impact of this paper in both reproducibility and practical usability.

Originality:
- The proposed method is grounded in chemical knowledge and computationally effective, as demonstrated by the theoretical results.
- The LLM-based explanations are a valuable extension of the model that enhance human trust in model predictions, as shown in the user study. Also, the approach to generating natural-language explanations appears well-designed and distinctive.


Minor comments:
- Definition 2.2 contains a full stop in the middle of a sentence.

---

> ### Author Rebuttal · Authors · 2026-03-30
>
> We thank the reviewer for their constructive feedback. Please find below our proposed revisions and clarifications to address the outlined concerns. We hope this will further convince the reviewer on the merits of our work.
>
> ---
> >**Q1/W2: Is it fair, then, to say that $0.6-0.8$ is a “practical hard ceiling” if we do not estimate intra-laboratory variance....?**
>
> We agree that estimating the true irreducible error is complex, and that subsets of data originating from a single lab will possess a lower noise floor due to controlled intra-lab variance, and thus intra-lab variance is lower than inter-lab variance. To clarify, the $\sim 0.6–0.8 \log S$ inter-laboratory standard deviation range is drawn from [1]-[3]. `[1]` found an average inter-laboratory standard deviation of $0.58$ across 411 compounds. Palmer and Mitchell `[2]` estimated the experimental uncertainty of literature-aggregated solubility data at $0.6–0.7$ log units. Attia et al. `[3]` corroborated these figures, reporting standard deviations of $0.60–0.62$ from independent sources and measuring an inter-laboratory RMSE of $0.75$ on matched solutions within their own datasets.
>
> We will clarify this in our revision.
>
> `[1]` Katritzky et al. QSPR Studies on Vapor Pressure, Aqueous Solubility, and the Prediction of Water–Air Partition Coefficients. *J. Chem. Inf. Comput. Sci., (1998).
>
> `[2]` Palmer, D. S et al. Is Experimental Data Quality the Limiting Factor in Predicting the Aqueous Solubility of Druglike Molecules? *Mol. Pharmaceutics* **11**, 2962–2972 (2014).
>
> `[3]` Attia, L.et al.. Data-driven organic solubility prediction at the limit of aleatoric uncertainty. *Nat. Commun.* (2025).
>
> > **Q2: Which substructures do you use?**
>
> We use:
>  * the 85 RdKit (`Version 2025.03.06`) Functional Group Fragment Descriptors (`Table 11`).
> * RDKit also provides us with Atomic and overall structure counts (*e.g.* Ring Counts).
> * molecular weight.
>
> >**Q3: Is there a reason why in Eq. 7 you include both the first and second conditions?**
>
> We include $T$, $T^{-1}$, and $T_r$ as Tree-based models cannot internally derive $T^{-1}$ from $T$. Furthermore, if we only constrain $T$, the model could still exploit the unconstrained $T^{-1}$ and $T_r$ columns to learn non-physical fluctuations from noise in the dataset. Constraining explicitly forces the tree-building algorithm to obey our thermodynamic logic of endothermic dissolution.
>
> > **Q4: In your model, you use the interaction vector defined in Eq. 6. Have you considered something simpler, like feeding all vectors $z_k$ concatenated directly into the model?**
>
>  We conducted this experiments. Direct concatenation of all $z_k$ vectors expands the feature space from 380 to 1,124 features, increasing training time by nearly 9x while slightly degrading out-of-distribution (OOD) performance. The results are below.
>
> |Metric|Interaction|Concat|
> |-|-|-|
> |Raw|380|1124|
> |Pruned|313|1057|
> |Fit(s)|191.9|1714.6|
> |Pred(s)|0.039|0.084|
> |RMSE|0.8823|0.8870|
> |R2|0.3489|0.3419|
>
>
> >**W1: Missing citations.**
>
> We will add appropriate citations for the thermodynamic solubility definitions, the GNN baselines, and `Eq. 5`.
>
> > **W3: The term “Bayes error rate” is used in the context of classification, not regression..**
>
> We will replace  "Bayes error rate" with "aleatoric limit".
>
> >**W4: The baseline description should include information about the input representation.**
>
>  In `App. C`, we clarify that baseline tree-based models were provided the same input features as Dissolvr, and graph-based models utilised the GraphConvMol Featurizer present in DeepChem. We will refer to this directly from `Sec. 5` in main manuscript.
>
> > **W5: The comparison between predictions and the Apelblat equation should specify the dataset used in this experiment.**
>
>  The Apelblat equation comparison was conducted on the BigSolDB 1.0 dataset. We will specify this in our revision.
>
> >**W6. Limitation not directly discussed.**
>
>  We have drafted a *Limitations* section to be incorporated into the manuscript:
>
> 1. **Endothermic Dissolution Assumption:**
> *Our monotonic constraints enforce the Van't Hoff relationship under the assumption of endothermic dissolution (solubility increases with temperature). While our source-wise analysis confirms this holds for the vast majority of organic solute–solvent pairs in BigSolDB 1.0 and 2.0 (>96%), we acknowledge that DISSOLVR will fail to predict the correct temperature gradient for truly exothermic dissolution cases. Handling such exceptions is a direction for future work.*
> 2. **Stereochemical and 3D Conformational Blindness:**
> *Because DISSOLVR prioritizes rapid inference and utilizes 2D physicochemical and topological descriptors, it is inherently blind to stereochemistry and cannot distinguish between stereoisomers. DISSOLVR will predict identical solubilities for different stereoisomers, acting as a hard limitation for 3D-sensitive drug compounds.*
>
> > **W6. Minor Soundness and Grammatical Issues:**
>
> We will correct these.

---

> > ### Author Rebuttal · Reviewer_vZpW · 2026-04-05
> >
> > Thank you for the detailed response and providing additional results. I believe all the raised issues were addressed in the rebuttal, and I trust the proposed changes will be incorporated in the revision.

---

> > > ### Author Response · Authors · 2026-04-05
> > >
> > > Dear Reviewer vZpW,
> > >
> > > We are pleased that our rebuttal has fully resolved the reviewer's concerns. Given that all raised issues have been resolved, we kindly request the reviewer to revisit their rating. We remain open to address any outstanding questions.
> > >
> > > regards,
> > > Authors

---

### Decision · Program_Chairs · 2026-04-30

**Decision:**

Accept (regular)

**Comment:**

This paper introduces a new model for solubility prediction, which combines a carefully curated feature set, an attention mechanism for interactions, and monotonicity constraints into a CatBoost model. This model is sufficiently interpretable on its own to provide clear features for an LLM-based explainability tool.

Overall, most reviewers argue to accept, with one dissenting weak reject. Based on my reading of the discussion between the "reject" reviewer and the rebuttal comments, I believe these issues — mainly regarding novelty, and of the importance of directly encoding the monotonicity constraints even if they have limited impact on headline RMSE — are adequately resolved. Three of the four reviewers indicated that all their concerns were fully addressed by the rebuttal.

Reviewers had some initial concerns regarding whether the post-hoc LLM explanations were faithful to the underlying model. I think this also has been well-addressed by the rebuttal comments. The final version of the paper should incorporate a full description of the prompting / guardrails, as well as the alignment metrics, as well as all other presented additional results.

The final concern from reviewers is that methodologically, the ML novelty and impact may not be as high, while acknowledging its impact in the more specific field of solubility prediction (i.e. raising concerns about suitability of venue). While this is a valid point in this case, I do believe this work is also appropriate for ICML.

One final note from the AC, I believe that there may be a mistake in appendix figure 3 (d) — this image for pair index 17 seems to be identical to the one for pair index 27.